# Essential metabolism for a minimal cell

Marian Breuer[1†], Emmy E Earnest[1‡], Chuck Merryman[2], Kim S Wise[2], Lijie Sun[2], Michaela R Lynott[2§], Clyde A Hutchison[2], Hamilton O Smith[2], John D Lapek[3#], David J Gonzalez[3], Valérie de Crécy-Lagard[4], Drago Haas[4¶], Andrew D Hanson[5], Piyush Labhsetwar[1**], John I Glass[2], Zaida Luthey-Schulten[1*]

[1]Department of Chemistry, University of Illinois at Urbana-Champaign, Urbana, United States; [2]J Craig Venter Institute, La Jolla, United States; [3]Department of Pharmacology and School of Pharmacy, University of California at San Diego, La Jolla, United States; [4]Department of Microbiology and Cell Science, University of Florida, Gainesville, United States; [5]Horticultural Sciences Department, University of Florida, Gainesville, United States

**Abstract** JCVI-syn3A, a robust minimal cell with a 543 kbp genome and 493 genes, provides a versatile platform to study the basics of life. Using the vast amount of experimental information available on its precursor, *Mycoplasma mycoides capri*, we assembled a near-complete metabolic network with 98% of enzymatic reactions supported by annotation or experiment. The model agrees well with genome-scale in vivo transposon mutagenesis experiments, showing a Matthews correlation coefficient of 0.59. The genes in the reconstruction have a high in vivo essentiality or quasi-essentiality of 92% (68% essential), compared to 79% in silico essentiality. This coherent model of the minimal metabolism in JCVI-syn3A at the same time also points toward specific open questions regarding the minimal genome of JCVI-syn3A, which still contains many genes of generic or completely unclear function. In particular, the model, its comparison to in vivo essentiality and proteomics data yield specific hypotheses on gene functions and metabolic capabilities; and provide suggestions for several further gene removals. In this way, the model and its accompanying data guide future investigations of the minimal cell. Finally, the identification of 30 essential genes with unclear function will motivate the search for new biological mechanisms beyond metabolism.

**\*For correspondence:**
zan@illinois.edu

**Present address:** [†] Maastricht Centre for Systems Biology (MaCSBio), Maastricht University, Maastricht, The Netherlands; [‡] SimBioSys Inc, Champaign, United States; [§] CB Therapeutics, San Diego, United States; [#] Pfizer Inc, La Jolla, United States; [¶] Sanofi, Vitry-sur-Seine, France; [**] The Land Institute, Salina, United States

## Introduction

Establishing the core requirements of cellular life is a fundamental challenge of biology. The question of the minimal set of biochemical functions necessary for a cell to grow and replicate has been studied from a number of angles for more than 20 years. It has long been suggested (*Morowitz, 1984*) that a model to study the basics of cellular life would be the mycoplasmas—a group of bacteria with small genomes (580–1350 kbp (*Herrmann, 1992*; *Fraser et al., 1995*)) lacking a cell wall, which evolved via extreme genome reduction from low GC content Gram-positive ancestors (*Pollack et al., 1997*). Mycoplasmas exist as parasites or saprotrophs and are adapted to scavenging nutrients and cellular building blocks from their niche environments, which enabled them to lose many metabolic capabilities.

The genome of the human urogenital pathogen *Mycoplasma genitalium* (580 kbp, 525 genes overall, 482 for proteins, 43 for RNAs), sequenced in 1995 (*Fraser et al., 1995*), is the smallest genome of any autonomously replicating cell found in nature and has thus been deemed a close approximation to a minimal genome (*Glass et al., 2006*). In particular, different efforts have been undertaken to establish a minimal set of genes based on the near-minimal *M. genitalium* genome. A comparison of the first two sequenced bacterial genomes (the Gram-positive *M. genitalium* (*Fraser et al., 1995*) and the Gram-negative *Haemophilus influenzae* (*Fleischmann et al., 1995*))

**eLife digest** One way that researchers can test whether they understand a biological system is to see if they can accurately recreate it as a computer model. The more they learn about living things, the more the researchers can improve their models and the closer the models become to simulating the original. In this approach, it is best to start by trying to model a simple system.

Biologists have previously succeeded in creating 'minimal bacterial cells'. These synthetic cells contain fewer genes than almost all other living things and they are believed to be among the simplest possible forms of life that can grow on their own. The minimal cells can produce all the chemicals that they need to survive – in other words, they have a metabolism. Accurately recreating one of these cells in a computer is a key first step towards simulating a complete living system.

Breuer et al. have developed a computer model to simulate the network of the biochemical reactions going on inside a minimal cell with just 493 genes. By altering the parameters of their model and comparing the results to experimental data, Breuer et al. explored the accuracy of their model. Overall, the model reproduces experimental results, but it is not yet perfect. The differences between the model and the experiments suggest new questions and tests that could advance our understanding of biology. In particular, Breuer et al. identified 30 genes that are essential for life in these cells but that currently have no known purpose.

Continuing to develop and expand models like these to reproduce more complex living systems provides a tool to test current knowledge of biology. These models may become so advanced that they could predict how living things will respond to changing situations. This would allow scientists to test ideas sooner and make much faster progress in understanding life on Earth. Ultimately, these models could one day help to accelerate medical and industrial processes to save lives and enhance productivity.

yielded 256 orthologous genes that were suggested to approximate a minimal set of bacterial genes (*Mushegian and Koonin, 1996*); a subsequent comparative study, including genomes from several free-living and endosymbiotic bacteria, proposed a minimal set of 206 genes (*Gil et al., 2004*). A limitation of this approach lies in the possibility of the same function being fulfilled by non-orthologous proteins in different organisms, in which case it would not be captured by searching for orthologous genes. Transposon mutagenesis studies (*Hutchison et al., 1999*) avoid this drawback by directly probing the dispensability of individual genes in a single organism via random gene disruption, and testing for viability. In *M. genitalium*, such studies have suggested 382 out of the 482 protein-coding genes to be essential (*Glass et al., 2006*).

An important limitation of deriving a minimal gene set from essentiality information on individual genes lies in the fact that more than one gene can fulfill the same function, and while neither gene is essential individually, at least one of them has to be present in a functional minimal genome. Thus, while removal of either gene would be nonlethal, removing both would create a synthetic lethality. This problem can, in principle, be circumvented by sequential gene deletion starting from a given wild-type organism (as partially done for *Escherichia coli* and *Bacillus subtilis* (*Juhas et al., 2014*; *Pósfai et al., 2006*)), with testing for viability and growth rate after each deletion. In principle, this would not only yield the information on a minimal genome, but also would produce a living organism controlled by such a genome. However, the time and resource costs of minimizing a genome by serial deletion of dispensable genes are prohibitive.

In 2016, we developed a successful bottom-up approach to design a minimal genome and create a living cell controlled by it (*Hutchison et al., 2016a*). Starting with the gene sequence from the 1079 kbp genome of the ruminant pathogen *Mycoplasma mycoides capri* serovar LC GM12, a minimal genome of 531 kbp was designed and constructed containing 473 genes (438 protein-coding genes and 35 genes for RNAs) (*Hutchison et al., 2016a*). The resulting strain, JCVI-syn3.0 (NCBI GenBank: https://www.ncbi.nlm.nih.gov/nuccore/CP014940.1 (*Hutchison et al., 2016b*)), has a genome smaller than that of any independently-replicating cell found in nature and is considered to be our 'working approximation to a minimal cell'. This achievement was the culmination of a series of breakthroughs in synthetic biology. In 2007, the successful transplantation of an *M. mycoides capri* LC GM12 genome into a *Mycoplasma capricolum* recipient cell was reported (*Lartigue et al.,*

*2007*), transforming the recipient cell to the species of the implanted DNA. In 2008, the complete genome of *M. genitalium* was synthesized from scratch, starting with chemically synthesized oligonucleotides and stepwise recombination in vitro and subsequently in *Saccharomyces cerevisiae* (yeast), using the available genetic manipulation tools to assemble the genome as a plasmid inside the yeast cell (*Gibson et al., 2008*). These methods enabled the construction of JCVI-syn1.0, the first cell controlled by a synthetic genome (NCBI GenBank: https://www.ncbi.nlm.nih.gov/nuccore/CP002027.1) (*Gibson et al., 2010a*; *Gibson et al., 2010b*). This was accomplished by synthesizing of a copy of the *M. mycoides capri* LC GM12 genome along with vector sequences that allowed cloning in *E. coli* and yeast, and its subsequent transplantation into *M. capricolum* recipient cells to yield JCVI-syn1.0. These milestones enabled the synthesis of reduced versions of the JCVI-syn1.0 genome with subsequent transplantation into *M. capricolum* to test for viability. The genome reduction process was guided by transposon mutagenesis studies on the original JCVI-syn1.0 genome, as well as on intermediate reduced genome versions. Successful genome minimization depended on identifying both essential genes, whose disruption is immediately lethal, and quasi-essential genes, whose disruption causes an observable growth disadvantage. Quasi-essential genes were identified by observing if cells with potentially growth-reducing gene disruptions were outgrown during sufficiently long competition experiments, so that cells sampled from much later generations no longer contained the disrupted gene. Three cycles of genome design, assembly and growth testing yielded JCVI-syn3.0 (*Hutchison et al., 2016a*).

JCVI-syn3.0 contains all the genes of JCVI-syn1.0 that are required for growth. This includes both essential and quasi-essential genes. Individually non-essential genes were removed in the design for JCVI-syn3.0 to the greatest extent possible without causing synthetic lethality or a major growth disadvantage from simultaneous knockouts. However, in a few cases, genes that appear to be non-essential were retained for ease of genome design and construction. Intriguingly, the role of a considerable fraction of the minimal genome of JCVI-syn3.0 could not be elucidated in spite of extensive bioinformatic analyses. At the time of publication of the minimal cell, 149 of the genes (~31% of the genome) could not be assigned a completely specific biological function. Assignment to a broad functional category could not even be made for a subset of 79 genes. These genes of unknown or poorly defined function potentially point toward required features of cellular life yet to be discovered.

The original minimal cell JCVI-syn3.0 genome was assembled by combining individually minimized 1/8 chromosome segments (*Hutchison et al., 2016a*). Phenotypic traits of JCVI-syn3.0 included extensive filamentation and vesicle formation during growth, and a doubling time of 2–3 hr (compared to the spheroidal morphology and 1 hr doubling time conferred by the JCVI-syn1.0 genome). To address these phenotypic disadvantages, an alternative design of segment six was found to restore consistent morphologic features and increase the growth rate, while retaining a near-minimal genome. This new design incorporated 19 additional genes from JCVI-syn1.0 segment six that were not present in JCVI-syn3.0, including those encoding the cell partitioning proteins FtsZ and SepF along with others of unknown function; in addition, two other genes retained in JCVI-syn3.0 segment six were removed. The complete genome sequence of this strain, termed JCVI-syn3A, is available through NCBI under the accession number https://www.ncbi.nlm.nih.gov/nuccore/CP016816.2. (*Glass, 2017*) This entry contains 498 genomic features, however three of those are pseudo-genes and two are genes required for cloning in yeast.

JCVI-syn3A has a doubling time of ~2 hr and consistently forms spherical cells of approximately 400 nm in diameter. With a 543 kbp genome containing 493 genes of which 452 code for proteins and 38 for RNAs, JCVI-syn3A still has a smaller genome than any natural autonomously replicating cell while providing a robust and versatile platform to study the basics of life. In particular, this minimal cell opens up the possibility to pursue the construction of a complete in silico model including the function of all genes. The map of protein coding genes (*Figure 1*) clearly shows the fundamental importance of Syn3A as a platform to study the principles of life. Among the model bacteria *E. coli* and the related and well-studied (*Güell et al., 2009*; *Kühner et al., 2009*; *Yus et al., 2009*; *Maier et al., 2011*; *Wodke et al., 2013*) *Mycoplasma pneumoniae*, JCVI-syn3A has the smallest ratio of genes involved in metabolism to those in genetic information processing. With 91 it also has the smallest number of genes that are considered to have no known (unclear) function compared to 311 and 1780 for *M. pneumoniae* and *E. coli*, respectively (see *Table 1* and

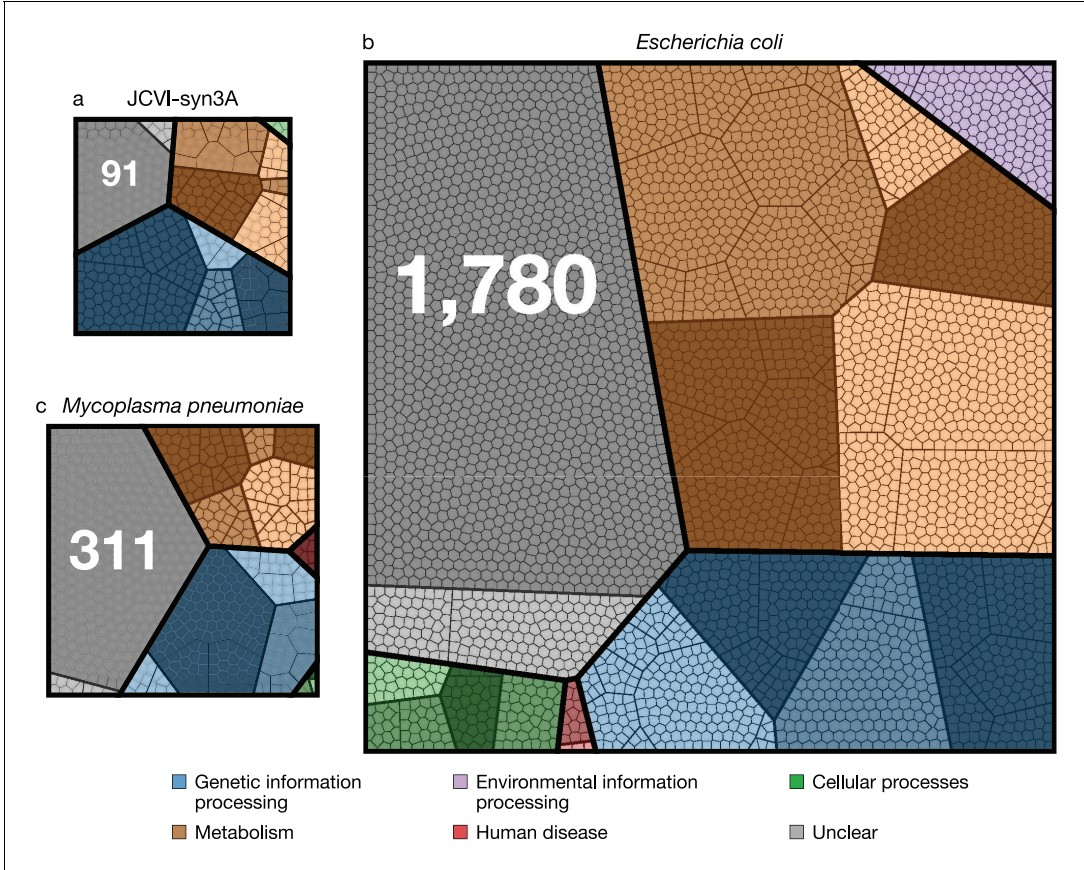

**Figure 1.** Comparison of protein coding genes in the genomes of JCVI-syn3A (NCBI GenBank: https://www.ncbi.nlm.nih.gov/nuccore/CP016816.2 (*Glass, 2017*)), *M. pneumoniae* (NCBI GenBank: https://www.ncbi.nlm.nih.gov/nuccore/U00089.2 (*Himmelreich et al., 2014*)), and *E. coli* (NCBI GenBank: https://www.ncbi.nlm.nih.gov/nuccore/NC_012967.1 (*Jeong et al., 2017*)) with 452, 688, and 4637 coding genes, respectively. Each color represents a primary functional class, each contiguous shaded region corresponds to a secondary functional class, within each of the shaded regions the bold lines separate tertiary functional classes, finally each polygonal cell represents a single gene. The functional class hierarchy is presented in *Supplementary file 1A*. The ratio of metabolic to genetic information processing genes—0.67, 0.79, and 2.23 respectively—is smallest for JCVI-syn3A. The JCVI-syn3A genome contains both the smallest absolute number of genes of unclear function and the smallest percentage, 91 (20 %), compared to *M. pneumoniae* with 311 (45 %) and *E. coli* with 1780 (38 %).

*Supplementary file 1C–1D* for an itemized account of the functional categories for the three genomes).

A model for ribosome biogenesis that includes DNA replication, transcription, translation, and ribosome assembly in slow growing *E. coli* has already been developed (*Earnest et al., 2015*; *Earnest et al., 2016*). As its components have on average 50% sequence identity to those genetic information processing genes in JCVI-syn3A, this model is assumed to be applicable to JCVI-syn3A as well. Hence, the next important step in modeling JCVI-syn3A is the reconstruction of its metabolic network.

The metabolic reconstruction presented here is based on the curated genome annotation, extensive experimental information from the literature on *M. mycoides capri* and other mycoplasma species, and accompanying transposon insertion and proteomics data. Our model features 338 reactions organized in nine subsystems (see *Supplementary file 1B*), involving 304 metabolites, catalyzed by gene products of 155 genes, thus covering a third of the genes of JCVI-syn3A. The reconstruction process enabled us to suggest several annotation refinements and updates, and yielded a metabolic network that is fairly complete.

Together with the reconstructed biomass composition of JCVI-syn3A and estimates of its reaction flux constraints and energy expenses, the reconstructed metabolic network was cast into a flux-balance analysis (FBA) model (*Orth et al., 2010*). FBA yields the set of steady-state reaction fluxes

**Table 1.** Breakdown of protein coding genes in JCVI-syn3A into functional classes.

| Functional hierarchy | | Protein | | Genes | | Essentiality | | | |
|---|---|---|---|---|---|---|---|---|---|
| | | % | # unique | % | # unique | # E | # Q | # N | # model |
| Cellular processes | Cell Growth | 1.02 | 4 | 0.88 | 4 | 1 | 0 | 3 | 0 |
| | Defense | 0.23 | 2 | 0.44 | 2 | 1 | 0 | 1 | 1 |
| | *Subtotal* | 1.25 | 6 | 1.33 | 6 | 2 | 0 | 4 | 1 |
| Genetic information processing | DNA Maintenance | 5.07 | 38 | 8.41 | 38 | 25 | 9 | 4 | 3 |
| | Folding, Sorting and Degradation | 9.58 | 25 | 5.53 | 25 | 18 | 7 | 0 | 7 |
| | Transcription | 3.92 | 14 | 3.32 | 15 | 8 | 5 | 2 | 0 |
| | Translation | 39.5 | 129 | 29.7 | 134 | 95 | 28 | 11 | 25 |
| | *Subtotal* | 58.1 | 206 | 46.9 | 212 | 146 | 49 | 17 | 35 |
| Metabolism | Biosynthesis | 4.27 | 29 | 6.86 | 31 | 26 | 4 | 1 | 27 |
| | Central Carbon Metabolism | 16.4 | 46 | 10.4 | 47 | 26 | 10 | 11 | 44 |
| | Energy Metabolism | 0.47 | 4 | 0.88 | 4 | 2 | 1 | 1 | 1 |
| | Membrane Transport | 9.37 | 54 | 12.6 | 57 | 37 | 16 | 4 | 46 |
| | Other Enzymes | 1.12 | 4 | 0.88 | 4 | 2 | 1 | 1 | 1 |
| | *Subtotal* | 31.6 | 137 | 31.6 | 143 | 93 | 32 | 18 | 119 |
| Unclear | Kegg ortholog defined | 1.04 | 8 | 1.77 | 8 | 3 | 2 | 3 | 0 |
| | No Kegg ortholog | 7.98 | 71 | 18.4 | 83 | 27 | 30 | 26 | 0 |
| | *Subtotal* | 9.02 | 79 | 20.1 | 91 | 30 | 32 | 29 | 0 |
| *Total* | | 100. | 428 | 100. | 452 | 271 | 113 | 68 | 155 |

through a metabolic network that maximize a pre-defined objective function, for example production of cellular biomass. The solution space of possible fluxes is constrained not only by the steady-state assumption, but also by specific flux limits accounting for maximal uptake/secretion rates or cellular energy expenses. If these flux limits are not known, the network stochiometry predicts the biomass yield achieved by the cell, that is gram biomass produced/gram carbon source taken up (or equivalently biomass production rate/carbon substrate uptake rate). If flux constraints, in particular substrate uptake rates are known or can be assumed, the yield as growth rate per uptake rate can be converted to an absolute growth rate. While measurements to derive such flux constraints are not available yet for JCVI-syn3A, some measurements are available from other mycoplasmas and bacteria that have the same high-affinity glucose transporter (PtsG) found in JCVI-syn3A. Using the glucose uptake rate measured in *M. pneumoniae* (**Wodke et al., 2013**) (which is similar to the one measured in slow-growing *E. coli* (**Fuhrer et al., 2005**)) and other constraint estimates allows us to provisionally predict a growth rate for JCVI-syn3A; this model growth rate is however sensitive to the assumed uptake rate (see Section 'Sensitivity analysis' in Appendix 1). In this article, the growth rate predicted by the model is presented with the understanding that for the aforementioned reasons, the prediction is provisional and comes with a degree of uncertainty. This uncertainty has no bearing on the prediction of in silico gene essentialities (see below), which can be obtained by removing certain genes in the model and their associated reactions, and testing whether FBA still predicts a nonzero growth rate for the resulting in silico knockout.

This FBA model for JCVI-syn3A allows for the analysis of the properties of minimized metabolism in JCVI-syn3A. In particular, gene essentiality can be compared between the model and experimental transposon mutagenesis data. Random gene disruption by bombardment with transposon insertions (**Hutchison et al., 1999**; **Glass et al., 2006**) was used to identify non-essential genes in JCVI-syn1.0 that to the most part were removed during the construction of JCVI-syn3.0 (**Hutchison et al., 2016a**); here, genome-scale transposon mutagenesis studies were carried out on JCVI-syn3A to survey the individual essentiality of all its remaining genes. We find that transposon- and model-derived gene essentiality agree well, with every in silico essential gene being at least quasi-essential in vivo (i.e. removal might not be immediately detrimental, but give a growth disadvantage). The metabolic reconstruction allows us to rationalize the non-essentiality of some genes, and to propose possible

further gene removals in JCVI-syn3A. These suggestions from the model are of particular interest as transposon mutagenesis experiments only probe the individual essentiality of genes and do not yield information on which genes could be removed simultaneously. The metabolic construction, on the other hand, allows us to suggest which genes might be simultaneously removed. At the same time, in silico and in vivo essentiality also show some discrepancies, which lead us to postulate new hypotheses about specific gene functions or metabolic capabilities. Protein expression profiles of essential and non-essential genes, classified by either transposon mutagenesis studies or FBA in silico gene knockouts, were not found to differ significantly–possibly indicating by and large the absence of expression regulation that would discriminate gene products based on their essentiality. Finally, the reconstruction process as well as the gene essentiality comparison have yielded a number of informed hypotheses and suggestions for specific experiments that will guide the ongoing experimental investigation of gene functions in the minimal cell.

## Results

While the minimal cell JCVI-syn3A is a new organism with little experimental data yet available, its natural precursor *M. mycoides capri* has been studied in depth, which informed all aspects of the metabolic model. To refer to genes in the JCVI-syn3A genome, we use the locus names of the form MMSYN1_xxxx as used in the annotation of JCVI-syn1.0 (*Hutchison et al., 2016a*) to allow us to discuss genes deleted in JCVI-syn3A more clearly. The MMSYN1_ prefix is omitted for brevity. Understanding the in vivo essentiality of genes in JCVI-syn3A is an important first step to the development of the metabolic reconstruction: this is presented first in Section 'Transposon mutagenesis experiments probe in vivo gene essentiality'. Using the protein expression profiles measured for JCVI-syn3A, the biomass composition of JCVI-syn3A is then derived in Section 'Biomass composition and reaction', as well as the biomass reaction used in the model. The construction and justification of the metabolic model is presented in Section 'Metabolic reconstruction'. The steady-state fluxes obtained from the model are then compared in Section 'Steady state fluxes' to experimental fluxes, as well as to protein abundance-based flux limits. The metabolic energy usage of JCVI-syn3A is analyzed in Section 'Energy usage'. Gene essentiality obtained from in silico gene knockouts is presented in Section 'In silico gene knockouts and mapping to in vivo essentiality'. Finally, we compare protein expression profiles between essential and non-essential proteins as identified in the model or in vivo in Section 'Abundances of essential and non-essential proteins'.

### Transposon mutagenesis experiments probe in vivo gene essentiality

Transposon insertion mutagenesis studies were performed in order to probe the dispensability of individual genes in JCVI-syn3A (see Section 'Materials and methods'). In this experiment, transposons are randomly inserted into the chromosomes of a population of cells that is then plated under selection for a drug resistance gene carried by the transposon (*Hutchison et al., 1999*; *Glass et al., 2006*). After transferring to a liquid culture ('passage zero', $P_0$), four serial passages are performed. DNA from the pooled colonies is isolated and sequenced to determine the location of transposon insertions within the genome at the beginning or at the end of the experiment. When determining transposon locations at the beginning of the experiment, $P_1$ is used over $P_0$ to limit any contamination from the DNA of non-viable cells. The number of insertions observed in a coding region can then be used to infer the importance of that gene. We note that not every insertion will necessarily obliterate a gene's function. A graphical presentation of the essentiality classification along with the distribution of transposon insertions over a portion of the genome is presented in *Figure 2*. It shows that *secA*/0095 is heavily hit with insertions in the 3' 25% of the gene (but practically nowhere else); however, SecA is certainly essential because it is a necessary component of the protein translocase, which inserts proteins such as transporters into the membrane. While the absence of gene products for genes carrying transposon inserts has not been confirmed experimentally, genes with relatively high insertion counts are more likely to be functionally disrupted and thus non-essential. Genes that are not required by the organism to grow in rich media will contain many transposon insertions ('non-essential' genes), whereas genes required for cell viability will be sparsely hit by transposon insertions. To identify genes whose disruption might not be immediately detrimental but might cause a growth defect apparent later on, sequencing of the transposon mutagenesis library was performed on $P_4$ cells as well. Cells with a gene disruption that is not immediately lethal but causes a

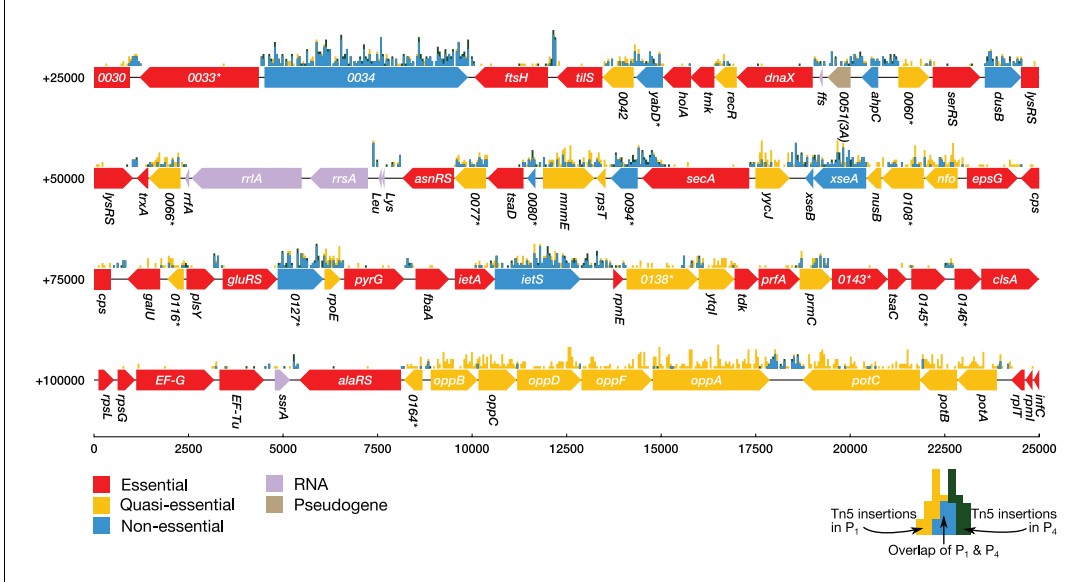

**Figure 2.** Classification of gene essentiality from transposon insertion data using a Poisson mixture model for a representative region of the JCVI-syn3A genome. Coding regions are colored by their predicted class: red (essential), yellow (quasi-essential), blue (non-essential). Lavender regions denote RNA and light brown regions are pseudogenes. The distributions of transposon insertions in passage 1 and passage 4 are represented by yellow and dark green histograms, respectively (bin size of 50 bp). The overlap of the two histograms is highlighted in blue. When a common gene name is not available, the four-digit locus tag for JCVI-syn1.0 is used instead. Locus number identifiers with the (3A) suffix represent newly identified open reading frames in JCVI-syn3A which are missing from the JCVI-syn1.0 annotation. Asterisks mark genes with unknown functionality.

The online version of this article includes the following figure supplement(s) for figure 2:

**Figure supplement 1.** Classification of gene essentiality from transposon insertion data using a Poisson mixture model for 0–275,000 bp.
**Figure supplement 2.** Classification of gene essentiality from transposon insertion data using a Poisson mixture model for 275,000–543,379 bp.
**Figure supplement 3.** Distribution of transposon insertion counts for $P_1$ (panel a) and $P_4$ (panel b) compared to the distribution inferred through the Poisson mixture model.

growth disadvantage will be outgrown after $P_4$, and the number of insertions for that gene will then significantly decrease from $P_1$ to $P_4$. These genes are denoted 'quasi-essential'.

A Poisson mixture model was used to partition the coding regions into two sets of genes based on the transposon insertion density. By comparing the assignment of genes into classes of sparse and dense transposon insertions between $P_1$ and $P_4$, essentiality can be inferred. This classification method considers a gene to be essential if it has been classified to have sparse transposon insertions in both $P_1$ and $P_4$, quasi-essential if it was classified to have dense transposon insertions for $P_1$ and sparse insertions for $P_4$, and non-essential if the gene was classified as densely hit for both $P_1$ and $P_4$. See Section 'Materials and methods' for a complete description of the classification method. *Figure 2—figure supplement 3* shows the fit of the model to the distribution of transposon insertion counts per gene.

In six out of 452 instances, the mixture model failed to classify the gene, either due to low assignment confidence or due to increased insertions from $P_1$ to $P_4$. The short ribosomal proteins S9 (*rpsI/*0637), L27 (*rpmA/*0499), and L31 (*rpmE/*0137) were manually assigned as essential since they are necessary to construct a functional ribosome. The gene *secA/*0095 could not be automatically classified since the mixture model predicted the gene to be more heavily hit with insertions in $P_4$ than in $P_1$; it was assigned as essential as it is a major component of the translocase assembly. The insertions occurred in the C-terminus linker domain considered to be important for binding to phospholipids and preprotein translocation. A gene of unclear function (0235) was predicted by the model to be essential at a slightly higher probability than quasi-essential (0.471 vs. 0.416, respectively); however, it was manually assigned to be quasi-essential following its previous assignment in JCVI-syn2.0 (*Hutchison et al., 2016a*). Thioredoxin (*trxA/*0065) was assumed to be essential since its associated reductase (*trx/*0819) was predicted to be essential by the mixture model. Only one gene was misclassified: the ATP synthase subunit ε (*atpC/*0789), initially classified as non-essential, was manually

reassigned to essential since all other ATPase subunits (*atpD*/0790 through *atpB*/0796) were essential according to the mixture model. The majority of transposon insertions in *atpC*/0789 are found in the $3'$ region, similar to the pattern seen in *secA*. However, it is possible that the $\epsilon$ subunit may not actually be essential since in *M. pneumoniae*, transposon insertions into the *atpC* (MPN597) lead to viable cells with decreased cytadherence activity (*Shimizu et al., 2014*).

The set of genes classified quasi-essential could potentially include essential genes which cannot be discriminated using these transposon insertion data. For these misclassified genes, it is possible that although expression of the gene product essential for cell growth has been halted, previously translated essential proteins remain in the cell in sufficient quantities to maintain cell growth and division up to $P_1$. A further discussion of this argument is presented in Section 'Completeness of the model' and Appendix 1.

The genes identified as non-essential by the Poisson mixture model may contain 'weakly' quasi-essential genes, that is disrupted genes which confer a minor growth disadvantage. This behavior would manifest as a decrease in transposon insertions between $P_1$ and $P_4$, but not such a steep decline that the gene is observed with little to no insertions. To identify these 'weakly' quasi-essential genes, the genes classified as non-essential were subjected to further classification using $k$-means clustering of the ratio of the number of transposon insertions in $P_4$ to $P_1$ assuming two clusters (see *Figure 2—figure supplement 3*). Of the 118 genes initially classified non-essential, 42 were reclassified as quasi-essential.

The assignment of essentiality classes and distribution of transposon insertions over the entire genome are presented in *Figure 2—figure supplements 1,2*, and *Supplementary file 3*. Genomic positions of transposon insertions are listed in *Supplementary file 2*. *Figure 3* summarizes the breakdown of the essential, quasi-essential, and non-essential genes according to the functional classes. Of the 452 coding genes in JCVI-syn3A, 60% are essential, 25% are quasi-essential, and 15% are non-essential by this analysis. The detailed breakdown of the JCVI-syn3A genome into these classes (*Table 1*) shows that of the 91 genes of unclear function, 30 are essential, 32 are quasi-essential, and 29 are non-essential. Those 30 essential genes could represent new biological mechanisms not yet defined and should motivate the search to discover their function (*Alberts, 2011*).

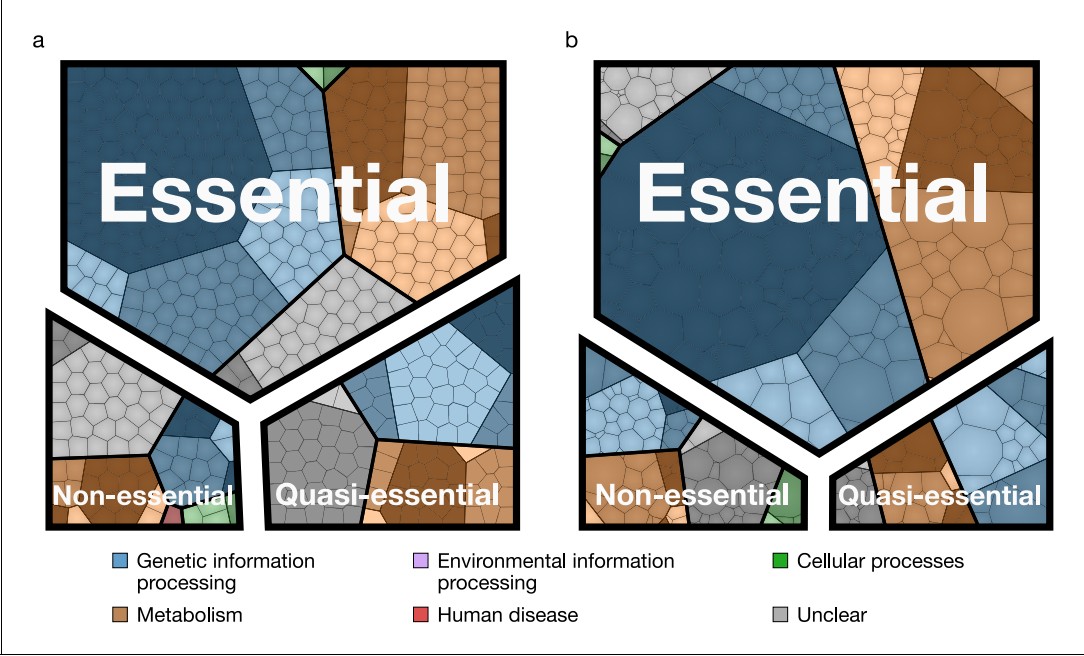

**Figure 3.** Essential, quasi-essential, and non-essential protein coding genes in JCVI-syn3A across four functional classes. (a) Distribution across genome (cell areas all equal). (b) Distribution across proteome (cell areas proportional to protein copy number in an average cell). Among non-essential proteins, the three most abundant ones are *ftsZ*/0522, the peptidase 0305 and 0538 (unclear function). A detailed breakdown of the JCVI-syn3A genome into these classes is available in *Table 1*.

Since on average only one transposon insertion occurs per cell and the identification of insertion locations within the genome is performed over an ensemble of cells, these transposon mutagenesis studies can only reveal *individual* gene essentialities. To probe the essentiality of groups of genes, one would need to perform targeted multiple knockout studies. However, for metabolic genes, flux balance analysis of the metabolic reconstruction can predict the essentiality of groups of genes. In Targeted gene removal experiments, the individual gene essentiality results are expanded to include the assignment of essentiality to combinations of genes in silico, leading to potential combinations of genes to remove to further minimize the genome. The classifications of the genes used in the metabolic reconstruction are shown in *Table 2*.

Preliminary triple knockout experiments involving various sets of non-essential genes lead to cells with greatly impaired growth rates (data not shown). The fact that ~15% of the genes in JCVI-syn3A are individually non-essential is not inconsistent with the near-minimality of the genome as a whole: it is not possible to remove all non-essential genes without vastly decreasing the growth rate or outright killing the cell. Furthermore, a genome comprised only of essential and quasi-essential genes is non-viable as well, since the removal of a non-essential gene can cause a previously quasi-essential gene to become non-essential in the new construct. As JCVI-syn3A grows more slowly than JCVI-syn1.0 (2 hr doubling time vs. 1 hr), a gene disruption that in JCVI-syn1.0 led to outgrowth by unaffected competitor cells might still survive through passage four in JCVI-syn3A. As a result, genes that were classified as quasi-essential in JCVI-syn1.0 can appear non-essential in JCVI-syn3A, and could in principle be removed as well—for the price of some gradual further decrease in growth rate. This lack of a clear cutoff again underscores the 'trade-off between genome size and growth rate' taking place during genome minimization (*Hutchison et al., 2016a*).

## Biomass composition and reaction

The cellular components of JCVI-syn3A fall into three categories: macromolecules, lipids and capsule, and small molecules and ions. *Appendix 1—table 1* lists the mass fractions for all components included in the JCVI-syn3A biomass composition. These mass fractions are used to derive the coefficients in the biomass reaction depicted in *Figure 4* for each component based on its molecular weight. The different biomass components are summarized below with the full discussion and derivation in Appendix 1. The growth-associated maintenance (GAM) ATP cost shown in *Figure 4* is described in Section 'GAM/NGAM'.

### Macromolecules

The macromolecular mass fractions are based on the experimental composition of *M. mycoides capri* (*Razin et al., 1963*), which is assumed to provide a very good approximation for JCVI-syn3A (which was derived from an *M. mycoides capri* substrain). The DNA fraction is slightly increased from the reported 5% to 5.5%, which corresponds to exactly one chromosome in JCVI-syn3A. Assuming almost all RNA to be present as ribosomal RNA (rRNA) and around 4600 bases per ribosome yields an upper limit of ~670 ribosomes per average cell. This number would correspond to ~20,000 ribosomes in *E. coli* when scaled by cell volume, which is within the observed growth dependent range of 8000–73,000 for *E. coli* (*Bremer and Dennis, 2008*). The RNA base composition is based upon the close relative *M. mycoides capri* serovar LC (*M. mycoides capri* LC) strain Y (*Mitchell and Finch, 1977*). (*M. mycoides capri* LC Y is referred to as '*M. mycoides mycoides* goat strain Y' in the older literature, but has recently been included in subspecies *capri* (*Manso-Silván et al., 2009*). The DNA composition is determined by the GC content of the genome (24%). The absolute number of proteins can be estimated from the average protein molecular weight in JCVI-syn3A, which is obtained from the proteomics studies reported in Section 'Abundances of essential and non-essential proteins' (see also Section 'Mass Spectrometry Based Proteomics' within Section 'Methods and methods' for experimental details). For an average cell, this amounts to ~77,000 proteins; the resulting protein volume density in a 400 nm spherical cell is $2.3 \times 10^6$ proteins/µm$^3$, which compares well to the estimated density of $3.5 - 4.4 \times 10^6$ proteins/µm$^3$ in *E. coli* (*Milo, 2013*). The protein amino acid composition is computed directly from the proteomics data.

In addition to a generic protein species (describing the average JCVI-syn3A protein), two specific proteins are included: acyl carrier protein (ACP, *acpA*/0621) and dUTPase (*dut*/0447). ACP carries a 4'-phosphopantetheine moiety in its holo form, and including the holoenzyme in the biomass

**Table 2.** Genes modeled in the metabolic reconstruction.

The 'MMSYN1_' prefix on the locus tags has been omitted for brevity. The reaction column provides the specific reaction name or general description of the gene (if involved in multiple reactions). Reaction names may appear multiple times if there are multiple gene products that can catalyze that reaction. Column $Ess_{Tn5}$ contains a · if the gene is non-essential, a □ if it is quasi-essential, or a ■ if it has been determined to be essential through the transposon mutagenesis experiments. A dagger in this column indicates that the automatic essentiality assignment required manual intervention. Column $Ess_{FBA}$ contains a · if the gene is non-essential or a ■ if it has been determined to be essential through FBA. Loci marked with an asterisk are genes that are non-essential only 'technically' with respect to FBA (see Section 'In silico gene knockouts and mapping to in vivo essentiality'). The doubling times predicted by FBA for non-essential genes were all 2.02 hr, with the exception of single knockouts of loci *pdhC/*0227 through *ackA/*0230, which all had doubling times of 3.22 hr; locus *punA/*0747 with a doubling time of 2.04 hr; and locus *gltP/*0886 with a doubling time of 2.03 hr.

| Locus | Reaction | $Ess_{Tn5}$ | $Ess_{FBA}$ | Locus | Reaction | $Ess_{Tn5}$ | $Ess_{FBA}$ | Locus | Reaction | $Ess_{Tn5}$ | $Ess_{FBA}$ |
|---|---|---|---|---|---|---|---|---|---|---|---|
| **Amino acid metabolism** | | | | **Cofactor metabolism** | | | | 0798 | UPPRT | ■ | ■ |
| 0381* | AHCi | · | · | 0823 | 5FTHFPGS | □ | ■ | 0330 | dAdn kinase 1 | · | · |
| 0163 | ALATRS | ■ | ■ | 0390 | FMETTRS | □ | ■ | 0382 | dAdn kinase 2 | · | · |
| 0535 | ARGTRS | ■ | ■ | 0291 | FMNAT | ■ | ■ | **Transport** | | | |
| 0076 | ASNTRS | ■ | ■ | 0443 | FTHFCL | ■ | ■ | 0822 | 5FTHFabc | □ | ■ |
| 0287 | ASPTRS | ■ | ■ | 0799 | GHMT | □ | ■ | 0876 | AA permease 1 | · | · |
| 0837 | CYSTRS | ■ | ■ | 0684 | MTHFC | □ | ■ | 0878 | AA permease 2 | □ | · |
| 0687 | GLNTRAT | ■ | ■ | 0259 | NADK | ■ | ■ | 0789 | ATPase | ■[†] | ■ |
| 0688 | GLNTRAT | ■ | ■ | 0378 | NADS | ■ | ■ | 0790 | ATPase | ■ | ■ |
| 0689 | GLNTRAT | ■ | ■ | 0614 | NCTPPRT | ■ | ■ | 0791 | ATPase | ■ | ■ |
| 0126 | GLUTRS_Gln | ■ | ■ | 0380 | NNATr | ■ | ■ | 0792 | ATPase | ■ | ■ |
| 0405 | GLYTRS | ■ | ■ | **Lipid metabolism** | | | | 0793 | ATPase | ■ | ■ |
| 0288 | HISTRS | ■ | ■ | 0621 | ACP | ■ | ■ | 0794 | ATPase | ■ | ■ |
| 0519 | ILETRS | ■ | ■ | 0419 | ACPPAT | ■ | ■ | 0795 | ATPase | ■ | ■ |
| 0634 | LEUTRS | ■ | ■ | 0513 | ACPS | ■ | ■ | 0796 | ATPase | ■ | ■ |
| 0064 | LYSTRS | ■ | ■ | 0512 | AGPAT | ■ | ■ | 0879 | CA2abc | ■ | ■ |
| 0432* | MAT | ■ | · | 0117 | APG3PAT | ■ | ■ | 0836 | COAabc | ■ | ■ |
| 0012 | METTRS | ■ | ■ | 0139 | BPNT | □ | ■ | 0642 | EcfA | ■ | ■ |
| 0528 | PHETRS | ■ | ■ | 0147 | CLPNS | ■ | ■ | 0643 | EcfA | ■ | ■ |
| 0529 | PHETRS | ■ | ■ | 0697 | DAGGALT | □ | · | 0641 | EcfT | ■ | ■ |
| 0282 | PROTRS | ■ | ■ | 0114 | DAGPST/DAGGALT | ■ | ■ | 0233 | GLCpts | ■ | ■ |
| 0133 | Peptidase 1 | · | · | 0304 | DASYN | ■ | ■ | 0234 | GLCpts | ■ | ■ |
| 0305 | Peptidase 2 | □ | · | 0420 | FAKr | ■ | ■ | 0694 | GLCpts | ■ | ■ |
| 0444 | Peptidase 3 | □ | · | 0616 | FAKr | ■ | ■ | 0779 | GLCpts | ■ | ■ |
| 0479 | Peptidase 4 | □ | · | 0617 | FAKr | □ | ■ | 0886 | GltP | □ | · |
| 0061 | SERTRS | ■ | ■ | 0115 | GALU | ■ | ■ | 0685 | Kt6 | ■ | ■ |
| 0222 | THRTRS | ■ | ■ | 0218 | GLYK | ■ | ■ | 0686 | Kt6 | ■ | ■ |
| 0308 | TRPTRS | ■ | ■ | 0733 | PGMT/PPM | ■ | ■ | 0401 | LIPTA | · | · |
| 0613 | TYRTRS | ■ | ■ | 0214 | PGPP | □ | ■ | 0787 | MG2abc | ■ | ■ |
| 0260 | VALTRS | ■ | ■ | 0875 | PGSA | ■ | ■ | 0314 | NACabc | ■ | ■ |
| **Central metabolism** | | | | 0113 | PSSYN | ■ | ■ | 0165 | Opp | □ | · |
| 0230 | ACKr | ■ | · | 0813 | UDPG4E | ■ | ■ | 0166 | Opp | □ | · |
| 0493 | AGDC | □ | · | 0814 | UDPGALM | ■ | ■ | 0167 | Opp | □ | · |
| 0495 | AMANK | · | · | **Macromolecules** | | | | 0168 | Opp | □ | · |
| 0494 | AMANPEr | · | · | 0394 | Lon | □ | ■ | 0169 | Opp | □ | · |
| 0732 | DRPA | · | · | 0650 | Met peptidase | ■ | ■ | 0345 | P5Pabc | ■ | ■ |

*Table 2 continued on next page*

*Table 2 continued*

| Locus | Reaction | Ess$_{Tn5}$ | Ess$_{FBA}$ | Locus | Reaction | Ess$_{Tn5}$ | Ess$_{FBA}$ | Locus | Reaction | Ess$_{Tn5}$ | Ess$_{FBA}$ |
|-------|----------|-------------|-------------|-------|----------|-------------|-------------|-------|----------|-------------|-------------|
| 0213 | ENO | ■ | ■ | 0201 | Pept. deformylase | ■ | ■ | 0425 | Plabc | ■ | ■ |
| 0131 | FBA | ■ | ■ | | Nucleotide metabolism | | | 0426 | Plabc | ■ | ■ |
| 0726 | G6PDA | · | · | 0651 | (D)ADK | ■ | ■ | 0427 | Plabc | ■ | ■ |
| 0607 | GAPD | ■ | ■ | 0413 | ADPT | ■ | ■ | 0877 | RIBFLVabc | ■ | ■ |
| 0451 | GAPDP | ■ | · | 0129 | CTPS | ■ | ■ | 0008 | RNS | ■ | ■ |
| 0475 | LDH_L | ■ | · | 0347 | CYTK | □ | ■ | 0009 | RNS | ■ | ■ |
| 0435 | MAN6PI | · | · | 0515 | DCMPDA | □ | · | 0010 | RNS | ■ | ■ |
| 0227 | PDH_E2/_acald | · | · | 0447 | DUTPDP | ■ | ■ | 0011 | RNS | ■ | ■ |
| 0228 | PDH_E3 | · | · | 0203 | GK | ■ | ■ | 0195 | SPRMabc | □ | ■ |
| 0220 | PFK | ■ | ■ | 0216 | GUAPRT | □ | ■ | 0196 | SPRMabc | □ | ■ |
| 0445 | PGI | ■ | ■ | 0747 | PNP | □ | · | 0197 | SPRMabc | □ | ■ |
| 0606 | PGK | ■ | ■ | 0344 | PPA | ■ | ■ | 0706 | THMPPabc | □ | ■ |
| 0729 | PGM | ■ | ■ | 0771 | RNDR | □ | ■ | 0707 | THMPPabc | □ | ■ |
| 0831 | PRPPS | ■ | ■ | 0772 | RNDR | ■ | ■ | 0708 | THMPPabc | □ | ■ |
| 0229 | PTAr | ■ | · | 0773 | RNDR | □ | ■ | | | | |
| 0221 | PYK | ■ | ■ | 0140 | TMDK1/DURIK1 | ■ | ■ | | | | |
| 0262 | RPE | □ | ■ | 0045 | TMPK | ■ | ■ | | | | |
| 0800 | RPI | □ | ■ | 0065 | TRDR | ■ | ■ | | | | |
| 0316 | TKT | □ | ■ | 0819 | TRDR | ■ | ■ | | | | |
| 0727 | TPI | ■ | ■ | 0537 | UMPK | ■ | ■ | | | | |

equation enforces flux through the corresponding prosthetic group attachment reaction. dUTPase is included for technical reasons discussed in Section 'Nucleotide metabolism'. We use cellular abundances of 138 (ACP) and 10 (dUTPase), derived from the proteomics experiments. The resulting mass fractions are then subtracted from the total protein mass fraction.

## Lipids and capsule

Based on the experimental lipid composition of *M. mycoides capri* serovar capri PG3 (*Archer, 1975*) and *M. mycoides capri* LC Y (*Plackett, 1967a*), the model includes the phospholipids phosphatidyl-glycerol and cardiolipin, the glycolipid monogalactosyl-diacylglycerol (Gal-DAG), cholesterol, diacyl-glycerol and free fatty acids. For fatty acids, palmitic acid (C16:0) and oleic acid (C18:1 cis-9) are considered to be the two most important representatives, and an average 'fatty acid' with a molecular weight averaged between palmitate and oleate is used in all lipid species. In addition, a galactan polysaccharide capsule is included in the biomass. *M. mycoides capri* LC GM12 (the strain from which JCVI-syn3A is derived) is known to produce a galactan polysaccharide (specifically, poly-$\beta-1 \rightarrow 6$-galactofuranose) (*Schieck et al., 2016*); while it is not yet experimentally known whether the minimal cell still produces this galactan, genetic features suggest it does. As other *M. mycoides capri* LC strains form a polysaccharide capsule but secrete negligible amounts of polysaccharide (*Bertin et al., 2015*), the JCVI-syn3A galactan is assumed to form a capsule as well and is included as polygalactosyl-diacylglycerol (lipogalactan).

## Small molecules and ions

In addition to the macromolecules and lipids, we also include pools of free amino acids, nucleotides and deoxynucleotides in our biomass, as well as cofactors and ions expected to be needed in JCVI-syn3A. A minimal medium for JCVI-syn3A has yet to be obtained, so we use the minimal media reported for *M. mycoides capri* LC Y (*Rodwell, 1969*) and *M. pneumoniae* (*Yus et al., 2009*) as a guideline for required ions and cofactors: Any compound present in a minimal medium is required by the cell, and the compound or its downstream product(s) need to be included in the biomass

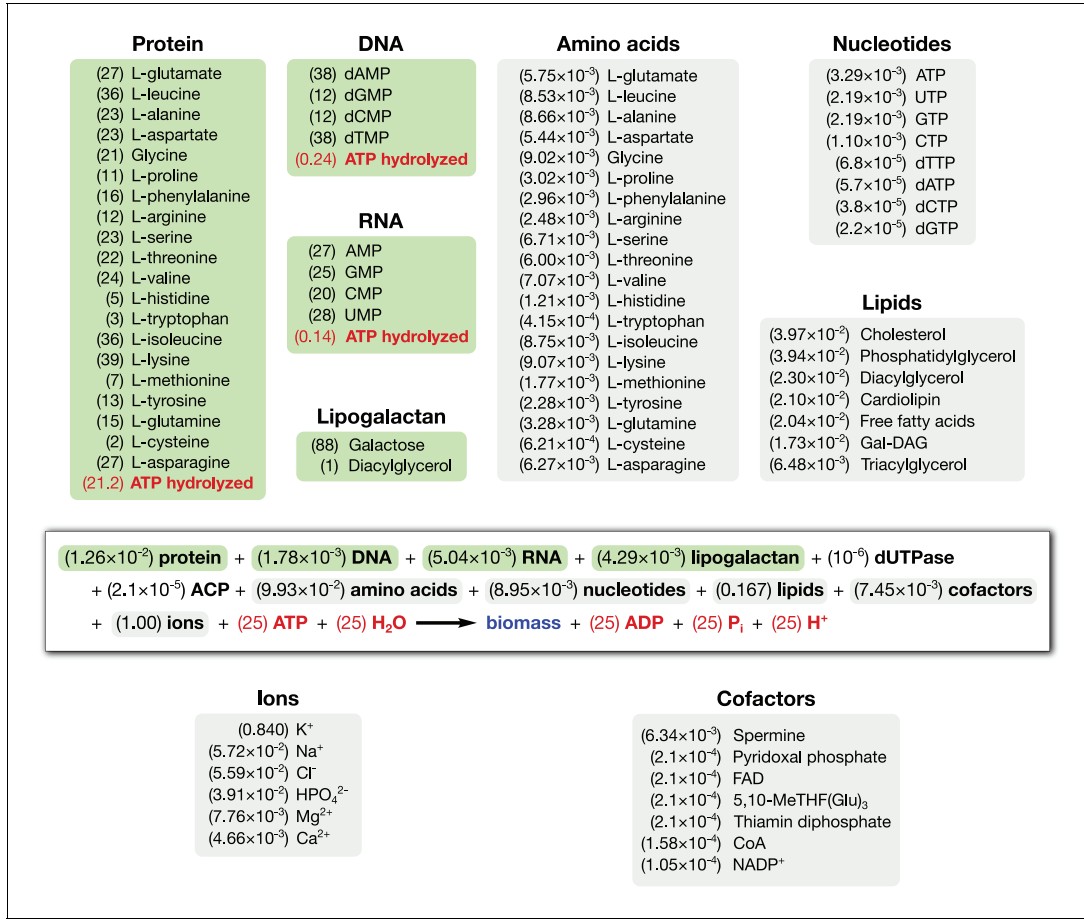

**Figure 4.** Biomass reaction equation for JCVI-syn3A. This reaction consumes biomass precursors (macromolecules, lipids, capsule, small molecules) (black) and consumes energy in the form of ATP (red) to produce biomass (blue). Values in parentheses are the stoichiometric coefficients in mmol compound per gram cellular dry weight (mmol gDW$^{-1}$). The macromolecular compositions are highlighted in green (stoichiometric coefficients within the macromolecule, unitless) and the compositions of lipids and small molecule pools are highlighted in gray (mmol gDW$^{-1}$). ATP expenses within green boxes denote total macromolecular synthesis costs (based on macromolecular fractions in the biomass) and the ATP expense in the main equation denotes the nonquantifiable part of the growth-associated maintenance cost (GAM; see Section 'GAM/NGAM').

composition. From the two media mentioned, all inorganic ions are included in the JCVI-syn3A biomass composition except for sulfate, for which there is no known need in JCVI-syn3A. The vitamin choline from the *M. pneumoniae* minimal medium is also excluded as *M. mycoides capri* does not synthesize its own phosphatidylcholine (*Plackett, 1967a*). Transition metal ions are also not included. While these media are used to determine which compounds to include in the biomass composition, their intracellular concentrations/mass fractions are obtained from measurements in *M. mycoides capri* (*Leblanc and Le Grimellec, 1979*; *Mitchell and Finch, 1979*; *Neale et al., 1983a*) or other mycoplasmas (*Linker and Wilson, 1985*) or taken from the iJW145 *M. pneumoniae* (*Wodke et al., 2013*) and iJO1366 *E. coli* models (*Orth et al., 2011*).

## Metabolic reconstruction

The metabolic reconstruction of JCVI-syn3A features 338 reactions involving 304 metabolites, catalyzed by gene products of 155 genes, thus covering a third of the genome. The scope of the reconstruction includes all reactions associated with providing the components of the reconstructed biomass (see *Figure 4*). Not covered are metabolic functions outside the 'core' functions, in particular RNA modifications and damage repair reactions. While many RNA modification enzymes are known already, the prevalence of specific RNA modifications in the RNA pool is not yet known. A few RNA modification enzymes are however discussed with regard to folate metabolism in Section 'The role of folate metabolism'. The majority of damage reactions and possible repair

thereof are mostly not yet known, and are hence omitted save for two genes in cofactor and nucleotide metabolism. Approximately 30 genes pertaining to RNA modification are listed in our KEGG ortholog search as 'Genetic Information Processing' and will be included in a future model for ribosome biogenesis and tRNA biogenesis.

The model reactions are organized in nine subsystems, which are listed in *Supplementary file 1B* together with their respective number of reactions and genes included. Among these subsystems, 'Biomass production' contains the biomass reaction discussed in Section 'Biomass composition and reaction'. 'Exchange' contains the model reactions that describe metabolite exchange with the media. All other subsystems are discussed in detail in the following subsections. *Figure 5* shows a global view of the metabolic network (excluding biomass and exchange reactions). Individual maps for the subsystems for central, nucleotide, cofactor, lipid, macromolecule and amino acid metabolism are depicted in *Figures 6–7* and Figures 9–12. *Table 2* lists all genes included in the reconstruction, together with their in vivo and in silico essentiality.

Experimentally, JCVI-syn3A is grown in the rich and not fully defined SP4 media (*Glass et al., 2015*; *Tully et al., 1979*), since a defined media supporting its normal growth has yet to be obtained. Consequently, a rich in silico medium that provides for all biomass precursors the cell can take up is assumed, with glucose as the only energy source.

## Central metabolism

A schematic diagram of central metabolism in JCVI-syn3A is provided in *Figure 6*. The only annotated sugar importer in the JCVI-syn3A genome is the glucose PTS system comprising PtsI (*ptsI*/0233), PtsH (*ptsH*/0694), Crr (glucose-specific IIA component, *crr*/0234) and PtsG (*ptsG*/0779). The phosphate-transfer reaction chain from phosphoenolpyruvate (PEP) to glucose (*Postma et al., 1993*) is lumped together into a single reaction that imports glucose and phosphorylates it with PEP. The presence of ManA (mannosamine-6-phosphate isomerase, *manA*/0435) and NagB (glucosamine 6-phosphate deaminase, *nagB*/0726) suggests possible utilization of mannose and glucosamine as well, which is supported by mutation studies suggesting that in *M. mycoides mycoides* strain T1, PtsG is involved in the uptake of all three sugars (*Lee et al., 1986*). Thus PTS-mediated combined uptake and phosphorylation is included for glucose, glucosamine and mannose. Furthermore, the presence of a putative *nagA* operon (0493 through 0495) suggests possible uptake of N-acetylmannosamine or N-acetylneuraminate (sialic acid). Therefore, an uptake reaction for N-acetylmannosamine is included, noting that this is probably not imported through the glucose PTS system, as concomitant phosphorylation would render the putative N-acetylmannosamine kinase NagC (0495) redundant. Lacking more information, we assume that this import reaction is mediated by an ATP-binding cassette (ABC) transporter instead.

As uptake and secretion measurements for JCVI-syn3A are not yet available, similar uptake rates to those measured in *M. pneumoniae* (*Wodke et al., 2013*) are used, as has been done in previous mycoplasma models (*Ferrarini et al., 2016*). Since glucose is considered to be the only energy source, upper limits of 0.0 mmol gDW$^{-1}$ h$^{-1}$ on mannose, glucosamine and N-acetylmannosamine uptake are applied. A maximum glucose uptake rate of 7.4 mmol gDW$^{-1}$ h$^{-1}$ measured in mid-exponential phase *M. pneumoniae* is used. We note that since mannose and glucosamine would compete for the same PTS importer with glucose, their uptake would not increase the overall sugar uptake.

Four possible sugar sources are assumed to feed into glycolysis via fructose-6-phosphate (F6P): glucose, mannose, glucosamine and N-acetylmannosamine. Only the putative N-acetylmannosamine-6-phosphate epimerase NanE (*nanE*/0494) has been annotated; however, it seems likely that this gene forms part of an operon together with its two adjacent genes. The RAST annotation pipeline (*Overbeek et al., 2014*) suggests the putative ROK family gene 0495 codes for N-acetylmannosamine kinase (NagC). Its genomic context suggests 0493, annotated as a putative dipeptidase, codes for N-acetylglucosamine-6-phosphate deacetylase (NagA). This would complete the N-acetylmannosamine utilization pathway and would be consistent with the putative amide cleavage functionality.

Starting from fructose-6-phosphate, the annotation of JCVI-syn3A contains a complete glycolytic pathway as well as the non-oxidative branch of the pentose phosphate pathway up to the nucleotide precursor phosphoribosylpyrophosphate (PRPP), in line with experimental studies on *M. mycoides capri* LC Y (*Cocks et al., 1985*; *Mitchell, 1976*). As in other mycoplasmas (*Pollack et al., 1997*), the

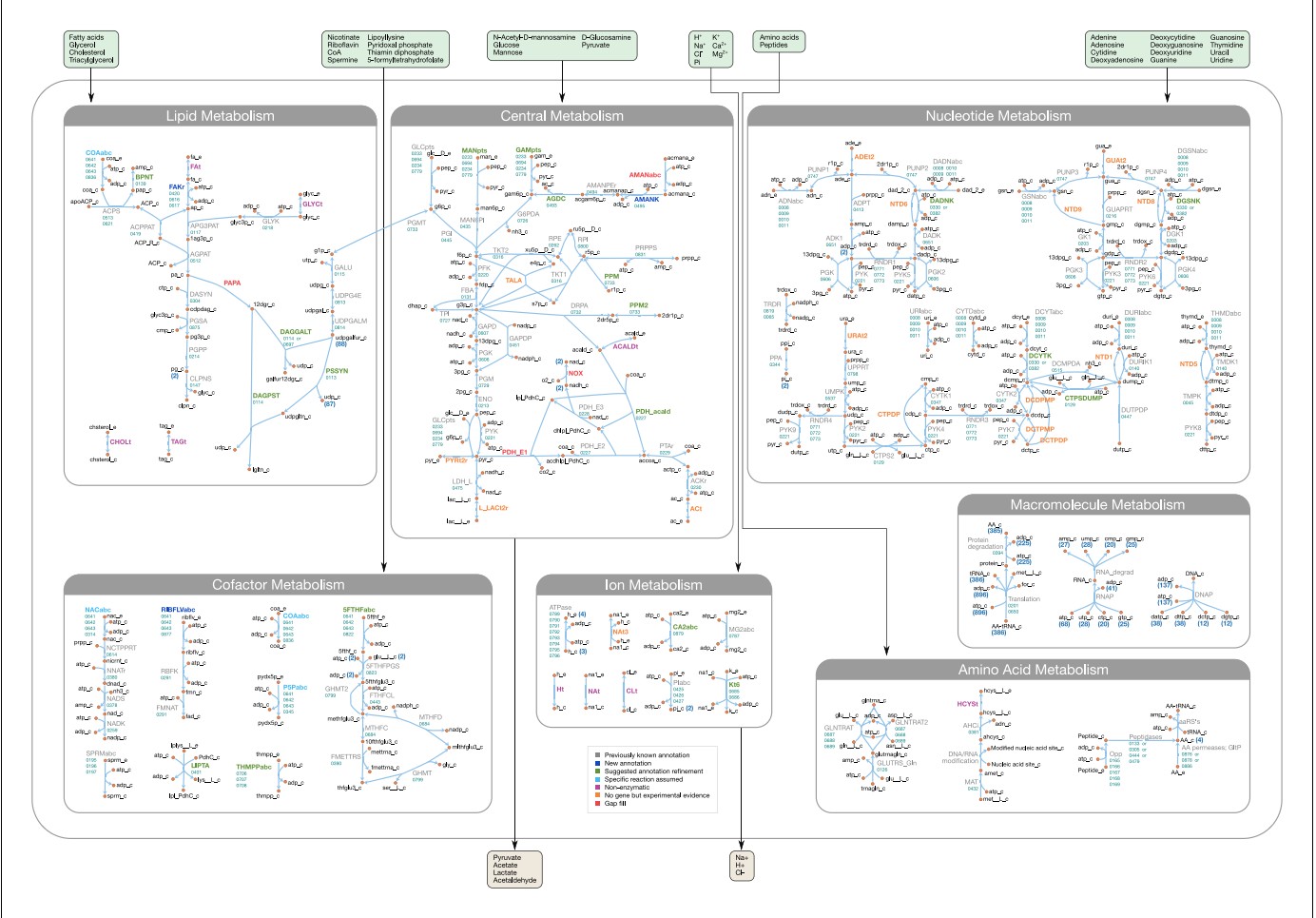

**Figure 5.** Overview of the metabolic reconstruction of JCVI-syn3A, drawn with Escher (*King et al., 2015*). Orange nodes represent metabolites, labeled by their short names in the model (black); the suffixes '_c' and '_e' denote cytoplasmic and extracellular compartments, respectively. For clarity, H₂O, H⁺, PPᵢ and Pᵢ are generally omitted as reactants. Blue edges represent (enzymatic or spontaneous) reactions, labeled by reaction name (gray labels) and associated gene loci (gene-protein-reaction (GPR) rules, turquoise; omitting 'MMSYN1_' prefix). Blue parenthesized numbers denote reactants (products) which are consumed (produced) in stoichiometric quantities greater than one. In this map and subsequent maps, the following color scheme for highlighted reactions is used—blue: reaction based on new annotation, light green: reaction based on suggested annotation refinement, cyan: specific reaction assumed for generic annotation, light violet: non-enzymatic reaction, orange: reaction not accounted for by gene yet but supported by experimental evidence, and red: reaction included based on gap filling. Small boxes list metabolites that can be taken up (green boxes) or secreted (brown boxes) under physiological conditions.

gene for transaldolase (TALA) has not been identified; however, transaldolase activity has been detected experimentally (*Desantis et al., 1989*). In particular, it has been detected in *M. mycoides capri* LC Y cell extracts (*Cocks et al., 1985*), albeit the detected specific activity was rather weak (13 µmol min⁻¹ g cell protein⁻¹) and the level of background noise in that study is not known. Thus, this reaction is included in the model but we note that the evidence seems ambiguous.

While the completion of the glycolytic pathway via lactate dehydrogenase (LDH, *ldh*/0475) is possible, several genes from the acetate fermentation branch have been deleted, namely the E1 subunit of pyruvate dehydrogenase (PDH_E1, MMSYN1_0225 and MMSYN1_0226) and NADH oxidase (NOX, MMSYN1_0223). However, the remaining subunits of the PDH complex (PDH_E2 and PDH_E3, *pdhC*/0227 and *pdhD*/0228), as well as the path from acetyl-CoA to acetate, are still present in the genome. NAD regeneration could possibly be carried out by one of the remaining oxidoreductases of unclear functionality. Until further information becomes available, the remaining PDH complex is assumed to act on pyruvate to yield acetyl-CoA and finally acetate.

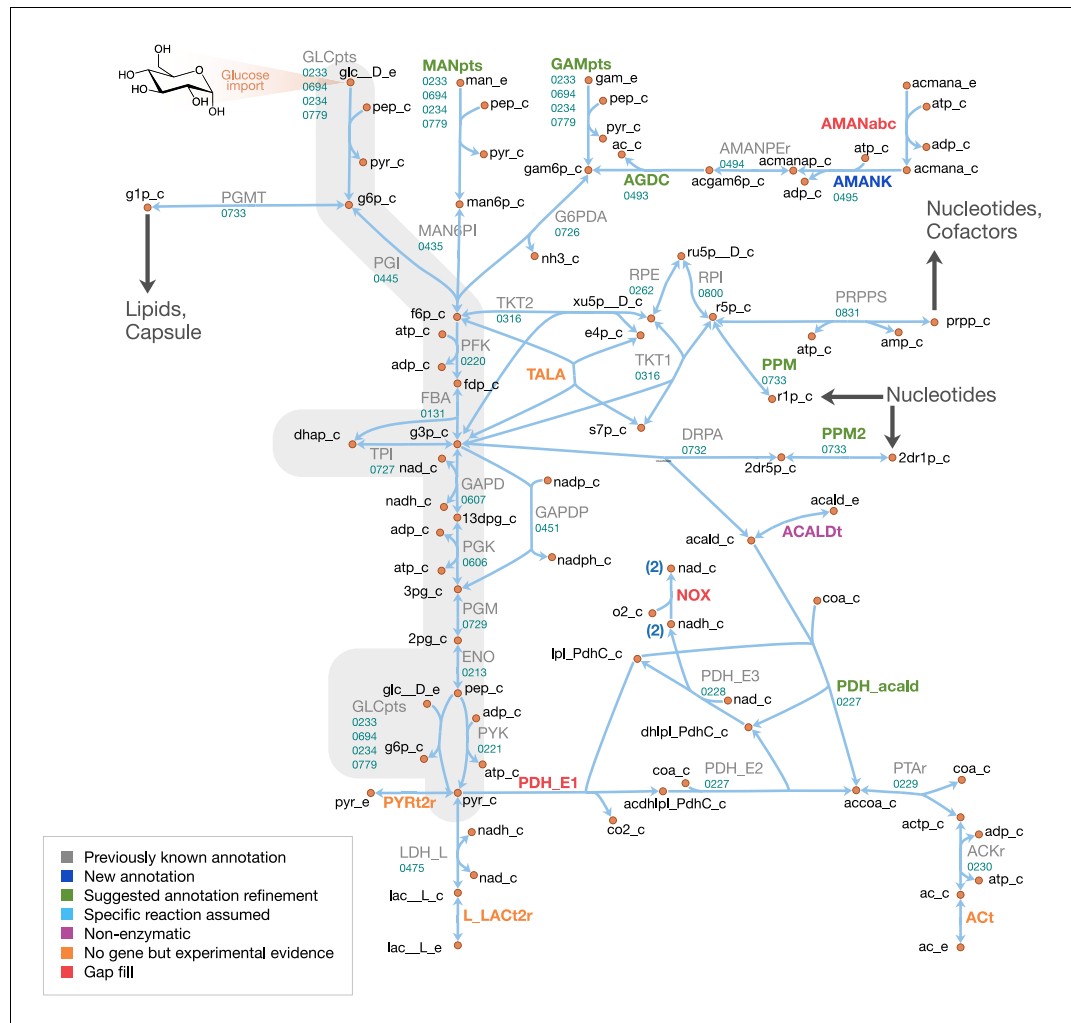

**Figure 6.** Central metabolism in JCVI-syn3A. Map components and labels as in *Figure 5*. Big arrows denote incoming or outgoing connections to other parts of the metabolic network. For context, the node representing glucose transport has been labeled explicitly and glycolysis has been highlighted in gray.

The online version of this article includes the following figure supplement(s) for figure 6:

**Figure supplement 1.** Steady-state fluxes through central metabolism in JCVI-syn3A.

Another possible function for the remaining PDH complex would be oxidation of acetaldehyde to acetyl-CoA, which would not require a decarboxylation in the absence of the PDH_E1 subunit (see Appendix 1). Phosphopentomutase (PPM) and deoxyribose phosphate aldolase (DRPA) activity have been experimentally observed in *M. mycoides capri* LC Y (*Cocks et al., 1985*), enabling the breakdown of deoxyribose 1-phosphate (dR1P) into glyceraldehyde 3-phosphate (G3P) and acetaldehyde (acald). A gene for deoxyribose phosphate aldolase has been annotated in JCVI-syn3A (*deoC*/0732). A strong candidate for phosphopentomutase activity is the putative phosphoglucomutase (PGMT) gene *deoB*/0733. It is preceded by the deoxyribose phosphate aldolase gene (*deoC*/0732) and (in the original JCVI-syn1.0 genome) succeeded by the pyrimidine nucleoside phosphorylase MMSYN1_0734, that is it is neighboring two genes responsible for dR1P production and dR5P breakdown, respectively. At the same time, it shows some similarity (21% sequence identity) to the phosphopentomutase TK1777 from the archaeon *Thermococcus kodakaraensis*. TK1777 showed activity mainly against d1RP, but also weaker activity against glucose 1-phosphate (G1P) (*Rashid et al., 2004*). Thus, *deoB*/0733 is assumed to be the gene responsible for both activities.

Secretion of acetate (*Rodwell and Rodwell, 1954*), lactate (*Rodwell, 1967*) and pyruvate (*Rodwell, 1969*) has been observed in *M. mycoides*; with mutational data on *M. mycoides mycoides* T1

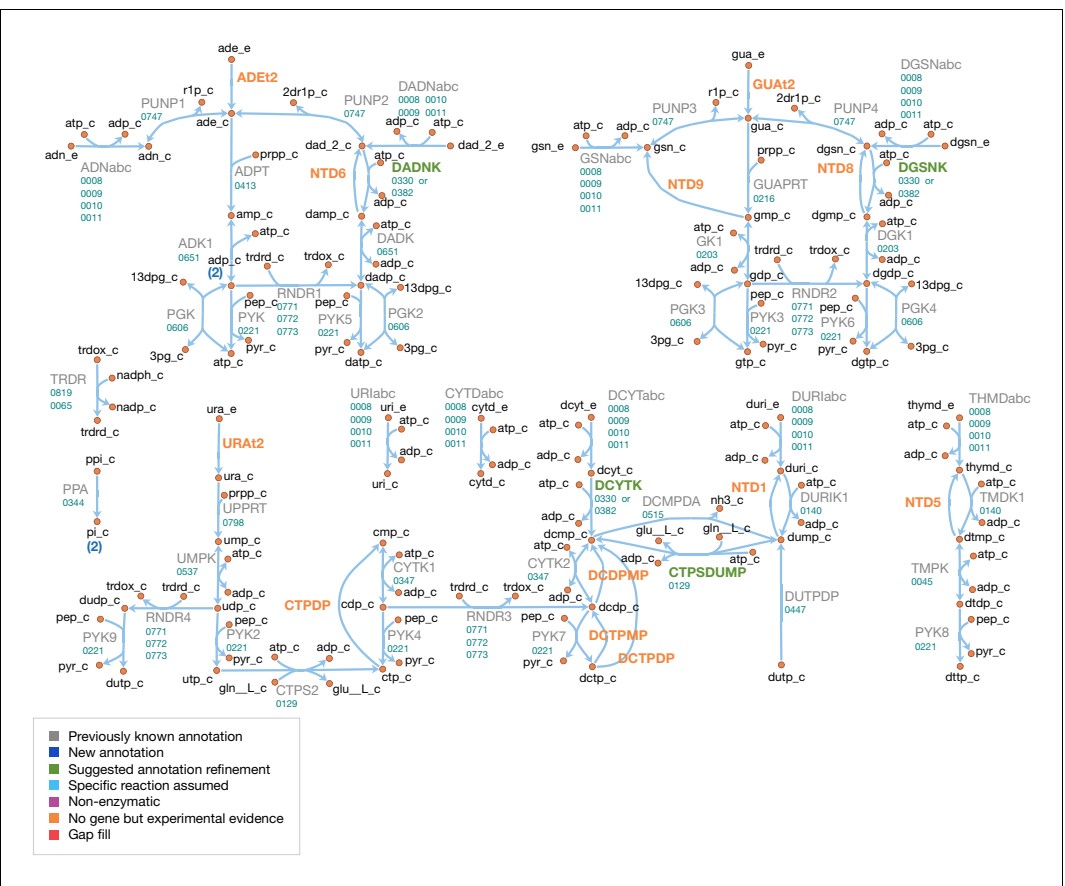

**Figure 7.** Nucleotide metabolism in JCVI-syn3A. Map components and labels as in **Figure 5**.

(**Lee et al., 1986**) indicating a common transporter for pyruvate and lactate. While it is not clear how mycoplasmas secrete acetate, lactate and pyruvate, proton symporters have been suggested for lactate and acetate (**Wodke et al., 2013**) and such reactions are assumed in other mycoplasma models (**Suthers et al., 2009**; **Karr et al., 2012**; **Ferrarini et al., 2016**). The genome of JCVI-syn3A contains several annotated efflux proteins, but all these show features of ATP-coupled transporters, suggesting they are not involved in lactate or acetate export. Thus, lactate, pyruvate and acetate secretion reactions are included as proton symports, noting that for the purposes of this model, this is equivalent to assuming secretion of neutral acid species. The acetate secretion rate is constrained to a maximum of 6.9 mmol gDW$^{-1}$ h$^{-1}$ following (**Wodke et al., 2013**), and the lactate secretion rate is kept unconstrained as the optimal FBA solution will always route as much flux as possible through the acetate pathway, yielding one more ATP per pyruvate. The pyruvate secretion rate is also left unconstrained; this reaction only carries flux under certain in silico gene knockout conditions.

## Nucleotide metabolism

A schematic diagram of nucleotide metabolism in JCVI-syn3A is presented in **Figure 7**. The JCVI-syn3A annotation contains a putative ribonucleoside (RNS) ABC import system (*rnsD/*0008 through *rnsB/*0011), which is assumed to import all nucleosides (ribo- and deoxyribo-), but no individual bases or free ribose, in accordance with the experimental characterization of the ribonucleoside ABC importer in *Streptococcus mutans* (**Webb and Hosie, 2006**). While intact nucleotides are rarely taken up as a whole, there have been reports of *M. mycoides capri* LC Y being capable of taking up deoxymononucleotides (**Neale et al., 1984**; **Youil and Finch, 1988**). However, no gene has been identified for this functionality in that strain, nor is there any hint that the minimized JCVI-syn3A still possesses this ability. We note that competition experiments suggest distinct uptake systems for nucleosides and nucleotides (**Maniloff, 1992**), that is the RNS importer should not allow for

nucleotide uptake. The presence of several nucleoside kinases as well as phosphoribosyltransferases in JCVI-syn3A suggests that this nucleotide uptake ability, if present at all, cannot provide nucleotides in sufficient amounts and thus no nucleotide uptake reactions are included. The mechanism of nucleobase uptake in mycoplasmas has not been established, but a proton symport mechanism has been suggested (*Maniloff, 1992*). This mechanism is used in the model as well, as done in other mycoplasma models (*Wodke et al., 2013*; *Suthers et al., 2009*; *Karr et al., 2012*; *Ferrarini et al., 2016*).

The further reconstruction of nucleotide metabolism is aided by the extensive experimental studies on nucleotide salvage pathways in *M. mycoides capri* LC Y (*Mitchell and Finch, 1977*; *Mitchell et al., 1978*; *Mitchell and Finch, 1979*; *Neale et al., 1983a*; *Neale et al., 1983b*; *Cocks et al., 1988*) and *M. mycoides mycoides* SC (*Wang et al., 2001*; *Welin et al., 2007*). The reactions detected or inferred are in agreement with the existing annotations, and help to refine possible specificities and suggest additional functionalities.

The genome of JCVI-syn3A contains three deoxynucleoside kinases: *tdk/*0140, *dak1/*0330, and *dak2/*0382. The thymidine kinase, 0140, is assumed to phosphorylate both thymidine and deoxyuridine. As further discussed in Appendix 1, it is furthermore assumed that *dak1/*0330 and dak2/0382 both act on deoxyadenosine, deoxyguanosine and deoxycytidine, but not significantly on any ribonucleosides. Therefore, AMP, GMP and UMP are only formed directly from their respective bases by the corresponding phosphoribosyltransferases (*hptA/*0216, *apt/*0413, and *upp/*0798).

The genome of JCVI-syn3A contains several mononucleotide kinases (*tmk/*0045, *gmk/*0203, *cmk/*0347, *pyrH/*0537, and *adk/*0651) that can phosphorylate all (deoxy-)mononucleotides except for dUMP (*Neale et al., 1983b*), but in line with other mycoplasmas (*Pollack et al., 2002*), the genome of JCVI-syn3A contains no gene for nucleoside diphosphate kinase (*ndk*). Instead, the glycolytic enzymes phosphoglycerate kinase (PGK) and pyruvate kinase (PYK) have been found to phosphorylate other dinucleotides besides ADP in several mycoplasmas (*Pollack et al., 2002*). Specifically, PYK was found to act on all (deoxy-)dinucleotides and PGK was found to act on all purine (deoxy-)dinucleotides, but not on pyrimidines. These reactions complete the pathways from the mononucleotides to the final (deoxy-)trinucleotides. We note that the apparent absence of cytidine kinase activity implies that the only route to cytidine nucleotides goes through CTP synthase (CTPS, *pyrG/*0129; aminating UTP to CTP). All deoxytrinucleotides except dTTP can be obtained either from their deoxynucleosides or from the corresponding ribodinucleotide through the action of ribonucleotide diphosphate reductase (RNDR, *nrdE/*0771 through *nrdF/*0773).

In addition to these synthetic pathways, JCVI-syn3A also contains several catabolic reactions. The phosphorolysis of purine nucleosides observed in *M. mycoides capri* LC Y (*Mitchell and Finch, 1977*; *Mitchell et al., 1978*) can be carried out by purine nucleoside phosphorylase (PNP, *punA/*0747). PNP is assumed to also act on purine deoxynucleosides, as this activity has been demonstrated in *M. capricolum* and *M. gallisepticum* (*McElwain and Pollack, 1987*). However, no pyrimidine nucleoside phosphorylase activity is assumed to be left in JCVI-syn3A (see Appendix 1).

Hydrolase activity against several mononucleotides (GMP, dAMP, dGMP, dUMP and dTMP), a dinucleotide (dCDP) and several trinucleotides (CTP, dCTP and dUTP) has been experimentally observed (*Mitchell et al., 1978*; *Neale et al., 1983a*; *Cocks et al., 1988*). A putative dUTPase is annotated in JCVI-syn3A (*dut/*0447). We note the presence of several hydrolases of unclear function that may possibly carry out the other reactions. Thus, all these hydrolysis reactions are included in the network, without the assignment of a specific gene. Experimental studies suggest a common enzyme for all deoxymononucleotidase reactions, but separate enzymes for dUTP and dCTP hydrolysis (*Cocks et al., 1988*).

While the observation of dUTPase activity in the natural *M. mycoides capri* does not itself imply that this activity has to be present in JCVI-syn3A, possible DNA incorporation of dUTP (formed from UDP through RDNR and subsequent phosphorylation) is a problem all organisms must contend with, as exemplified by the essentiality of dUTPase in *E. coli* (*el-Hajj et al., 1988*) and *S. cerevisiae* (*Gadsden et al., 1993*). The situation is exacerbated in JCVI-syn3A after the deletion of the repair enzyme uracil-DNA glycosylase (MMSYN1_0436), such that hydrolysis of dUTP is the only apparent defense mechanism against its incorporation into DNA. Inclusion of this reaction (*Figure 7*; DUTPDP) is therefore warranted even if the annotation of the candidate gene *dut/*0447 is only tentative. To avoid RNDR being in silico essential solely by virtue of being the only source of dUTP in the network

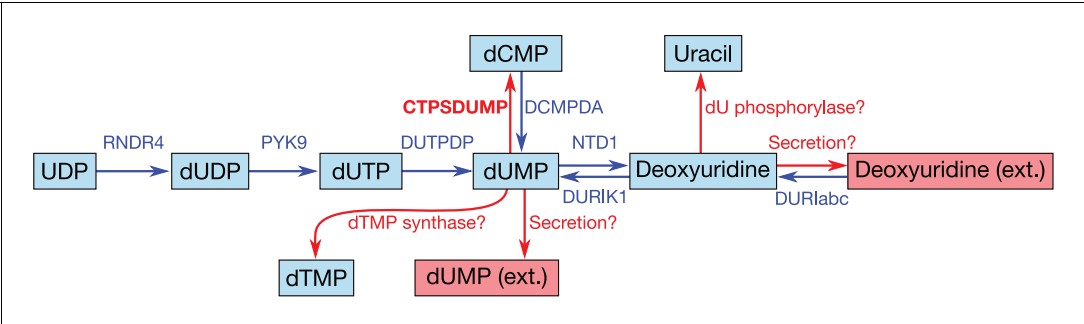

**Figure 8.** Apparent dead-end of dUMP/deoxyuridine and possible solutions. Internal metabolites are highlighted with cyan boxes, external ones with red boxes. Blue arrows denote reactions incorporated during model reconstruction—no reaction leads away from the dUMP/deoxyuridine pair. Red arrows denote hypothetical reactions that could possibly solve this dead-end. In the model, we have adopted the hypothetical CTP synthase reaction converting dUMP to dCMP (see also *Figure 7*; CTPSDUMP).

(through reactions RNDR4 and PYK9; see *Figure 7*), dUTPase is included directly in the biomass, rather than enforcing a minimal flux through dUTP formation.

The breakdown of dUTP to dUMP, however, raises the question of the downstream degradation. The uptake requirement of *M. mycoides capri* for some form of thymine (*Rodwell, 1969*; *Neale et al., 1983a*), in spite of availability of dUMP, indicates the absence of thymidylate synthase activity (an otherwise common usage for dUMP as a precursor for dTMP), in line with the lack of an annotation for a thymidylate synthase gene in JCVI-syn3A. While the aforementioned hydrolysis reaction (*Figure 7*; NTD1) would enable degradation of dUMP to deoxyuridine, the deletion of the pyrimidine nucleoside phosphorylase MMSYN1_0734 renders deoxyuridine a dead-end. Thus, the issue arises of how JCVI-syn3A disposes of the dUMP/deoxyuridine formed. The first possibility is through pyrimidine nucleoside phosphorylase activity either by some unidentified paralog of MMSYN1_0734, or by some side activity of the purine nucleoside phosphorylase (*punA/*0747). The second possibility would be the export of either dUMP or deoxyuridine. We note that the possibility to recycle deoxyuridine through pyrimidine nucleoside phosphorylase in the natural *M. mycoides capri* (and thereby also dUMP after its dephosporylation) renders an additional dedicated export system for either metabolite unlikely, but side activity of some other system would be possible.

While the aforementioned dependence of *M. mycoides capri* on external thymine/thymidine rules out any thymidylate synthase activity high enough to meet cellular dTTP needs, such activity has been reported for *M. mycoides mycoides* SC (*Wehelie et al., 2010*); however, the reported activity was extremely low (~10 pmol/min/mg cell protein) and no responsible gene could be identified. If such activity was present in JCVI-syn3A as well at a higher level, it might provide for a way to dispose of dUMP. Furthermore, the presence of deoxycytidylate deaminase (DctD, *dctD/*0515) enables the conversion of dCMP to dUMP (which is used in wild-type *M. mycoides capri* to ultimately convert thymine to thymidine (*Neale et al., 1983a*). No experimental evidence is available of this enzyme running in the reverse direction to aminate dUMP with free ammonia to form dCMP. However, CTP synthase (CTPS, *pyrG/*0129), which catalyzes the conversion of UTP to CTP, spends ATP and uses glutamine as an amino donor, which suggests that DctD catalyzing the amination without ATP and from free ammonia is unlikely. Instead, the question arises whether CTPS in JCVI-syn3A may have a relaxed substrate specificity. The *E. coli* CTPS was found to not act on UMP (*Lieberman, 1956*; *Long and Pardee, 1967*) or dUTP (*Scheit and Linke, 1982*), but may have activity against UDP (*Lieberman, 1956*; *Long and Pardee, 1967*). The *Lactococcus lactis* enzyme has been found to act on dUTP (*Willemoës and Sigurskjold, 2002*) as does one of the isozymes in *S. cerevisiae* (*Pappas et al., 1999*). As relaxed enzymatic substrate specificity in mycoplasmas is a common phenomenon, a further broadening of the substrate range of CTPS in JCVI-syn3A seems possible. The preceding hypothetical mechanisms are summarized in *Figure 8*. Currently, the available data does not allow for the determination of which of these potential dUMP disposal mechanisms occurs in JCVI-syn3A; however, the increased substrate spectrum of CTP synthase currently seems the most plausible, thus it is included in the model to deal with the produced dUMP.

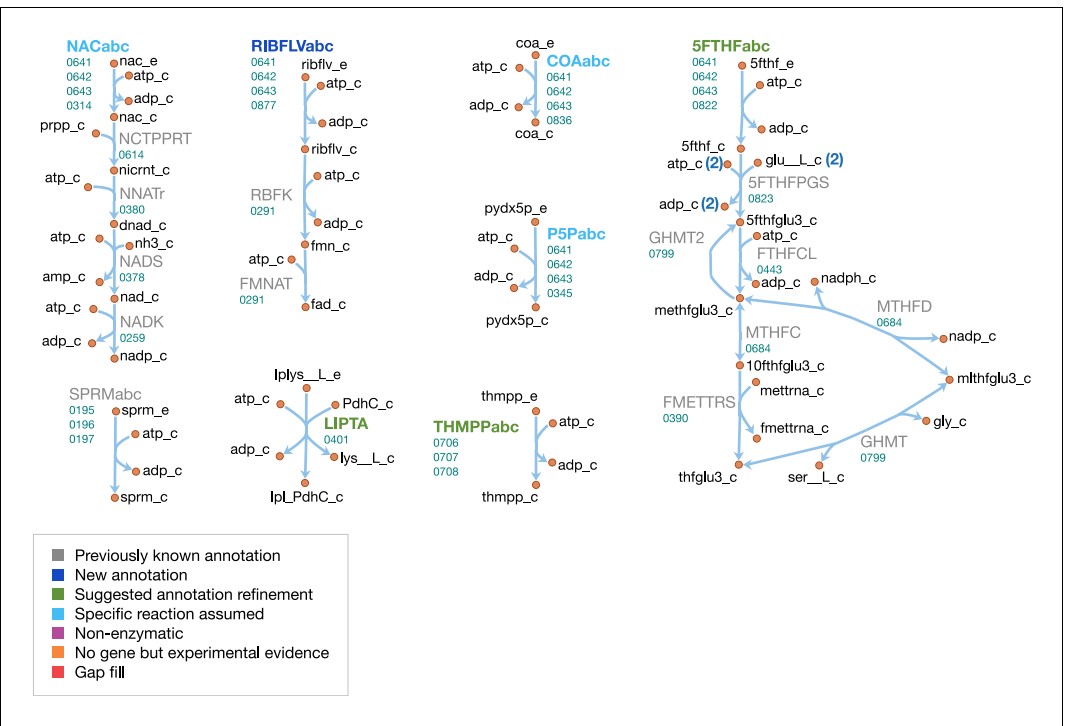

**Figure 9.** Cofactor metabolism in JCVI-syn3A. Map components and labels as in **Figure 5**.

## Cofactor metabolism

The cofactor metabolism of JCVI-syn3A is shown in **Figure 9**. Many vitamins are known to be taken up through the energy coupling factor (ECF) system (**Rodionov et al., 2009**), which consists of a membrane permease EcfT, a dimer of the ATPase EcfA, and a substrate-binding subunit EcfS (**Erkens et al., 2012**). The gene for the latter can either occur in a cluster with the genes for EcfA and EcfT, or several *ecfS* genes can be spread across the genome, their protein products displaying distinct substrate specificities but interacting with a common EcfT(EcfA)$_2$ module in the membrane. The JCVI-syn3A genome annotation lists three consecutive genes *ecfT*/0641, *ecfA*/0642 and *ecfA*/0643 and four *ecfS* genes spread throughout the genome (*ecfS*/0314, *ecfS*/0345, *folT*/0822, and *ecfS*/0836). Folate, riboflavin, coenzyme A, nicotinate, and pyridoxal are all imported through the ECF system. We note that the absence of related salvage enzymes in JCVI-syn3A necessitates the uptake of complete coenzyme A; this is in accordance with the experimental requirement of *M. mycoides capri* LC Y for coenzyme A rather than coenzyme A precursors (**Rodwell, 1967**; **Rodwell, 1969**), as well as the apparently incomplete coenzyme A salvage pathway already in JCVI-syn1.0. As no kinase has been identified for pyridoxal, pyridoxal phosphate is assumed to be imported directly.

For the case of folate, *folT*/0822 is assumed to be the necessary substrate-binding unit due to sequence conservation (**Danchin and Fang, 2016**) and its adjacency to *folC* (**Eudes et al., 2008**). While the exact form of folate in the SP4 medium is not known, we note that generally, 5-formyl-tetrahydrofolate (5-formyl-THF, folinic acid) is the most stable folate derivative, and is known to be imported by FolT (**Eudes et al., 2008**). Furthermore, *M. mycoides capri* LC Y was found to be unable to utilize folate itself (**Neale et al., 1981**), in line with JCVI-syn3A lacking a gene coding for dihydrofolate reductase. It is thus assumed that folate is taken up in the form of 5-formyl-THF. Genes for the proteins driving the folate cycle consist of GlyA (*glyA*/0799), FolD (*folD*/0684) and Fmt (*fmt*/0390), together with the repair enzyme YgfA (*ygfA*/0443/FTHFCL). YgfA is not only required to utilize imported 5-formyl-THF, but also to recycle the 5-formyl-THF produced from the hydrolysis of 5,10-methenyl-THF (a side reaction of GlyA (**Stover and Schirch, 1990**)). The RAST annotation pipeline (**Overbeek et al., 2014**) suggests that the putative membrane protein gene 0877 is *ribU*, coding for the riboflavin-specific ECF component. As the substrate specificity of the remaining EcfS

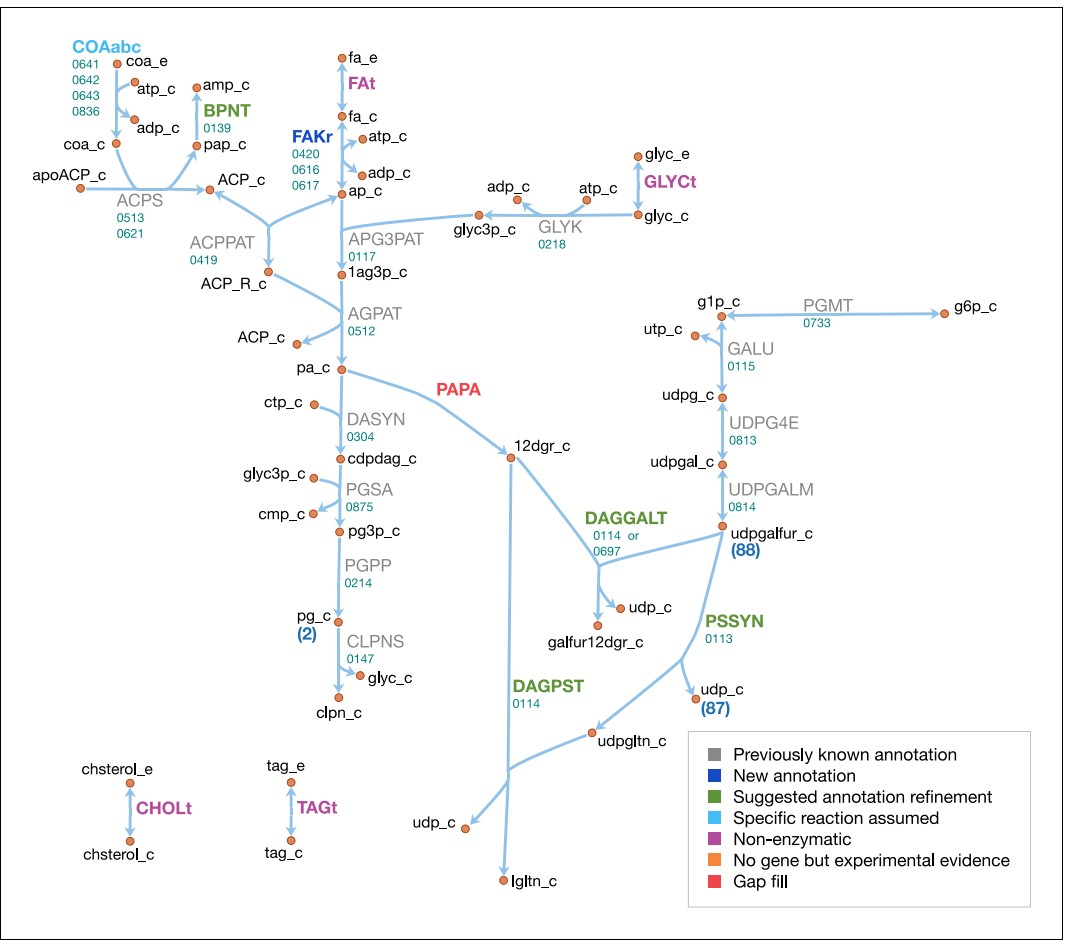

**Figure 10.** Lipid and capsule metabolism in JCVI-syn3A. Map components and labels as in *Figure 5*.

components (*ecfS/*0314, *ecfS/*0345 and *ecfS/*0836) is unclear, they are tentatively assigned to nicotinate, pyridoxal phosphate, and coenzyme A respectively. For downstream conversion, the genome contains pathways for NAD(P) and FMN/FAD formation from nicotinate and riboflavin.

Spermine and thiamine have their own uptake systems (*potC/*0195 through *potA/*0197 and *thiB/*0706 through 0708, respectively). For thiamine, the deletion of the corresponding diphosphokinase (MMSYN1_0261) suggests that thiamine diphosphate (ThDP) must be taken up directly. Sequence and structural information suggest this to be possible (see Appendix 1 and *Appendix 1—figure 1*). While free lipoate is a component of the minimal media for *M. mycoides capri* LC Y (*Rodwell, 1969*) and *M. pneumoniae* (*Yus et al., 2009*) and is a possible ECF system substrate (*Rodionov et al., 2009*), two putative lipoyl transferases have been deleted in JCVI-syn3A (MMSYN1_0224 and MMSYN1_0464), such that free lipoate cannot be used to provide the lipoyl moiety of pyruvate dehydrogenase subunit E2 (*pdhC/*0227). Instead, simultaneous import and transamidation of lipoate from lipoyllysine onto PdhC, catalyzed by the peptidase 0401, is tentatively assumed in the model (see Appendix 1).

## Lipids and capsule

The lipid and capsule metabolism of JCVI-syn3A is presented in *Figure 10*. In the metabolic reconstruction, the import of four lipid components is necessary: free fatty acids, glycerol, cholesterol and triacylglycerol. These are all assumed to be imported through passive processes. Free fatty acids have been found to be incorporated into the membrane of the mollicute *Spiroplasma floricola* both actively and via passive diffusion (*Tarshis and Salman, 1992*). Glycerol is usually imported through dedicated transport systems, yet the glycerol permease GlpF (MMSYN1_0217) has been deleted in JCVI-syn3A. However, it is known that cells can take up glycerol by passive membrane permeation

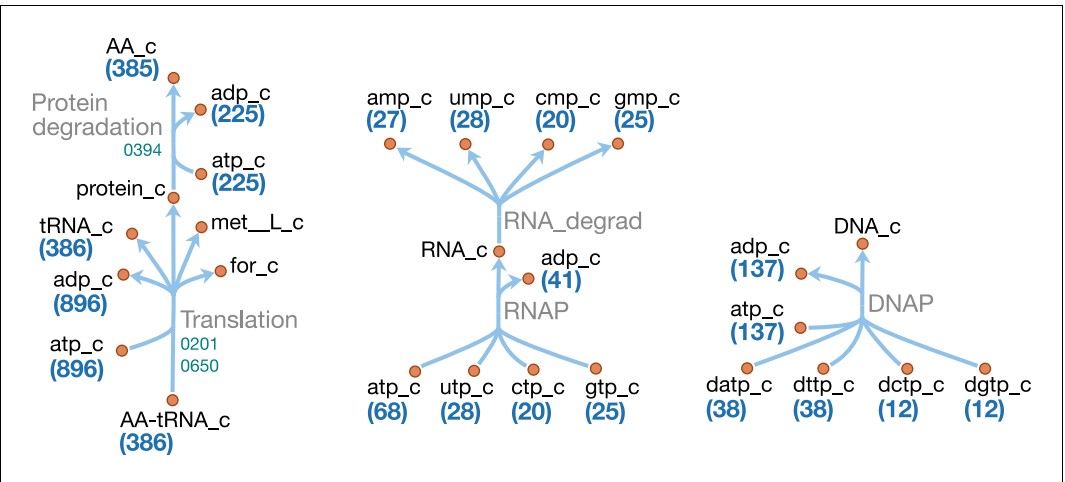

**Figure 11.** Macromolecule metabolism in JCVI-syn3A. Map components and labels as in *Figure 5*. The detailed (amino acid-specific) stoichiometry of the protein synthesis and degradation reactions can be found in *Supplementary file 4*. Protein synthesis reactions for the proteins explicitly included in the model (apo-ACP, dUTPase and PdhC) are analogous to the translation reaction shown and are therefore not included in the map.

(*McElhaney et al., 1973*; *Eze and McElhaney, 1981*) and physicochemical data suggests this could provide sufficient glycerol uptake to fuel the lipid synthesis needs of JCVI-syn3A (see Appendix 1). Cholesterol is known to be incorporated into membranes spontaneously (*McLean and Phillips, 1981*; *Bittman et al., 1990*) and has been suggested to be incorporated by simple physical absorption in *M. mycoides capri* cells as well (*Rigaud and Leblanc, 1980*). Triacylglycerol was identified as a membrane component in *M. mycoides capri*, but it is not known whether it is still included in the membrane of JCVI-syn3A, so a passive uptake reaction for it is included, noting that the presence of triacylglycerol in the biomass expression then only affects the model by lowering the amounts needed of other lipid species.

The existing annotation with the refinements for the two glycosyltransferases *epsG/*0113 and *cps/*0114 contains nearly complete pathways to produce all membrane components identified in the biomass, with the only gaps occurring in the fatty acid phosphorylation and diacylglycerol (DAG) production pathways. JCVI-syn3A shares the fatty acid utilization pathway from *Staphylococcus aureus* (*Parsons et al., 2014*), which starts with phosphorylation of free fatty acids in the membrane and subsequent binding to glycerol phosphate (by *plsY/*0117) and acyl carrier protein (ACP, by *plsX/*0419). Holo-ACP is formed from apo-ACP and coenzyme A by ACP synthase (*acpS/*0513), releasing adenosine 3',5'-bisphosphate (pAp). The DHH phosphoesterase family protein *ytqI/*0139 is 30% identical to the experimentally confirmed bifunctional oligoribonuclease/pAp phosphatase NrnA from *M. pneumoniae* (MPN140) (*Postic et al., 2012*). Thus, it is assumed that *ytqI/*0139 catalyzes the degradation of pAp to AMP.

Fatty acid kinase consists of a kinase FakA and a fatty acid-binding protein FakB. Both JCVI-syn3A and *S. aureus* contain two copies of FakB, and these have been demonstrated in *Staphylococcus aureus* to display distinct substrate specificities, one acting preferably on unsaturated fatty acids and the other on saturated fatty acids. Assuming similar specificities in JCVI-syn3A would be consistent with the assumed prevailing fatty acids (palmitate and oleate). No annotation for FakA exists in JCVI-syn3A, however, (*Parsons et al., 2014*) reports the location of the *fakA* gene in *M. pneumoniae* (MPN547), which shows 33% sequence identity to *fakA/*0420 and shares the same genomic context (located upstream of *plsX*). Thus, it is assumed that *fakA/*0420 is the missing FakA subunit, completing the lipid assembly pathway from free fatty acids to cardiolipin.

As discussed above, JCVI-syn3A has the pathway from glucose-6-phosphate to UDP-galactofuranose, the sugar building block for galactosyl-diacylglycerol (Gal-DAG) and the galactan lipopolysaccharide. In *M. genitalium*, DAG is produced from phosphatidate via phosphatidate phosphatase (PAPA). While no such enzyme is annotated in JCVI-syn3A, and a BLAST search with the *M. genitalium* gene (MPN455) against JCVI-syn3A scores no hits, we note that the presence of a number of

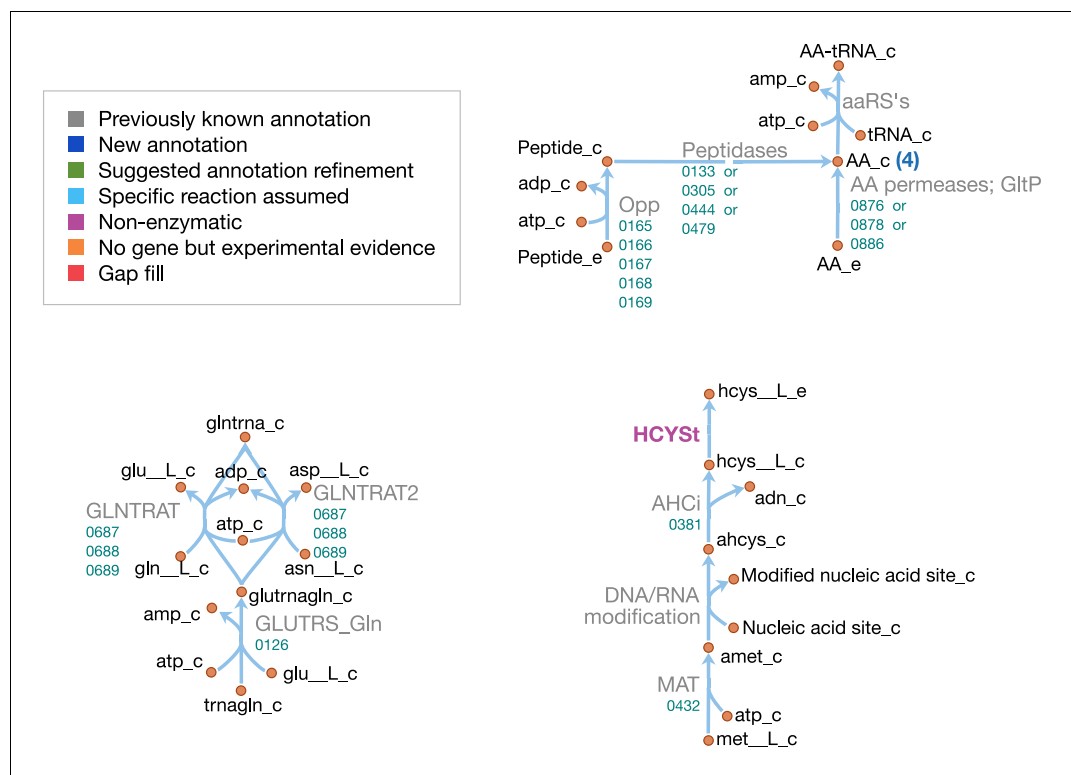

**Figure 12.** Amino acid metabolism in JCVI-syn3A. Map components and labels as in *Figure 5*. As amino acid metabolism in JCVI-syn3A constitutes sets of analogous reactions (for each amino acid or peptide), we use generic reactions in the upper right part of the map. The ABC importer Opp catalyzes tetrapeptide uptake reactions in the model ([amino acid]4abc in *Supplementary file 4*); the AA permeases (incl. GltP) catalyze amino acid proton symport reactions ([amino acid]t2[p]r in *Supplementary file 4*). The peptidases catalyze peptide hydrolysis reactions ([amino acid]4P in *Supplementary file 4*). The aminoacyl tRNA synthetases ('aaRS's' in the map) catalyze charging of tRNAs ([amino acid]TRS in *Supplementary file 4*). Synthesis of Gln-tRNA_Gln requires transamidation of initially mischarged Glu-tRNA^Gln and the corresponding reactions are shown on the lower left. In the *S*-adenosylmethionine pathway on the lower right, we note that nucleic acid modification reactions (indicated by the edge labeled 'DNA/RNA modification') were not included in the model due to lack of sufficient information on kind and abundance of nucleic acid modifications in JCVI-syn3A.

unassigned phosphatases in the genome of JCVI-syn3A makes it plausible that one of them could act on phosphatidate. An alternative possibility might be phosphatidate phosphatase side activity by phosphatidylglycerophosphatase (*pgpA/*0214), which has been reported for phosphatidylglycerophosphatase B (PgpB) in *E. coli* (*Dillon et al., 1996*). With no gene yet for phosphatidate phosphatase but plausible candidates, this reaction is included as a gap fill.

The product DAG serves as the lipid moiety for the synthesis of lipogalactan (catalyzed by *epsG/* 0113 and *cps/*0114, see Section 'Biomass composition and reaction') and Gal-DAG. Lacking further evidence, Gal-DAG is assumed to be formed by either of the two glycosyltransferases, *cps/*0114 and 0697.

Possibly related to lipid metabolism, the existing annotation lists a putative choline/ethanolamine kinase (0906); however, as *M. mycoides capri* LC is experimentally known to not produce phosphatidylcholine (*Plackett, 1967a*), we assume that this kinase has some yet to be determined substrate and do not include it in the model.

## Macromolecules and amino acids

Schematics of the macromolecular and amino acid reaction networks are provided in *Figure 11* and *Figure 12*. The genome of JCVI-syn3A contains a putative oligopeptide ABC importer (Opp/Ami, *oppB/*0165 through *oppA/*0169). In *Lactococcus lactis*, the Opp system has been found to import peptides of four to at least 35 amino acids with little dependence of uptake rates on peptide length

or amino acid composition (*Detmers et al., 1998*; *Doeven et al., 2004*). For the sake of simplicity, peptides imported are assumed to be representative homotetrapeptides of all amino acids except cysteine, since cysteine contained in peptides cannot be easily utilized by mycoplasmas (*Yus et al., 2009*). The model assumes that any of four peptidases (*ietS*/0133, 0305, 0444, and 0479) can split these peptides into individual amino acids.

In addition, the glutamate/aspartate permease *gltP*/0886, as well as two amino acid permeases of unknown specificity (0876 and 0878) have been identified in JCVI-syn3A. The substrate specificities of these two amino acid permeases are not known. However, *M. mycoides capri* LC Y has been found capable of taking up all amino acids in their free form (*Rodwell, 1969*) (glutamic and aspartic acid not investigated); thus, the least constraining assumption is made that both permeases can take up all amino acids, except for glutamic and aspartic acid, whose uptake is already enabled by GltP. Proton symport reactions are assumed for each amino acid except glutamate and aspartate, which are symported by GltP while translocating two protons per substrate, as observed in *E. coli* (*Tolner et al., 1995*).

In order to distinguish between free nucleotides and amino acids and those incorporated in nucleic acids and proteins, explicit macromolecular synthesis reactions (DNA replication, RNA transcription, protein translation) are included that consume amino acids and nucleotides according to the assumed macromolecular compositions. These reactions produce representative DNA, RNA and protein species that enter the biomass reaction according to the mass fractions in *Appendix 1— table 1*. Similar macromolecular synthesis costs as in *E. coli* (*Feist et al., 2007*; *Neidhardt et al., 1990*) are assumed, that is 1.37 ATP/nucleotide in DNA synthesis, 0.41 ATP/nucleotide in RNA synthesis, 2 ATP/amino acid in tRNA charging, and 2.32 ATP/peptide bond in protein synthesis. Protein products are represented as species containing 385 amino acids (based on the average protein length obtained from the JCVI-syn3A proteomics, and accounting for the N-terminal methionine cleaved off the nascent peptide–see below). DNA and RNA are represented as species of 100 bases each to keep the nucleotide reaction coefficients small (and also since no average RNA length is known).

In addition, separate translation reactions are included for three additional proteins: ACP and PdhC, whose prosthetic group attachment is included in the model; and dUTPase, which is included explicitly in the biomass. Protein translation in the model uses charged tRNA that are produced from one synthetase for each amino acid, except for glutamine. Instead, glutamyl-tRNA(Gln) is transamidated to glutaminyl-tRNA(Gln) via an amidotransferase (*gatB*/0687 through *gatC*/0689) using glutamine or asparagine as amino donor. Translation is assumed to be initiated with formylated methionyl initiator tRNA (fMet-tRNA$_{fMet}$). While the natural *M. mycoides capri* has been found to be able to initiate protein synthesis with the unformylated species (*Neale et al., 1981*), the folate cycle (including the Met-tRNA$_{fMet}$ formyltransferase *fmt*/0390) is quasi-essential in JCVI-syn3A and the assumption is hence made that formylation cannot be omitted in JCVI-syn3A. Translation reactions for both apo-ACP and the generic protein species include deformylation and methionine cleavage, in line with ca. 80% of proteins in a proteome assumed to have the initial methionine cleaved (*Frottin et al., 2006*), and based on the alanine in second position of the apo-ACP sequence *acpA*/0621, favorable for methionine cleavage (*Frottin et al., 2006*). For PdhC (*pdhC*/0227) and dUTPase (*dut*/0447), only deformylation is considered, as the phenylalanine and isoleucine in second position of their sequences are not favorable for methionine cleavage (*Frottin et al., 2006*). The excess formate is assumed to be secreted by passive means.

Explicit protein and RNA degradation reactions are included in the model to account for protein and mRNA turnover in the cell and to regenerate the amino acid and nucleotide pools, respectively. In line with most other mycoplasma genomes (*Gur and Sauer, 2008*), the JCVI-syn3A genome only contains two AAA+ proteases, Lon (*lon*/0394) and (putatively) FtsH (*ftsH*/0039). Lon has been found to degrade ssrA-tagged peptides in place of ClpXP in *Mesoplasma florum* (*Gur and Sauer, 2008*) and *M. pneumoniae* (*Ge and Karzai, 2009*). It is tentatively assumed to be the main protease for protein turnover in general in the metabolic reconstruction. An ATP expense of 225 ATP per protein of 385 residues is assumed (see Appendix 1).

Lower bounds on the protein and RNA degradation reactions of $3.5 \times 10^{-4}$ mmol gDW$^{-1}$ h$^{-1}$ and $7.7 \times 10^{-3}$ mmol gDW$^{-1}$ h$^{-1}$, respectively, are imposed assuming similar degradation rates as in *M. pneumoniae* (*Wodke et al., 2013*), which shares the protease Lon. Assuming that the entire proteome is subject to turnover (as observed in *M. pneumoniae* (*Maier et al., 2011*)), the protein

degradation constraint corresponds to an average protein half life time of 25 hr, which describes the time an average protein persists throughout cell divisions before being actively degraded; it is based on $^{13}$C mass-spectrometry measurements of protein degradation in *M. pneumoniae* that corrected for protein dilution by cell growth (*Maier et al., 2011*). The protein degradation constraint thus accounts for the ATP expense arising from the slow (compared to the cell cycle time) but non-zero active protein degradation. With an ATP consumption of 225 ATP per degraded protein, protein breakdown thus requires an ATP expense of 0.079 mmol gDW$^{-1}$ h$^{-1}$. In addition, the imposed protein degradation flux implies an additional ATP cost due to the required additional protein synthesis (see Section 'GAM/NGAM'). The RNA degradation constraint corresponds to an mRNA half-life time of around 1 min, assuming mRNA to account for ~5% of all cellular RNA (*Westermann et al., 2012*; *Rosenow et al., 2001*) (and assuming other RNA degradation to be negligible in the exponential growth phase). This is comparable to the mRNA half-life time of 2 min determined in *Mycoplasma gallisepticum* (*Kirk and Morowitz, 1969*). While RNA degradation itself is assumed to not consume ATP, it incurs an indirect ATP cost due to the required additional RNA synthesis, analogous to the situation for protein degradation.

Finally, we note the presence of methionine adenosyltransferase (*metK*/0432), which produces the methyl donor *S*-adenosylmethionine used by several nucleic acid methylation enzymes; these nucleic acid modifications are, however, not included in the model due to lack of sufficient information on kind and abundance of such modifications in JCVI-syn3A. The demethylation product *S*-adenosylhomocysteine is broken down by *S*-adenosylhomocysteine hydrolase (*mtnN*/0381) to homocysteine. As there is no obvious way for further breakdown of homocysteine in JCVI-syn3A, it is assumed to be secreted for the time being. As a precedent, secretion—rather than further conversion—of homocysteine has been suspected in *Plasmodium falciparum* (*Beri et al., 2017*). As the magnitude of the required efflux rate is expected to be small, secretion is furthermore assumed to occur via passive permeation.

## Ion uptake

The ion transport reactions of JCVI-syn3A are shown in *Figure 13*. The JCVI-syn3A annotation lists importers for magnesium (*mgtA*/0787) and phosphate (Pst system, *pstS*/0425 through *pstB*/0427); *Danchin and Fang (2016)* have suggested the putative magnesium import gene *corA*/0879 could import calcium as well based on its similarity to the *B. subtilis* calcium importer YloB (*Raeymaekers et al., 2002*). While the sequence identity of *corA*/0879 to YloB (CAB31439) is lower than between YloB and the deleted calcium importer MMSYN1_0246 (22% vs. 30%), we note that the putative calcium-binding sites in YloB (*Raeymaekers et al., 2002*) are equally well conserved in both JCVI-syn3A genes, and include *corA*/0879 as the calcium importer. Active transport processes for Na$^+$ and K$^+$ are inferred from biochemical evidence (see Appendix 1). Based on this, an ATP-consuming K$^+$/Na$^+$ antiport reaction is catalyzed by KtrAB (*natA*/0685 and *trkA*/0686) in the model, and an Na$^+$/H$^+$ antiport reaction is included as well without a specific gene assignment. Import systems have not been identified for sodium or chloride, but chloride has been proposed to freely permeate the membrane of *M. mycoides capri* (*Schummer and Schiefer, 1983*) and sodium has been suggested to 'leak' through membranes of *M. gallisepticum* (*Linker and Wilson, 1985*). Alternatively, leakage through some other transporter cannot be ruled out. Thus, passive uptake reactions are assumed for sodium and chloride. Finally, a passive H$^+$ influx reaction is used, which is counteracted by the proton-extruding ATPase. Lacking more information, all ion uptake rates in the model are left unconstrained.

## Damage reactions

On top of the enzymes involved in required metabolic reactions, we note the presence of possible damage reactions and repair enzymes to repair this damage. As discussed, 5,10-methenyl-THF is hydrolyzed to 5-formyl-THF as a side reaction of GlyA (*Stover and Schirch, 1990*) and YgfA catalyzes the opposite direction. 5-formyl-THF is not only the only form of THF taken up in the model, but is also a potent inhibitor of folate-related enzymes (*Stover and Schirch, 1992*), and as such YgfA is an important repair enzyme. To account for the known damage reaction of 5,10-methenyl-THF hydrolysis, a small lower bound (0.01 mmol gDW$^{-1}$ h$^{-1}$) is imposed to ensure non-zero flux through this reaction. Reconversion of 5-formyl-THF to 5,10-methenyl-THF via YgfA consumes 1 ATP

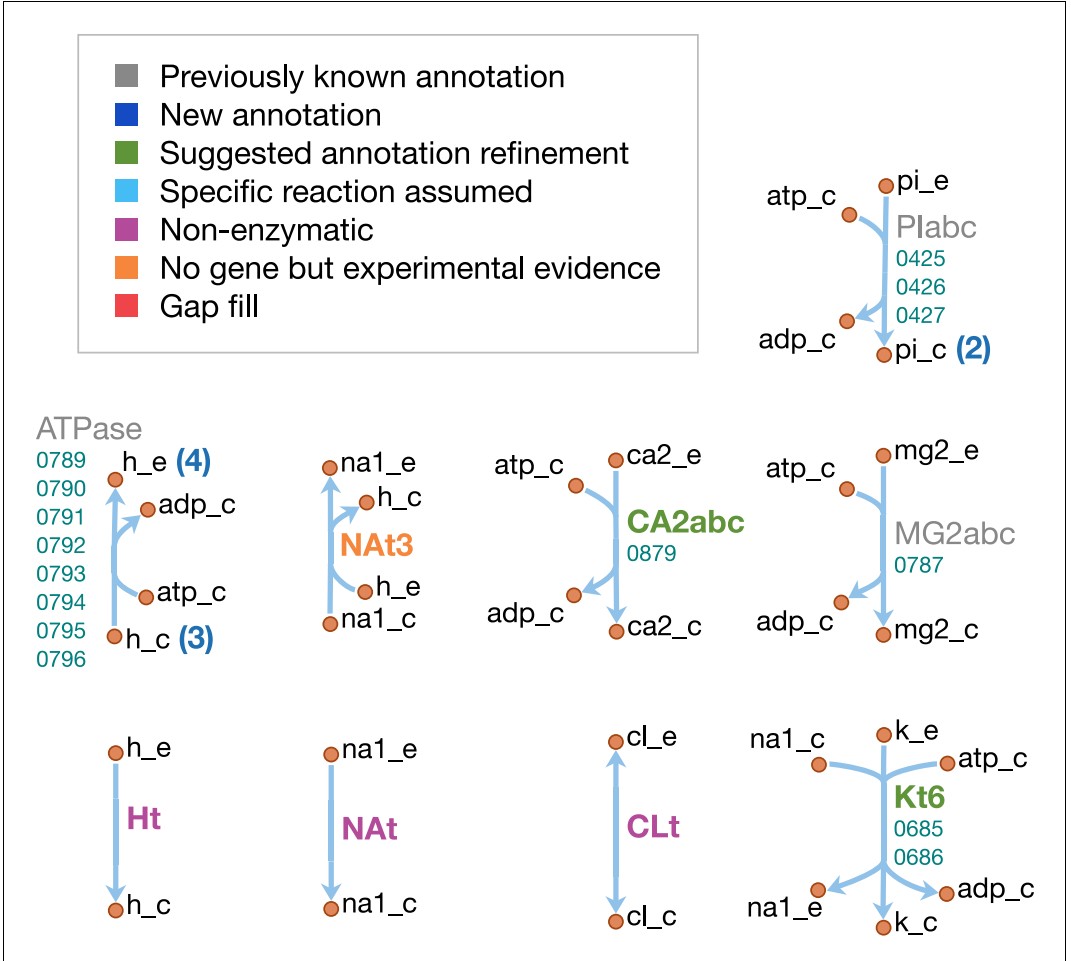

**Figure 13.** Ion transport reactions in JCVI-syn3A. Map components and labels as in *Figure 5*.

per 5-formyl-THF and thus requires 0.01 mmol gDW$^{-1}$ h$^{-1}$ ATP. Another damage preemption/repair enzyme included in the model is dUTPase (*dut*/0447), which hydrolyses dUTP and prevents its erroneous incorporation into DNA. Rather than enforcing a minimal flux through this reaction, dUTPase is included directly in the biomass for technical reasons (see Section 'Nucleotide metabolism').

## GAM/NGAM

In order to account for cellular energy expenses, the growth- and non-growth-associated maintenance costs (GAM and NGAM) need to be included in the model. The GAM describes the energy cost associated with cellular growth, and therefore enters the biomass reaction as a certain amount of ATP spent per unit biomass production. The NGAM describes the basic, growth rate-independent cellular energy requirements and is therefore frequently implemented as a lower constraint on a separate ATP hydrolysis reaction (*Feist et al., 2006*; *Orth et al., 2011*; *Wodke et al., 2013*). Both parameters can be measured experimentally (e.g. from chemostat measurements (*Thiele and Palsson, 2010*)); however, to our knowledge no such measurements are available for any mycoplasma, and thus the cellular energy expenses must be estimated differently.

The GAM consists of a component that can be related to macromolecular synthesis energy costs and a non-quantifiable part. The macromolecular synthesis costs are outlined in Section 'Macromolecules and amino acids'; for the macromolecular composition of JCVI-syn3A, they yield a total cost of 21.54 mmol gDW$^{-1}$ ATP. For the non-quantifiable part, 25 mmol gDW$^{-1}$ ATP is used following the previously published *M. pneumoniae* model (*Wodke et al., 2013*). Together with the ATP costs for macromolecular synthesis, this yields a total GAM of 46.54 mmol gDW$^{-1}$, which is

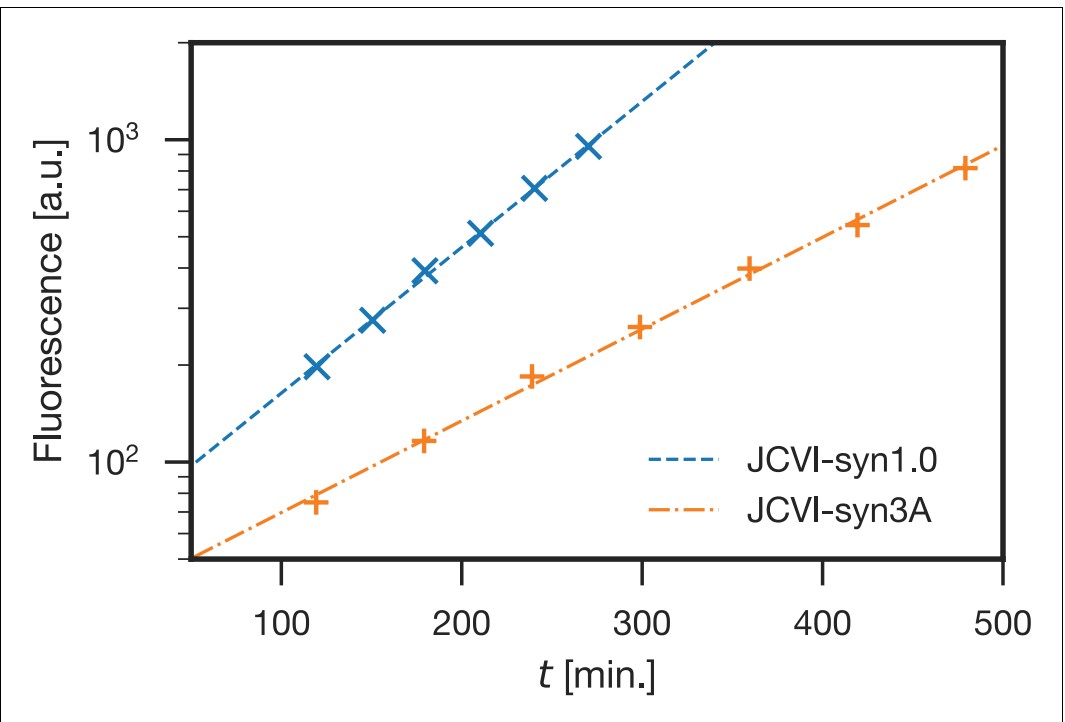

**Figure 14.** Comparison of growth curves of JCVI-syn1.0 and JCVI-syn3A. JCVI-syn1.0 has a doubling time of 66 min (blue; '×' markers), whereas JCVI-syn3A has a doubling time of 105 min (orange; '+' markers). Doubling times ($t_d$) were calculated as described in Section 'Materials and methods', plotting fluorescence staining of cellular DNA vs. time, fitted by exponential regression curves. The regression curves for JCVI-syn1.0 and JCVI-syn3A have $R^2$ values of 0.9986 and 0.9976, respectively.

comparable to the 53.95 mmol gDW$^{-1}$ estimated for *E. coli* (*Orth et al., 2011*). The use of these values is supported by the overall conservation of the gene expression apparatus.

The NGAM captures non-growth related energy expenses, such as macromolecular turnover and maintenance of the transmembrane pH gradient, which has been suggested to be a considerable energy expense in mycoplasmas (*Wodke et al., 2013*). These two energy expenses are included in the JCVI-syn3A model as an approximate NGAM. Rather than accounting for all NGAM expenses through a single ATP hydrolysis reaction, the total NGAM is distributed across several reactions (akin to the quantifiable GAM fraction). The total macromolecular turnover costs can be obtained from the assumed protein and RNA degradation fluxes ($3.5 \times 10^{-4}$ mmol gDW$^{-1}$ h$^{-1}$ and $7.7 \times 10^{-3}$ mmol gDW$^{-1}$ h$^{-1}$, respectively; see Section 'Macromolecules and amino acids'); the ATP costs in protein and RNA synthesis; and the associated costs for tRNA charging and nucleotide phosphorylation, respectively. Specifically, a constant part of the protein/RNA synthesis fluxes (and, preceding these, tRNA charging and nucleotide phosphorylation) is routed through protein/RNA degradation, and only the surplus beyond this constant flux actually contributes to model growth. These turnover-associated ATP costs amount to 2.73 mmol gDW$^{-1}$ h$^{-1}$ in the model for JCVI-syn3A. The maintenance of a transmembrane pH gradient in *M. mycoides capri* PG3 has been experimentally demonstrated (*Benyoucef et al., 1981a*; *Benyoucef et al., 1981b*) and an approximate value for the H$^+$ extrusion rate by the transmembrane ATPase has been obtained (*Benyoucef et al., 1981b*). The observed rate of ~70 nmol/min/mg cellular protein corresponds to a proton flux of ~2.30 mmol gDW$^{-1}$ h$^{-1}$ in JCVI-syn3A, which translates to an ATP consumption of 0.57 mmol gDW$^{-1}$ h$^{-1}$ (4H$^+$/ATP). This value is hence implemented as a lower bound on the model ATPase reaction. The resulting total NGAM in the model for JCVI-syn3A is then 3.30 mmol gDW$^{-1}$ h$^{-1}$, which comes out similar to the iJO1366 model for *E. coli* (3.15 mmol gDW$^{-1}$ h$^{-1}$) (*Orth et al., 2011*).

## Steady state fluxes

With the chosen parameters, the model yields an in silico growth rate of μ = 0.34 hr$^{-1}$, corresponding to a doubling time of $t_d = 2.02$ hr ($t_d = \ln(2)/\mu$); this comes close to the experimental doubling

time of ca. 105 min (see *Figure 14*). This exact agreement is sensitive to the choices for uptake/ secretion and GAM parameters however (see Section 'Sensitivity analysis' in Appendix 1 and *Appendix 1—figure 2*), and the in silico growth rate should thus be more understood as a provisional prediction. This does not constitute a problem for the subsequent analyses; when the impact of in silico gene knockouts on growth rate is studied in Section 'In silico gene knockouts and mapping to in vivo essentiality', nearly all knockouts either abolish the growth rate entirely (lethal knockouts) or do not affect it at all. Thus, this analysis is not affected by the rather qualitative nature of the growth rate prediction by the model.

In the rest of this section, the steady-state fluxes produced by the model are compared to literature data, and to theoretical flux limits obtained from protein abundances and enzyme turnover numbers ($k_{cat}$). While no fluxomics data is yet available for JCVI-syn3A, some experimental fluxes from the literature allow for a few comparisons. The purine incorporation flux into RNA has been determined for *M. mycoides capri* LC Y (*Mitchell et al., 1978*) as 0.5–1.0 nmol/min/mg cellular protein, corresponding to 0.016–0.033 mmol gDW$^{-1}$ h$^{-1}$. This is close to the in silico net flux of ATP and GTP into RNA of 0.047 and 0.043 mmol gDW$^{-1}$ h$^{-1}$, respectively. Here, the net flux is defined as the difference between NTP consumed by the RNA polymerase reaction and NMP released by the RNA degradation reaction. The in silico K$^+$ uptake of 0.29 mmol gDW$^{-1}$ h$^{-1}$ also falls within a factor of two of the experimental uptake rate of 0.49 mmol gDW$^{-1}$ h$^{-1}$ (15 nmol/min/mg cellular protein) in *M. mycoides capri* PG3 (*Benyoucef et al., 1982a*). These comparisons serve as an internal consistency check on the model, demonstrating that the in silico uptake/incorporation rates as resulting from biomass composition and growth rate indeed reproduce the experimental values.

Furthermore, it has been reported that *M. mycoides capri* LC Y spends 10% of its glucose uptake on polysaccharide capsule production (*Plackett, 1967a*). The in silico fluxes through central metabolism are depicted in *Figure 6—figure supplement 1*. As can be seen, the in silico flux through phosphoglucomutase (PGMT, *deoB*/0733, leading from glucose-6-phosphate to galactan and Gal-DAG in *Figures 6* and *10*) is lower than the experimental value, amounting to only 1.8% (0.135 mmol gDW$^{-1}$ h$^{-1}$, from 7.4 mmol gDW$^{-1}$ h$^{-1}$ glucose taken up). However, the model does qualitatively reproduce the further splitting between galactan and Gal-DAG production, which gives a ratio of ~22:1 (UDP-galactofuranose consumption of 0.129 mmol gDW$^{-1}$ h$^{-1}$ vs. 0.006 mmol gDW$^{-1}$ h$^{-1}$), compared to a ratio of ~64:1 from $^3$H labeling (*Plackett, 1967a*). Thus, with the chosen parameters, the model reproduces several experimental fluxes; with the only significant difference occurring in capsule production.

FBA flux predictions were also compared to reaction flux bounds ($V_{max}$) calculated from protein abundances and enzyme turnover numbers ($k_{cat}$) (*Labhsetwar et al., 2013*; *Labhsetwar et al., 2017*). The protein abundances were derived from proteomics experiments (see Section 'Abundances of essential and non-essential proteins') and the turnover numbers were extracted from the BRENDA database (*Scheer et al., 2010*) (see *Figure 15*, *Figure 15—figure supplement 1* and Appendix 1). $V_{max}$ values could be obtained for 105 'non-pseudo' reactions (i.e. excluding exchange, biomass and macromolecular reactions). Of these, 86 had non-zero fluxes. The zero-flux reactions include for example reactions pertaining to alternative sugars, which are unused in the assumed medium. Of the reactions with non-zero fluxes, only 19 reactions required fluxes in the FBA optimal solution higher than their proteomics-derived $V_{max}$ (see *Figure 15—figure supplement 1A*). The reaction with the lowest $V_{max}$/flux ratio is adenylate kinase, which is predicted to carry a flux of 2.23 mmol gDW$^{-1}$ h$^{-1}$, compared to a proteomics-derived $V_{max}$ of only 0.01 mmol gDW$^{-1}$ h$^{-1}$. However, the $k_{cat}$ for this enzyme as found for *B. subtilis* in BRENDA is 0.053 s$^{-1}$, which falls in the lower tenth percentile of the $k_{cat}$ data for all reactions in the model. The second-smallest $V_{max}$ to flux ratio is found for aspartate-tRNA synthetase (0.11 mmol gDW$^{-1}$ h$^{-1}$ in model vs. 0.005 mmol gDW$^{-1}$ h$^{-1}$ from proteomics/$k_{cat}$); other amino acyl-tRNA synthetases with significantly low $V_{max}$/flux ratio (< 0.25) are the ones for threonine and serine. These three amino acyl-tRNA synthetases have the smallest $k_{cat}$ numbers among all amino acyl-tRNA synthetases. The third-lowest $V_{max}$ to flux ratio is found for fructose bisphosphate aldolase (7.21 mmol gDW$^{-1}$ h$^{-1}$ in model, 0.39 mmol gDW$^{-1}$ h$^{-1}$ from proteomics/$k_{cat}$). This protein has 227 copies in the cell on average, which places it among the least abundant proteins in central metabolism. Furthermore, the $k_{cat}$ value for this enzyme retrieved from BRENDA for *Bacillus cereus* is 2.95 s$^{-1}$, which is also one of the lowest values found among proteins in central metabolism. The only other reactions with a $V_{max}$/flux ratio less than 0.25 are: DASYN (*cdsA*/0304), which produces the lipid intermediate CDP-diacylglycerol; ACPS

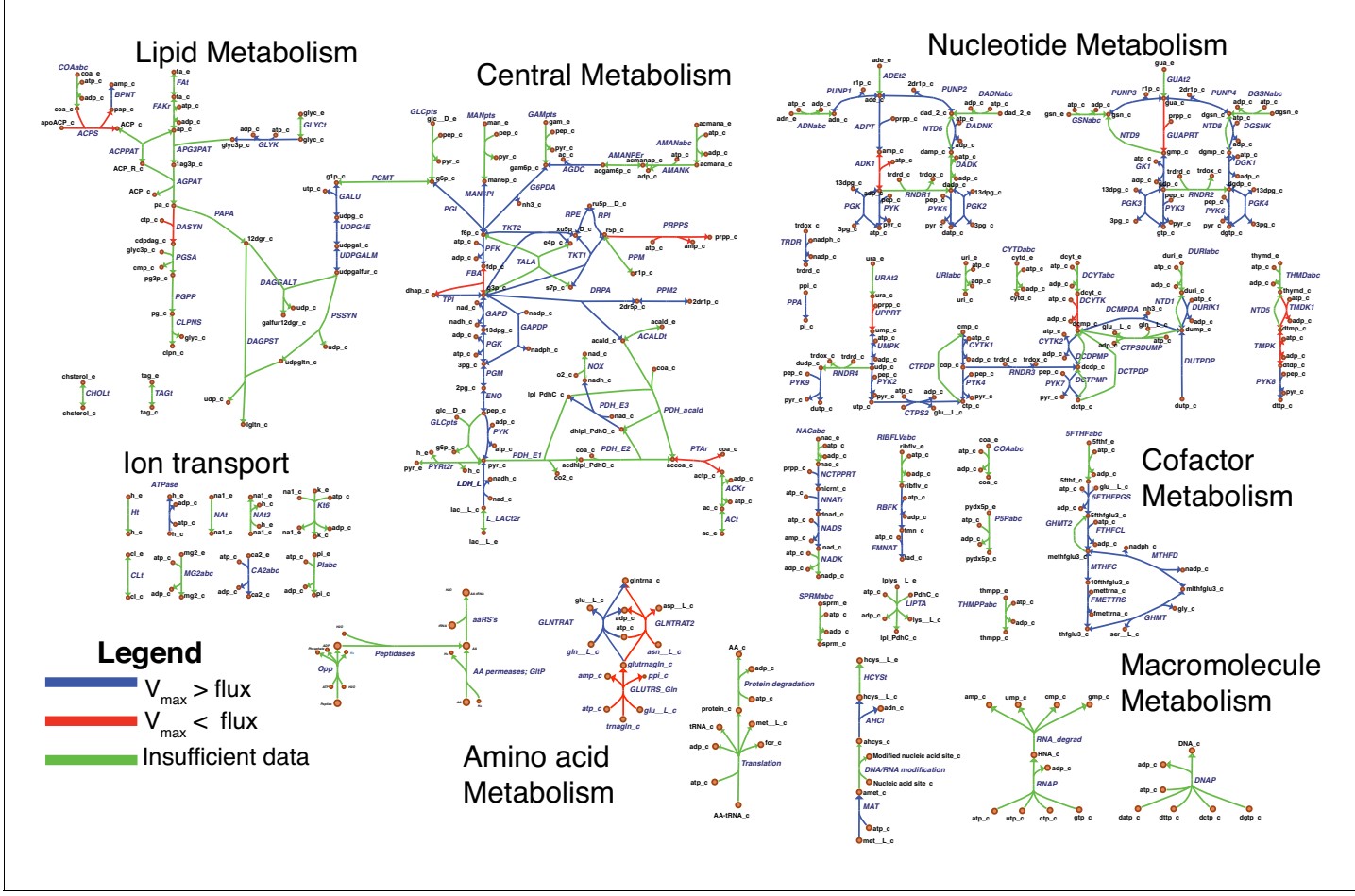

**Figure 15.** Comparison of FBA steady-state fluxes $\nu$ to maximal fluxes $V_{\max}$ obtained from protein abundances and turnover numbers from BRENDA and the literature. Map components and labels as in *Figure 5*, with reaction highlighting and gene loci/gene-protein-reaction rules omitted. Each edge is colored according to the ratio between $V_{\max}$ and $\nu$: Blue indicates $V_{\max}>\nu$, red indicates $V_{\max}<\nu$ and green indicates that no $V_{\max}$ could be obtained (because of either missing turnover number or missing protein abundance; or because reaction is not enzymatic to begin with).

The online version of this article includes the following figure supplement(s) for figure 15:

**Figure supplement 1.** Statistics of FBA steady-state fluxes $\nu$ vs. maximal fluxes $V_{\max}$ comparison (see *Figure 15*).

(*acpS/*0513), which attaches the 4′-phosphopantetheine to apo-ACP; and GUAPRT (*hptA/*0216), which produces GMP from guanine. DASYN has a $V_{\max}$ to flux ratio of 0.19, that is only slightly decreased. ACPS has both the lowest $k_{cat}$ among the model enzymes (0.001 s⁻¹) and one of the lowest protein abundances (just one copy per cell per our proteomics data). If either value turned out to be not accurate, this could easily raise the $V_{\max}$/flux ratio above the current level of 0.07. However, the discrepancy observed for GUAPRT is interesting in the light of the aforementioned mononucleotide uptake capabilities in *M. mycoides capri* (*Neale et al., 1984*; *Youil and Finch, 1988*) (see Section 'Nucleotide metabolism'). While there is no other evidence for the possible conservation of this capability in JCVI-syn3A, this flux bound might suggest that this uptake capability still at least partially exists in JCVI-syn3A, and might be worth investigating. All other reactions with $V_{\max}$ lower than the FBA flux differ by less than a factor of four; even though the FBA flux exceeds the estimated $V_{\max}$, the disagreement is rather modest. For the reactions that do show higher disagreement, we note that the $k_{cat}$ values obtained tend to be on the lower end either within the respective group of reactions, or across the model. This suggests that these $k_{cat}$ values might merit closer investigation. Overall, the proteomics-derived bounds are consistent with the FBA fluxes, with only a handful of reactions showing significant discrepancies.

It should be noted that the rates of enzyme-catalyzed reactions in vivo are typically less than $V_{max}$ to allow the cell to respond to increases in substrate concentration. Accordingly, $V_{max}$ is expected to be greater than the metabolic flux necessary to sustain the cell, such that the flux required under typical growth conditions can be achieved without enzyme saturation. In line with this argument, a histogram of $V_{max}$ values for reactions in the model shows the bulk of reactions to have a $V_{max}$ 1–3 orders of magnitudes higher than the flux required in the FBA solution (see *Figure 15—figure supplement 1B*).

## Energy usage

*Table 3* breaks down the energy consumption in JCVI-syn3A (as percentage of total ATP consumption, see Section 'Materials and methods'). The upper five categories correspond to individual subsystems of the metabolic model. The lower five categories provide a breakdown of GAM and NGAM expenses into individual components. As discussed in Section 'GAM/NGAM', a part of the protein and RNA synthesis (and, by extension, the tRNA charging and nucleotide phosphorylation fluxes) is routed through protein and RNA degradation, constituting the turnover-associated part of the NGAM; the resulting fraction of total ATP consumption is listed as 'NGAM$_{Turnover}$' in *Table 3*. Analogously, 'NGAM$_{ATPase}$' denotes the ATP expense for the ATPase-associated part of the NGAM. 'GAM$_{Macromolecules}$' and 'GAM$_{tRNA\ charging}$' denote the ATP expenses for growth-associated protein/ RNA synthesis (subsystem 'Macromolecules' in the model) and tRNA charging (subsystem 'Amino acid metabolism'), respectively. Analogously, 'Nucleotide metabolism' only includes ATP expenses beyond RNA turnover (i.e. NTP production for growth-associated nucleic acid synthesis and nucleotide usage in other subsystems). 'GAM$_{Nonquant}$' denotes the non-quantifiable fraction of the GAM.

In line with JCVI-syn3A relying heavily on uptake of pre-formed precursors and further conversion through salvage pathways only, it spends only ~6% of ATP on (small molecule) metabolic processes (i.e. lipids, cofactors and nucleotides, plus PRPP synthesis in the pentose phosphate pathway). The vast majority of energy (75%) is spent directly on growth, that is macromolecular synthesis and tRNA charging and the non-quantifiable contribution to GAM. A modest fraction of ~16% of cellular energy expenses falls to the NGAM (macromolecular turnover and ATPase). These numbers stand in striking difference to *M. pneumoniae*, for which non-growth associated maintenance accounts for 71–88% of total cellular ATP consumption (in the accounting for *M. pneumoniae*, the NGAM does not include protein/RNA turnover) (*Wodke et al., 2013*). This correlates with *M. pneumoniae* having a doubling time between 8 and 20 hr (*Yus et al., 2009*; *Wodke et al., 2013*), that is four to ten times slower than JCVI-syn3A.

The ATP breakdown also reveals that in spite of the minimal cell's heavy reliance on uptake of pre-formed precursors, transport processes only account for ~3% of ATP consumption. While the optimal FBA solution only takes up amino acids through the permeases (0876, 0878, and *gltP/*0886) using proton symport reactions, the ATP expense on transport does not increase significantly (only to ~5%) when forcing amino acid uptake through the ATP-consuming Opp peptide importer. This illustrates how JCVI-syn3A can maintain a relatively fast growth rate in spite of its extreme genome

**Table 3.** Cellular ATP expenses by category (in percent of total ATP consumption).

| Category | ATP expense [%] |
| --- | --- |
| Nucleotide metabolism | 3.6 |
| Pentose phosphate pathway | 1.7 |
| Lipid metabolism | 0.7 |
| Cofactor metabolism | 0.1 |
| Transport | 3.4 |
| GAM$_{Macromolecules}$ | 18 |
| GAM$_{tRNA\ charging}$ | 16 |
| GAM$_{Nonquant}$ | 41 |
| NGAM$_{Turnover}$ | 13 |
| NGAM$_{ATPase}$ | 2.7 |

minimization and reliance on fermentative ATP production: By importing pre-formed precursors or recovering them through salvage reactions, the cell expends a minimal amount of energy to obtain the final macromolecular precursors and passes this savings in energy along to the production of biomass.

The other important currency in the cell are reduction equivalents in the form of NADPH, which in JCVI-syn3A is produced by GapN (GAPDP, *gapdh/*0451) and, in tiny amounts, by FolD (MTHFD, *folD/*0684). The only consumer of NADPH in the model is ribodinucleotide reductase (RNDR, *nrdE/*0771 through *nrdF/*0773). In vivo, however, NADPH is expected to also be needed for expenses not captured by the model, including RNA modification (dihydrouridine synthesis) and response to oxidative stress: The reduction of protein disulfide bonds formed by oxidative stress is mediated by thioredoxin (*Ben-Menachem et al., 1997*), and coenzyme A disulfide reductase (*cdr/*0887) serves to reduce coenzyme A disulfide dimers to the free thiol-carrying monomers. NADPH production through GAPDP diverts flux from the ATP-producing GAPD/PGK branch in glycolysis, effectively incurring an ATP cost for NADPH production. In order to probe the cellular capacity for NADPH production, *Appendix 1—figure 2G* shows a plot of in silico doubling time as a function of imposed NADPH consumption (imposed via an artificial NAPDH oxidation reaction with $O_2$, introduced for testing purposes). Within a considerable margin, the doubling time rises shallowly with NADPH consumption: for example, at 3.5 mmol gDW$^{-1}$ h$^{-1}$ (a quarter of the maximally possible flux through GAPDP), the model doubling time only rises by 25% to ~2.5 hr. This suggests that even though NADPH usage is not fully captured by the model, the cell should be able to accommodate a considerable amount of NADPH demand without strong impact on the growth rate (see also Section 'Sensitivity analysis' in the Appendix 1).

Finally, there is also some experimental information that allows for a comparison of cellular energetics, specifically of basal energy expenses. In *Benyoucef et al. (1981b)*, the residual acid secretion in *M. mycoides capri* PG3 in a saline buffer after inhibition of ATPase has been measured to be around 110 nmol/min/mg cellular protein (corresponding to ~3.6 mmol gDW$^{-1}$ h$^{-1}$), which can be compared to the corresponding in silico acid secretion (which in turn is connected to ATP production). The measurements were performed in a saline buffer containing glucose but no other nutrients for growth. Under these conditions, the cell is not able to grow (*Leblanc and Le Grimellec, 1979*), but should be able to meet its basic energetic needs. Furthermore, since ATPase was inhibited with N,N′-dicyclohexylcarbodiimide (which abolishes both proton transduction and ATPase activity (*Hermolin and Fillingame, 1989*)), it should not consume ATP anymore under the experimental conditions. These conditions are simulated by setting the lower bound on ATPase proton extrusion to 0.0 mmol gDW$^{-1}$ h$^{-1}$ and changing the objective function in FBA from maximal growth rate to minimal glucose uptake. A residual acid secretion of 1.3–2.6 mmol gDW$^{-1}$ h$^{-1}$ results, which depends on the assumed lactate to acetate ratio, and falls within a factor of ~2 of the experimental value. This suggests that the basic cellular energy expenses—protein degradation, RNA and protein synthesis under non-growth conditions—are described reasonably well by this model. At the same time, hypotheses can be made as to what energy expenses could account for the observed remaining discrepancy. One expected factor is the unknown actual NADPH demand (and resulting effective ATP cost). In addition, a possibly significant energy sink not covered yet by the model are metabolite repair functions, of which thus far only two are included in the model, namely 5-formyl-THF cycloligase (*ygfA/*0443) and dUTPase (*dut/*0447). Metabolite repair usually consumes energy (*Linster et al., 2013*), and it would be interesting to see to what extent this could account for the current underestimation of basal energy expenses.

## In silico gene knockouts and mapping to in vivo essentiality

In addition to studying fluxes of the unperturbed model, the FBA framework also allows to study the impact of in silico gene disruptions by simulating knockouts in COBRApy (*Ebrahim et al., 2013*), that is by removing all reactions associated with a gene of interest from the model and calculating the growth rate from the resulting model. A knockout is defined to be lethal if the resulting growth rate is zero or the FBA problem becomes infeasible. By this definition, 123 of the 155 genes included in the model are essential (79%). In this analysis, two genes are currently non-essential in silico for 'technical' reasons: *metK/*0432 (methionine adenosyltransferase, MAT) and *mtnN/*0381 (*S*-adenosylhomocysteine (SAH) hydrolase, reaction ID: AHCi). These genes are part of the *S*-adenosylmethionine (SAM) pathway and would be connected through nucleic acid methylation reactions (consuming

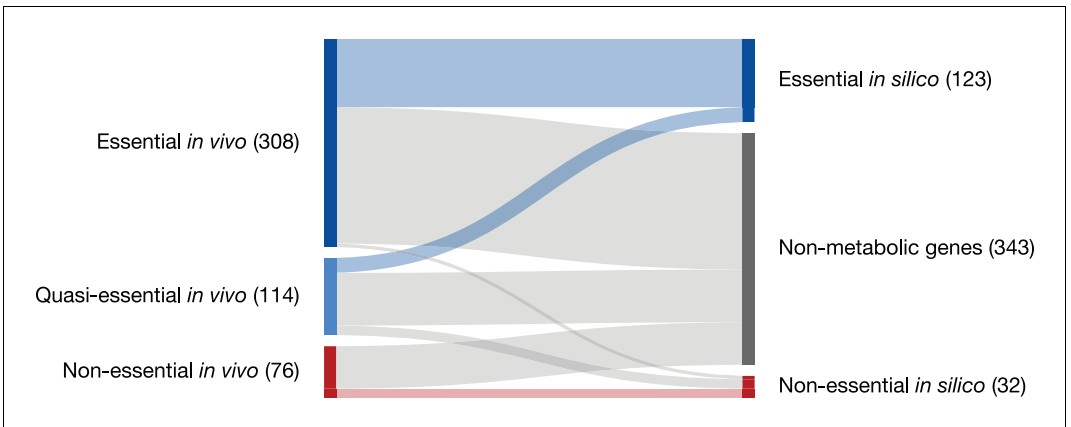

**Figure 16.** Partitioning of genes classified as essential, quasi-essential, and non-essential by transposon mutagenesis experiments into those which are in silico essential, in silico non-essential, and not modeled ('Non-metabolic'). All genes are included (i.e. also RNA genes and pseudogenes).

The online version of this article includes the following figure supplement(s) for figure 16:

**Figure supplement 1.** In silico double-gene knockouts between genes that are non-essential in single-gene knockouts.

SAM and producing SAH), which were not included in the model due to missing experimental information. As these two reactions currently cannot carry flux, it does not make sense to consider their in silico essentiality in the comparison to experiment.

An individual breakdown of in silico gene essentialities is provided in *Table 2*, which lists all genes modeled in silico, together with their catalyzed reaction (or general description for genes with several reactions, like the peptidases), and their essentiality in silico and in vivo. Genes non-essential in silico are found in amino acid, central and nucleotide metabolism as well as transport reactions, and only one gene in lipid metabolism. Some non-essentialities are functionally connected. For example, as the peptide importer Opp (*oppB*/0165 through *oppA*/0169) is non-essential in silico, the downstream peptidases have to be non-essential as well. A further analysis of in silico essentialities is featured in Sections 'Interpretation of individual gene essentialities', 'The role of folate metabolism', and 'A partial bypass to the pentose phosphate pathway'.

These in silico essentialities can be compared to the experimental transposon data (see Section 'Transposon mutagenesis experiments probe in vivo gene essentiality'). *Figure 16* shows an overall mapping between in silico and in vivo essentiality (including all genes in JCVI-syn3A, including RNA genes, pseudogenes and the two 'technical non-essentials'). A more detailed analysis for the genes included in the model is presented in *Table 4*, which displays the confusion matrix for the in silico to in vivo comparison, that is the distribution of model genes among the in silico and in vivo classifications. The left table represents the breakdown for all model genes except the two technical

**Table 4.** Confusion matrices for gene essentiality prediction.

'All genes' denotes agreement/disagreement between the model prediction and the transposon mutagenesis experiment considering all genes in the metabolic reconstruction (excluding the two 'technical non-essential' genes). 'Excluding AA' repeats the same comparison as 'All genes', with genes for amino acid utilization (uptake and peptidase genes) excluded. See *Table 2* for individual gene essentialities in silico and in vivo.

| Exp. / Model | All genes | | Excluding AA | |
|---|---|---|---|---|
| | **Essential** | **Non-essential** | **Essential** | **Non-essential** |
| **Essential** | 101 | 4 | 101 | 4 |
| **Quasi-essential** | 22 | 14 | 22 | 4 |
| **Non-essential** | 0 | 12 | 0 | 10 |

non-essentials, while the right table shows the breakdown if genes related to amino acid utilization are also excluded (see below). Whilst the model only distinguishes essential and non-essential genes, the experimental classification includes quasi-essentiality, which falls somewhere in between essentiality and non-essentiality. Thus, for any evaluation of predictive performance of the model, some assumption has to be made with regard to the in vivo quasi-essential genes. *Table 5* summarizes several statistics obtained for specific cases, discussed in the following.

Two limiting cases of interest are treating all quasi-essential genes as either in vivo essential (1) or in vivo non-essential (2). Given that the identification of quasi-essential genes was crucial for the successful genome minimization in JCVI-syn3.0 (*Hutchison et al., 2016a*), treating these genes as essential might be the biologically more relevant assumption. If all quasi-essential genes are considered essential (i.e. adding the numbers in the second row in *Table 4*, left matrix to the first row), the model displays an accuracy of 88%. (Accuracy = (TP+TN)/total; we opt to define essential genes as 'positive' and non-essential genes as 'negative', so that a true positive gene (TP) is essential in model and experiment; a false positive (FP) is essential in the model but non-essential in experiment; a true negative (TN) is non-essential in model and experiment; and a false negative (FN) is non-essential in the model but essential in experiment. 'Total' is the sum of all genes included in the analysis.) The resulting sensitivity (TP/(TP+FN)) is 87%, while the specificity (TN/(TN+FP)) is 100%: All in silico essential genes are at least quasi-essential in vivo, so there are no 'strong' false positive predictions (of genes to be essential that are actually non-essential in vivo). If, alternatively, all quasi-essential genes are considered non-essential in vivo (adding the numbers in the second row to the third row in the left confusion matrix in *Table 4*), the accuracy comes out a bit lower at 83%; the sensitivity increases to 96% while the specificity drops to 54%. This low specificity can be explained by considering the comparatively low number of in vivo non-essentials among the genes included in the model (12): Considering all quasi-essentials (two thirds of which are essential in the model) to be non-essential as well then leads to a large relative fraction of 'non-essentials' not detected by the model, even though the overall accuracy does not change much compared to case (1). As a more balanced measure of model prediction performance, *Table 5* also features the Matthews correlation coefficient (MCC) in the last column, which can range from −1.0 (perfect disagreement) via 0.0 (same agreement as a random model) to 1.0 (perfect agreement). For both cases described above (treating quasi-essentials as either all essential or all non-essential), the MCC comes out to ~0.59.

While this does not amount to perfect agreement, we note that the quasi-essentials in the middle row in *Table 4* (upper confusion matrix) actually encompass the vast majority of false model predictions. Thus, in addition to the two limiting cases presented above, it is also instructive to consider the prediction performance when including only those genes that can be classified as essential or non-essential in vivo, that is those genes that can be compared to the model classification without further assumptions. In this case, the specificity reaches 100% as in case (1) above, as there are again no false positives; the sensitivity reaches the same value as in case (2) above (96%) as there are only four false negatives; and the accuracy increases to 97% in this case. The MCC also comes out higher at 0.85. This demonstrates that the lower MCC and other metrics obtained before really arise from

**Table 5.** Accuracy, sensitivity, specificity and Matthews correlation coefficient calculated for several scenarios.

QE as E: Treating in vivo quasi-essentials as essentials; QE as NE: Treating quasi-essentials as non-essentials; No QE genes: Excluding all genes quasi-essential in vivo; QE as E, no AA genes: Excluding genes related to amino acid utilization, and treating quasi-essentials as essentials; QE as NE, no AA genes: Excluding genes related to amino acid utilization, and treating quasi-essentials as non-essentials.

|  | Accuracy | Sensitivity | Specificity | MCC |
|---|---|---|---|---|
| QE as E | 88% | 87% | 100% | 0.59 |
| QE as NE | 83% | 96% | 54% | 0.59 |
| No QE genes | 97% | 96% | 100% | 0.85 |
| QE as E, no AA gene | 94% | 94% | 100% | 0.72 |
| QE as NE, no AA genes | 82% | 96% | 39% | 0.46 |

the large number of quasi-essential genes included in the model, that are inherently difficult to describe in an FBA model: For example, nucleic acid stabilization by polyamines is a known essential process, and the minimal media for both *M. mycoides capri* LC Y (*Rodwell, 1969*) and *M. pneumoniae* (*Yus et al., 2009*) hence include spermine, which is thus a biomass component in the model for JCVI-syn3A. While this renders the corresponding uptake genes (*potC*/0195 through *potA*/0197) essential in the model, they are only quasi-essential in vivo (see *Table 2*).

Similarly, it is of interest to consider one set of genes whose functionality is difficult to capture precisely based on the currently available information, namely the genes pertaining to uptake and utilization of amino acids (in free or peptide form): As can be seen in *Table 2*, from the overall 14 'weak' false negative predictions (in silico non-essential genes that are quasi-essential in vivo), 10 comprise the peptide importer Opp (*oppB*/0165 through *oppA*/0169), two amino acid permeases (0878 and *gltP*/0886) and three of the four peptidases (0305, 0444 and 0479). As further discussed in Section 'Interpretation of individual gene essentialities', the in vivo essentiality of these genes is likely affected by their exact substrate profiles and maximal uptake rates. If all 12 genes related to amino acid utilization (i.e. the genes above plus *ietS*/0133 and 0876) are excluded from the prediction comparison, the right confusion matrix in *Table 4* is obtained. The resulting metrics are listed in the last two rows of *Table 5*, where the remaining quasi-essentials are again included in the in vivo essentials (upper row) or in the non-essentials (lower row). As can be seen, in the first case, the accuracy and sensitivity both rise to 94% compared to the full set of genes (88% and 87%, row one in *Table 5*); the MCC rises to 0.72. In the second case, the specificity drops from 54% to 39% compared to the full set of genes (row two in *Table 5*), and the MCC decreases to 0.46. However, this must be seen in the light of the excluded genes comprising mainly weak false negatives, that is quasi-essential genes that are non-essential in silico, and no weak false positives (quasi-essentials that are essential in silico). Thus, even though genes are excluded that show disagreement between model and experiment, the agreement worsens because these genes happened to be classified as 'true negatives' in case (2). The improved model metrics in case (1) for excluding amino acid genes thus seem more relevant.

In summary, this analysis demonstrates an overall good agreement between model and experiment, which is mainly impacted by the in vivo quasi-essential genes, whose essentiality is inherently difficult to capture in an FBA model. The disagreements observed (quasi-essential genes, and a few strong false negatives) are discussed in detail in Sections 'Interpretation of individual gene essentialities', 'The role of folate metabolism', and 'A partial bypass to the pentose phosphate pathway'. Some of them can be rationalized, while others lead to new hypotheses.

Finally, performing in silico double knockouts (*Figure 16—figure supplement 1*) yields just one synthetic lethality (i.e. lethality of a two-gene knockout where the individual knockouts are non-lethal)–namely, a double knockout of the two amino acid permeases 0876 and 0878, which prevents the cell from acquiring cysteine.

## Abundances of essential and non-essential proteins

Absolute cellular abundances (molecules per average cell) of JCVI-syn3A proteins were obtained from mass spectrometry based proteomics and the assumed protein dry mass fraction. Relative and absolute protein abundances were used in the reconstruction of the JCVI-syn3A biomass composition (see Section 'Biomass composition and reaction') and estimates of the $V_{\mathrm{max}}$ for reactions in the metabolic model. They also served for the further study of the JCVI-syn3A proteome, both with respect to the fraction of proteins with known functions, and in regard to expression of essential vs. non-essential proteins.

Comparing the overall JCVI-syn3A proteomics breakdown in *Figure 17a* to the genome breakdown in *Figure 1* shows that the 'Unclear' fraction is even smaller in the proteome than in the genome, suggesting that at least a generic function can be immediately assigned to >90% of the proteome. Furthermore, proteins classified as 'Metabolism' alone account for ~25% of the proteome. Considering all proteins included in the FBA model (i.e. also the synthetases classified as 'Genetic Information Processing') covers a subset of 40% of the proteome. Thus, studying expression features for genes in the model should yield relevant insights into the proteome as a whole.

*Figure 17b* compares distributions of absolute protein abundances between in silico essential, in silico non-essential and all proteins. *Figure 17c* shows the same comparison based on the transposon mutagenesis classification of essentiality (also including quasi-essential genes). As can be seen,

the expression profiles for essential and non-essential proteins are qualitatively similar both to each other and to the expression profile of all proteins in JCVI-syn3A. This holds for both the genome-wide transposon data-based comparison (*Figure 17c*) and the comparison for the subset of (mostly) metabolic FBA genes (*Figure 17b*). While this does not yet allow for strong conclusions, it does suggest the presence of little regulation, if at all, that would discriminate gene products based on their essentiality. This conclusion would be in line with the small number of identified regulatory proteins left in the genome of JCVI-syn3A.

## Discussion

### Completeness of the model

The creation of the first minimal bacterial cell JCVI-syn3.0 in 2016 provided a powerful platform for understanding the basics of life. As a first step along this road, we have combined the genetic information of JCVI-syn3A with the extensive amount of experimental information available for the natural *M. mycoides capri* and assembled a metabolic reconstruction and FBA model for the minimal cell. The majority of reactions in this model are supported by experimental evidence on the parent organism and related mycoplasmas. The model is near-complete with regard to accounting for the biomass components, describes cellular energy expenses well, shows good agreement with experimental transposon insertion data, and importantly has relatively few non-essential metabolic genes. It thus provides a foundation to study the features of the minimal metabolic network.

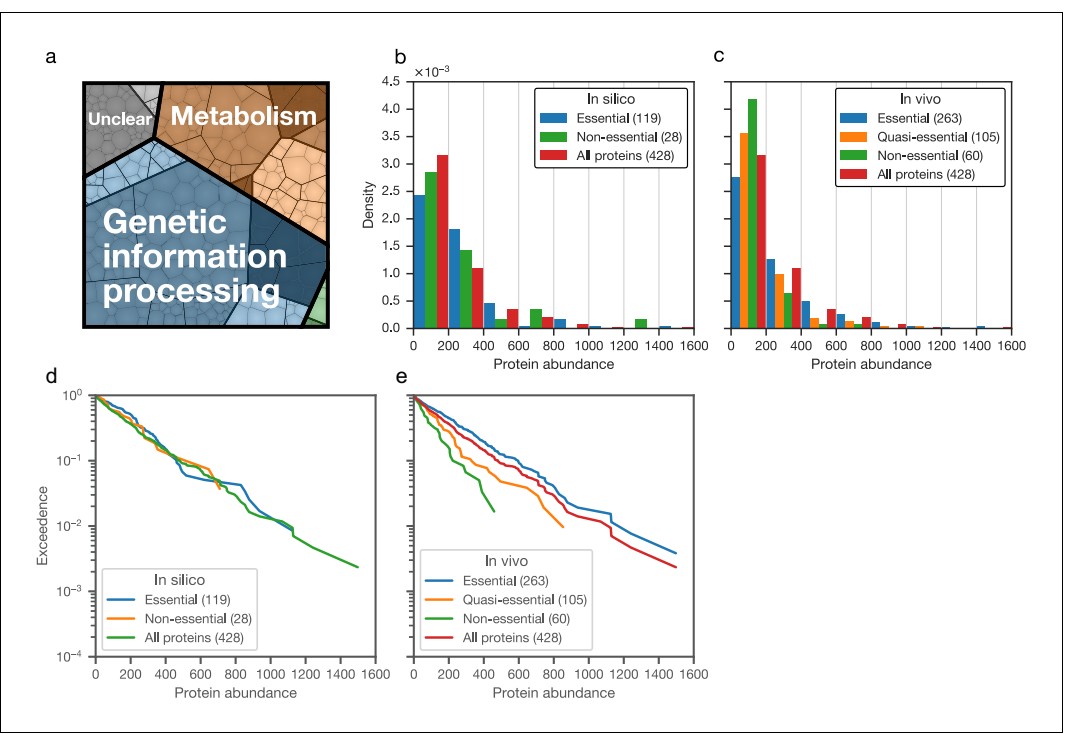

**Figure 17.** Distributions of absolute protein abundances (number of molecules per average cell) in JCVI-syn3A. (a) Breakdown of the JCVI-syn3A proteome into functional classes. The area of each cell is proportional to its relative abundance. (b,c) Histograms of absolute protein abundances. (b) Absolute abundances of model-included metabolic proteins essential or non-essential in silico compared to all protein abundances. 'Technical non-essential' proteins are not included (see Section 'In silico gene knockouts and mapping to in vivo essentiality'). (c) Absolute abundances for proteins classified by in vivo essentiality from transposon mutagenesis experiments. (d,e) Exceedence plots of absolute abundances for proteins classified by in silico or in vivo essentiality. The exceedence at a given protein abundance value x is the fraction of the protein set displaying an abundance higher than x. (d) Model-included proteins (classified by in silico essentiality) compared to all proteins. (e) Proteins classified by in vivo (transposon-based) essentiality.

The metabolic networks of lipids and cofactors are both functionally nearly minimal and in their reconstruction nearly complete. The reconstructed lipid network is consistent with all membrane components known from the biomass composition (save for the small fraction of triacylglycerol, which might or might not still be produced in JCVI-syn3A) and contains no redundant features (except for one more glycosyltransferase than required by the current reconstruction). The only remaining gap in lipid metabolism is the missing gene for phosphatidate phosphatase. In cofactor metabolism, the remaining questions are the substrate specificities of the EcfS transporter subunits, and the proposed lipoate uptake mechanism. Amongst the ion transport reactions, a gene for the $Na^+/H^+$ antiporter remains to be identified. Central and nucleotide metabolism display a number of potential redundancies (see *Table 2*), and several important reactions not accounted for by a gene yet. In central metabolism, these include the in silico essential transaldolase (TALA) reaction; and the reactions PDH_E1, NOX and export of lactate and acetate, all of which (except for lactate export) are required in the model to maintain the experimental doubling time of ~2 hr. In nucleotide metabolism, nucleobase uptake is an essential function still unaccounted for.

Even so, we obtain a number of gap-filled reactions of only 21—a fraction of 6% of all model reactions, or 9% of the 244 'non-pseudo' reactions. (Non-pseudo reactions are the subset of individual chemical or transport reactions in the model. This includes all model subsystems except for the artificial exchange and biomass reactions; and the macromolecule reactions, which describe non-metabolic processes taking place in all cells that are therefore not relevant for the number of gap fills). These 21 gap fills are obtained from the total number of 35 non-pseudo reactions without assigned gene after subtraction of 14 passive transport reactions assumed to take place without protein mediation (see e.g. discussion on passive permeative glycerol uptake in Appendix 1). This number of gap fills is considerably lower than in comparable models (see Section 'Comparison to *M. pneumoniae*'). Furthermore, from these 21 gap fills, only four are not supported by experimental evidence. Hence, 98% of all non-pseudo reactions are justified through gene assignments and/or experimental evidence, or are assumed to be passive. *Table 6* summarizes the overall features of the model.

**Table 6.** Summary of features of the metabolic model for JCVI-syn3A.
'Non-pseudo' reactions exclude exchange, biomass and macromolecule reactions. 'Annotation-supported' includes all non-pseudo reactions that have a gene assigned. 'Passive' reactions are transport processes assumed to take place without protein mediation. The meaning of 'technical' non-essential genes is explained in Section 'In silico gene knockouts and mapping to in vivo essentiality'.

| | | | |
|---|---|---|---|
| Model overview | Genes | 155 | |
| | Genome coverage | 31% | |
| | Metabolites | 304 | |
| | Reactions (total) | 338 | |
| | Reactions (non-pseudo) | 244 | |
| Reaction breakdown (% of non-pseudo) | Annotation-supported | 209 | 86% |
| | Passive | 14 | 6% |
| | Gap fills with exp. evidence | 17 | 7% |
| | Gap fills without exp. evidence | 4 | 2% |
| | Supported (annotation/exp./passive) | 240 | 98% |
| Essentiality *insilico* | Essential genes | 123 | 79% |
| | Non-essential genes | 30 | 19% |
| | 'Technical' non-essential genes | 2 | 1% |
| Essentiality *invivo* | Essential genes | 308 | 62% |
| | Quasi-essential genes | 114 | 23% |
| | Non-essential genes | 76 | 15% |

We also note that there are good candidates for many of the missing functions: The NOX reaction could conceivably be carried out by an oxidoreductase of unspecified function. Both 0029 and *fre/* 0302 code for putative oxidoreductases. The gene *fre/*0302 in particular has been suggested to be the missing NADH oxidase (*Danchin and Fang, 2016*) and might be a candidate for investigation. While no gene for transaldolase has thus far been identified in any mycoplasma, the alternative route proposed in *Vanyushkina et al. (2014)* would just require a phosphatase reaction, which could plausibly be carried out by one of a number of hydrolases in JCVI-syn3A of thus far unknown function. The same holds for further phosphatase reactions, including the phosphatidate phosphatase (PAPA) reaction in lipid metabolism and a number of hydrolase reactions in nucleotide metabolism. Substrate screening, informed by the metabolic reconstruction, might therefore be of interest for the hydrolases of unknown function. Finally, some of the reactions without assigned gene are transport processes (e.g. lactate/acetate export, nucleobase uptake). It stands to reason that these processes might be carried out by some of the many membrane proteins in JCVI-syn3A whose function could not be identified yet.

Our metabolic model of JCVI-syn3A thus features an overall quite complete metabolic network, and even though a small percentage of reactions could currently not yet be assigned to a gene, the presence of genes catalyzing these reactions is plausible and the majority of these reactions are supported by experimental evidence. We therefore believe that comparing our model to the experimental transposon mutagenesis data is informative. In the following we discuss the comparison of in silico and in vivo (transposon mutagenesis-based) essentiality. While a number of genes can be discussed individually (see Section 'Interpretation of individual gene essentialities'), two pathways need to be discussed as a whole: the folate cycle (see Section 'The role of folate metabolism') and the pentose phosphate pathway (see Section 'A partial bypass to the pentose phosphate pathway'). Overall, the analysis suggests a few new hypotheses and even yields suggestions for some genes or groups of genes that could still be removed from the genome of JCVI-syn3A to minimize the genome even further. In this way, it complements the transposon mutagenesis data that can only probe individual essentialities—and simultaneous knockouts prove challenging in experiment (see Section 'Transposon mutagenesis experiments probe in vivo gene essentiality').

## Interpretation of individual gene essentialities

The most remarkable observation in comparing individual gene essentialities are three in silico non-essential genes whose in vivo (quasi-)essentiality is challenging to rationalize, and is therefore mysterious. These are the last two enzymes of the acetate branch (phosphate acetyltransferase and acetate kinase, *pta/*0229 and *ackA/*0230) in central metabolism and dCMP deaminase (*dctD/*0515) in nucleotide metabolism. The former two are essential in vivo while the remaining pyruvate dehydrogenase subunits (*pdhC/*0227 and *pdhD/*0228) are not. The non-essentiality of *pdhC/D* implies that JCVI-syn3A can grow without acetate fermentation, as also predicted by the model. This however raises the question what other essential function phosphate acetyltransferase and acetate kinase perform. A conceivable possibility could be that knockout of *pta/*0229 would lead to sequestration of coenzyme A as acetyl-coenzyme A if pyruvate dehydrogenase itself was still active. This could impede any function that coenzyme A could have as a cellular redox buffer. Similarly, knocking out *ackA/*0230 with the remainder of the pathway still active could lead to accumulation of acetyl phosphate, whose capacity as a nonenzymatic protein acetylation agent (*Weinert et al., 2013*) might become deleterious for excess concentrations. dCMP deaminase converts dCMP to dUMP, which in the apparent absence of pyrimidine nucleoside phosphorylase is a dead end for which we hypothesize back-conversion to dCMP (see Section 'Nucleotide metabolism'). Thus, dCMP deaminase neither provides a relevant nucleotide nor does it seem to contribute to balancing cellular nucleotide pools; its quasi-essentiality therefore completely eludes rationalization. For all of these three enzymes, it might be interesting to reduce their expression and study the impact on the metabolome.

A number of other discrepancies suggest specific new biological hypotheses. 'Weak' false negatives (i.e. genes non-essential in silico but quasi-essential in vivo) occur in central, nucleotide, lipid and amino acid metabolism. In central metabolism, the assumed N-acetylglucosamine-6-phosphate deacetylase, 0493, is the only in vivo quasi-essential gene along the entire N-acetylmannosamine utilization pathway (with all other genes being non-essential in vivo), which strongly suggests some

other (or additional) function for this gene. Lactate dehydrogenase (*ldh/*0475) is essential in vivo but non-essential in silico—suggesting that the residual capacity of pyruvate dehydrogenase is not sufficient to sustain cell growth on its own, or that the truncated complex is non-functional entirely. Furthermore, it suggests that pyruvate cannot be secreted at the same rate as lactate and/or that the assumed residual NADH oxidase activity would not be high enough to regenerate NAD+ for the GAPD/PGK branch in lower glycolysis (assuming a limit to how much NADPH produced through the alternative GAPDP branch the cell can utilize). The idea of a lower pyruvate transport rate is supported by reports of *M. mycoides* oxidizing external pyruvate 2–3 times slower than lactate (*Miles et al., 1985*; *Shahram et al., 2009*). In nucleotide metabolism, the purine nucleoside phosphorylase (PNP, *punA/*0747) is quasi-essential in vivo, which suggests that either the nucleobase uptake capacities in JCVI-syn3A do not suffice to cover purine base demand; or that the flux through the pentose phosphate pathway on its own does not suffice to provide all required PRPP, and additional synthesis starting from R1P released by PNP is necessary (see also Section 'A partial bypass to the pentose phosphate pathway'). In lipid metabolism, the second glycosyltransferase, 0697, assumed to be redundant with *cps/*0114 for synthesis of Gal-DAG (and hence non-essential in silico), is quasi-essential in vivo—suggesting that either the activity of *cps/*0114 is not high enough to cover both Gal-DAG and capsule production, or that *cps/*0114 only catalyzes capsule production and is not involved in Gal-DAG synthesis at all. In amino acid metabolism, 10 genes related to amino acid uptake and utilization are non-essential in silico but quasi-essential in vivo: namely, all genes of the Opp peptide importer (*oppB/*0165 through *oppA/*0169), three of the peptidases (0305, 0444 and 0479), the glutamate/aspartate permease GltP (*gltP/*0886) and one of the amino acid permeases (0878). The quasi-essentiality of Opp (and, possibly functionally related, three peptidases) is consistent with *M. mycoides capri* LC Y requiring alanine in peptide form for optimal growth (*Rodwell, 1969*); the growth rate decreases a bit if instead only free alanine is provided. The quasi-essentiality of Opp suggests that this effect could be more pronounced in the SP4 media, which might not have free alanine at the 1 mM concentration used in *Rodwell (1969)*. The in vivo quasi-essentiality of at least one amino acid permease (0878) is plausible in light of peptide-incorporated cysteine being not easily utilized by mycoplasmas (*Yus et al., 2009*). In line with this, an in silico knockout of both amino acid permeases (0876 and 0878) is lethal. However, the in vivo quasi-essentiality of the glutamate/aspartate permease GltP is surprising in the light of *M. mycoides capri* LC Y reportedly not depending on these two amino acids being provided in the media (*Rodwell, 1969*). If GltP is knocked out in silico, the model instead acquires glutamic and aspartic acid in their peptide form through the peptide importer Opp. Additionally imposing and gradually decreasing an upper limit on Glu/Asp peptide uptake, however, also gradually decreases the in silico growth rate, demonstrating the model dependence on Glu/Asp uptake. (The proposed dUMP disposal reaction CTPSDUMP is switched off in this test, since it would otherwise allow for unrealistically high glutamate production.) The latter observation would be consistent with the in vivo quasi-essentiality of GltP if the Glu/Asp peptide uptake capability through Opp was truly limited. The ability of *M. mycoides capri* to grow without Glu/Asp must then depend on some functionality removed during minimization of the genome.

Three 'weak' false positives (essential in silico and quasi-essential in vivo) suggest relaxed substrate specificities or additional functionalities. In lipid metabolism, the quasi-essentiality of *fakB/*0617 suggests that the specificities of the two *fakB* genes might be relaxed and less complementary than in *S. aureus* (*Parsons et al., 2014*) (albeit the overlap of activity could not be sufficient to maintain a stable membrane composition in the long run). In nucleotide metabolism, cytidylate kinase (CYTK, *cmk/*0347) is only quasi-essential in vivo, but it is not apparent how the cell would produce dCTP in its absence, or deal with CMP (produced from RNA breakdown). The most obvious hypothesis would be relaxed substrate specificity of uridylate kinase (UMPK, *pyrH/*0537) to also act on CMP and dCMP (possibly with weaker activity), as is the case for the eukaryotic enzyme. Guanine phosphoribosyltransferase (GUAPRT, *hptA/*0216) is the only phosphoribosyltransferase that is only quasi-essential. This suggests either some small but not negligible guanosine kinase activity or some capability to import intact GMP. Unfortunately, *Neale et al. (1984)* only studied uptake of deoxynucleotides and CMP in *M. mycoides capri*, so that the possibility of GMP uptake seems conceivable but is not directly supported by experiment. However, it is noteworthy that GUAPRT is one of the few reactions whose $V_{\max}$ obtained from proteomics and turnover numbers is significantly smaller than the steady-state flux demanded by the FBA model (with a ratio of 0.07), and the only such reaction

within nucleotide metabolism (see Section 'Steady state fluxes'); this lends further support to the hypothesis of additional routes to GMP.

The remaining discrepancies of individual genes can be rationalized by genetic context or biological interpretation. In central metabolism, the NADPH-producing GapN (*gapdh/*0451) is non-essential in the model (with RNDR as the only consumer of NADPH) but essential in vivo. As discussed in Section 'Energy usage', there are most likely other cellular NADPH demands not captured yet by the model, which would explain the in vivo essentiality. In nucleotide and lipid metabolism, the genes *nrdE/*0771 and *nrdF/*0773 (subunits of RNDR) and *pgpA/*0214 (PGPP) are essential in the FBA model but only quasi-essential in the transposon data. However, the RNDR maintenance gene *nrdI* (*nrdI/*0772) is essential in vivo, in line with the in silico essentiality of RNDR. Similarly, the gene downstream from *pgpA* in the lipid synthesis pathway, cardiolipin synthase (*clsA/*0147), is essential, just as the entire pathway is in silico. This suggests that *nrdE/F* and *pgpA* might also be essential but not identified as such by the transposon mutagenesis analysis, possibly because their knockouts do not become lethal immediately (see Section 'Transposon mutagenesis experiments probe in vivo gene essentiality'). In Appendix 1, we provide a possible rationalization for these genes in terms of enzyme dilution and required fluxes. Finally, the genes for spermine uptake (*potC/*0195 through *potA/*0197), $3',5'$-adenosine bisphosphate breakdown (*ytqI/*0139) and the peptidase Lon (*lon/*0394) are all quasi-essential in vivo, suggesting that lack of nucleic acid stabilization by spermine, lack of protein turnover or buildup of $3',5'$-adenosine bisphosphate are not detrimental immediately—a circumstance not captured by the steady-state FBA model. This could be another possible example of time-delayed lethality upon knocking out an essential gene—where the detrimental effect might take time to manifest even if all protein flux were to cease quickly.

In addition to these discrepancies, we note a total of 12 genes non-essential both in silico and in vivo. These are strong candidates for removal attempts, observing that synthetic lethalities must be avoided (see also Section 'Targeted gene removal experiments').

## The role of folate metabolism

While all folate-related genes (the putative uptake gene *folT/*0822, the folylpolyglutamate synthase *folC/*0823 and all genes of the folate cycle) in cofactor metabolism are essential in silico, they are only quasi-essential in vivo, with the exception of the gene for 5-formyl-THF cyclo-ligase (*ygfA/*0443, which is essential in vivo). The in vivo essentiality of YgfA could mean inhibitory actions of YgfA's substrate 5-formyl-THF outside folate metabolism (see Appendix 1). The in silico essentiality of all folate genes arises *per construction* in the model since formylation of Met-tRNA$_{fMet}$ for translation is assumed, and a THF derivative is included in the biomass.

Intriguingly, however, *Neale et al. (1981)* demonstrated the ability of *M. mycoides capri* to omit Met-tRNA$_{fMet}$ formylation, and to initiate protein synthesis with the unformylated species, in the absence of folate in the media–without significant change in doubling time. This raises the question if there is a way to remove the folate-related genes in JCVI-syn3A. The discrepancy between *M. mycoides capri*, which can grow without folate, and JCVI-syn3A, where folate genes are quasi-essential, might arise out of the different experimental setups: The classification of genes in our transposon mutagenesis experiments is based on outgrowth of slow-growers in a competition experiment, whereas (*Neale et al., 1981*) necessarily could only study the impact of folate free media on the entire culture. Probing the fitness of a genotype by competition experiments is known to be more sensitive than mere growth rate measurements, as has for example been observed in studies of rRNA modification enzyme knockouts in *E. coli* (*Persaud et al., 2010*; *Burakovsky et al., 2012*): While the knockouts affected the log-phase growth rate little to none, they did lead to the mutants being outgrown by the wild type in competition experiments to different degrees. If a similar situation were the case for folate usage in *M. mycoides capri* as well, it could mean that folate non-utilization yields a fitness disadvantage that was not amenable to detection in *Neale et al. (1981)*.

Alternatively, the removal of some translation-related genes, like RNA modification enzymes, during genome minimization might have made the cell more susceptible to further interference with translation by disrupting folate-related genes, preventing both Met-tRNA$_{fMet}$ formylation and tRNA wobble position uridine modification (through MnmEG: *mnmE/*0081, *mnmG/*0885). (JCVI-syn3A also contains the folate-dependent 23S rRNA modification gene *RlmFO/*0434, which is non-essential in the transposon data however.) It is generally known that translational mistakes, such as those

introduced by enzyme knockouts, can sum up and become detrimental once a certain threshold is crossed. In the genome of JCVI-syn3A, there already seems to be a precedent for this threshold scenario: Thiouridine tRNA modification through IscS and MnmA (*iscS/*0441 and *mnmA/*0387) appears to be non-essential in *M. mycoides capri* LC Y, which can grow without a precursor for the IscS cofactor pyridoxal phosphate (*Rodwell, 1969*). However, the transposon mutagenesis data for JCVI-syn3A indicate both genes to be essential, which must hence be due to the removal of other genes. Thus, if the genes could be identified whose removal similarly rendered folate usage quasi-essential, reintroducing them might enable one to delete folate metabolism in JCVI-syn3A entirely. A possible candidate for such a gene might be the deleted *rmsB/*MMSYN1_0204. RmsB catalyzes the methylation of 16S rRNA C967 using *S*-adenosylmethionine and plays a role in translation initiation in *E. coli*—both in binding of tRNA$_{fMet}$ (*Burakovsky et al., 2012*) and in fidelity of initiation (*Arora et al., 2013*). Removal of *rmsB* in JCVI-syn3A might thus have exacerbated the effects of further translation perturbations by non-formylation of Met-tRNA$_{fMet}$ resulting from folate gene knockouts. This might thus be an example of different routes of minimization yielding different minimal genomes.

## A partial bypass to the pentose phosphate pathway

While all genes from the pentose phosphate pathway are essential in silico, they are only quasi-essential in vivo—except for the final gene, *prs/*0831 (PRPPS), which converts ribose 5-phosphate (R5P) to the pathway end product PRPP. This is noteworthy because with *deoB/*0733 (PGMT/PPM), there is an alternative route to R5P, bypassing the rest of the pentose phosphate pathway by providing pentose sugars from nucleoside breakdown. We note that PRPP itself is still needed, as is also evidenced by the essentiality of its utilizing enzymes encoded by the genes *apt/*0413 (ADPT), *upp/*0798 (UPPRT), and 0164 (NCTPPRT) in nucleotide and cofactor metabolism. However, the fact that the remainder of the pathway is only quasi-essential in vivo suggests that the demand on R5P could partially be covered by R1P from purine nucleoside phosphorylase (PNP, *punA/*0747; catalyzing reactions PUNP1–4).

For every equivalent of R1P released, PNP will also yield as the other product an equivalent of free purine base, which, in order to be utilized, would consume again the equivalent of PRPP generated from the R1P released. In this way, PNP would allow to exactly meet the PRPP demand for AMP and GMP synthesis; in the absence of pyrimidine nucleoside phosphorylase in JCVI-syn3A however (see Section 'Nucleotide metabolism'), the PRPP demand of uracil would still not be met (and neither would the much smaller demand of nicotinate ribonucleotide synthesis). For PNP alone to cover the R1P/PRPP demand, it would need to degrade more purine nucleosides and release purine bases than can be used in the cell, and the cell would have to secrete purine bases at the same rate as uracil is taken up and phosphoribosylated.

This conclusion is supported by an in silico test: Setting the base symport reactions in the model to reversible for testing purposes indeed renders Tkt, Rpi and Rpe non-essential in silico—but not Prs. For example, upon knocking out the first reaction of the pathway (TKT2), we observe that the flux through PRPPS equals that through PPM (providing R5P), which in turn equals the sum of the fluxes through PUNP1 and PUNP3 (providing R1P from nucleoside breakdown). Accordingly, PNP (*punA/*0747) becomes essential in silico under these circumstances. Furthermore, the model secretes about as much purine bases as it takes up uracil (nicotinate uptake accounting for the very small difference).

It is unclear how biologically realistic this scenario is, but proton-symport coupled secretion of bases released from nucleosides has been suggested for mycoplasmas, and has even been proposed to contribute to the transmembrane proton gradient (*Maniloff, 1992*). If such a nucleobase export capability existed but did not enable purine base secretion at the same rate as uracil is taken up at optimal growth rate, this would explain both why the pentose phosphate pathway is not fully essential in vivo, and why it is nonetheless still quasi-essential, as the R1P bypass would still not suffice to cover PRPP demands.

Except for the above test, we keep base uptake irreversible in the model as the biological feasibility of secretion is not known and cannot be easily implemented in the model. Specifically, it allows for feeding of excess (d)R1P into glycolysis, which renders PGI and the PTS system non-essential in silico even if fluxes through the pentose phosphate pathway and DRPA are constrained with the

$V_{max}$ values derived from proteomics and $k_{cat}$ values (see Section 'Steady state fluxes'). For testing purposes, the above test with knocking out TKT2 therefore also had the DRPA flux set to zero.

## Targeted gene removal experiments

The 12 true negatives from the essentiality comparison (genes non-essential both in silico and in vivo) are strong candidates for attempts to remove more genes, and hence minimize the genome of JCVI-syn3A even further. Suggested gene removal experiments are listed in *Table 7* and are discussed in the following. Except for *folT*/0822 in the last two suggested experiments, all listed genes can be removed simultaneously in silico, as any interdependencies between them do not pertain to essential functions. The resulting in silico doubling time is 3.2 hr, that is the same doubling time observed by knocking out any single gene along the acetate branch.

All the true negatives are either individual genes or belong to short pathways. The genes *manA*/0435 (mannose 6-phosphate isomerase) and *deoC*/0732 (deoxyribose phosphate aldolase) could be removed individually. The whole acetylmannosamine branch (*nanE*/0494, 0495 and *nagB*/0726) could be removed, with the exception of the proposed N-acetylglucosamine 6-phosphate deacetylase 0493, which per the transposon essentiality data seems to have some other/additional function. The remaining subunits of pyruvate dehydrogenase (*pdhC*/0227 and *pdhD*/0228) and the proposed lipoylpeptide importer 0401 could also be removed together per our reconstruction and transposon data. The remaining individual functions include the two deoxyadenosine kinases (*dak1*/0330 and *dak2*/0382), a peptidase (*ietS*/0133) and an amino acid permease (0876). The experimental non-essentiality of the peptidase and the amino acid permease supports the model assumption of broad substrate profiles and hence some redundancy among peptidases and amino acid permeases. The individual non-essentiality of the two deoxynucleoside kinases is also consistent with the assumed overlapping substrate profiles (see Appendix 1). However, it is not known whether the ribodinucleotide reductase (RNDR, *nrdE*/0771 through *nrdF*/0773) on its own could supply all dNTPs if both deoxyadenosine kinases were removed. Thus, simultaneous removal of *dak1*/0330 and *dak2*/0382 might incur a synthetic lethality or growth defect.

Furthermore, the observed quasi-essentiality of folate metabolism, in conjunction with the experimental observations of folate-free growth (*Rodwell, 1969*; *Neale et al., 1981*), raises the question if even this subsystem could be removed somehow. If this were the case, it would allow deleting a number of genes at once: On top of the five quasi-essential folate genes, these would also include the 5-formyl cycloligase, *ygfA*/0443, as the metabolic source of 5-formyl-THF would be gone; the remaining THF-dependent RNA modification enzymes (MnmEG: *mnmE*/0081, *mnmG*/0885; and RlmFO/0434); and the peptide deformylase *def*/0201, as nascent peptides would not be formylated anymore to begin with. Thus, a removal of 10 genes might be possible if JCVI-syn3A could be shown to grow without folate usage, or be enabled to do so (e.g. by reintroducing translation-related gene(s) whose removal might have raised the importance of Met-tRNA$_{fMet}$ formylation; see Section 'The role of folate metabolism').

**Table 7.** List of suggested gene removal experiments.

| Gene(s) | Description | Remark |
|---|---|---|
| *manA*/0435 | Mannose 6-phosphate isomerase | |
| *deoC*/0732 | Deoxyribose phosphate aldolase | |
| *nanE*/0494, 0495, *nagB*/0726 | N-acetylmannosamine 6-phosphate branch | |
| *pdhC*/0227, *pdhD*/0228, 0401 | pdhCD and proposed lipoate importer | |
| *dak1*/0330, *dak2*/0382 | Deoxynucleoside kinases | Synthetic lethality possible |
| *ietS*/0133 | Peptidase | |
| 0876 | Amino acid permease | |
| *folT* in JCVI-syn3A | Folate uptake and usage | Competition experiment with wild type to probe fitness cost in JCVI-syn3A |
| *folT* in JCVI-syn1.0 | Folate uptake and usage | Competition experiment with wild type to probe fitness cost in JCVI-syn1.0 |

To probe the fitness cost of folate non-usage in JCVI-syn3A more specifically and study whether this fitness cost is affected by the genome minimization, it would be of interest to carry out a specific competition study of a *folT* knockout vs. the wild type for both JCVI-syn3A and JCVI-syn1.0. If folate non-usage only occurs a significant fitness cost in JCVI-syn3A but not in JCVI-syn1.0, this would imply that JCVI-syn3A could be re-enabled to grow without folate as well.

## Suggestions for further experimental study

As presented in the preceding sections, the comparison between in silico and in vivo essentiality yielded a number of hypotheses and suggested several possible gene removal experiments (*Table 7*). Similarly, the metabolic reconstruction itself yielded a number of informed hypotheses, as well as raised specific questions. While the minimal genome has been experimentally obtained, understanding all genetic functions both individually and as a system remains an ongoing challenge. Thus, the hypotheses and questions raised in this work provide invaluable help in the ongoing effort to completely understand the minimal genome. In *Table 8*, we provide a list of suggested experiments other than gene removal/knockout studies, sorted by category and providing a rationale for each experiment.

## Comparison to *M. pneumoniae*

*M. pneumoniae* is an important systems biology model organism that has been extensively studied (*Güell et al., 2009*; *Kühner et al., 2009*; *Yus et al., 2009*; *Maier et al., 2011*; *Wodke et al., 2013*) so a comparison to its metabolic map should be of interest. With a published metabolic reconstruction (iJW145 (*Wodke et al., 2013*)) that includes 304 reactions involving the products of 145 genes it is similar in size to the metabolic reconstruction of the minimal cell JCVI-syn3A with its 338 reactions and 155 genes. Utilizing the vast experimental information on *M. mycoides capri*, the natural precursor of JCVI-syn3A, as well as information on JCVI-syn3A homologs in other organisms, enabled us to obtain a smaller percentage of gap fills (i.e. model reactions assumed to be enzymatic yet having no gene assigned) of 6% out of all model reactions, compared to 25% in the *M. pneumoniae* model iJW145; or 9% for JCVI-syn3A vs. 32%, if exchange, macromolecular and biomass reactions are excluded from the total number of reactions in each model. The JCVI-syn3A model yields a higher degree of in silico essentiality (79% vs. 56% for the 131 'metabolic proteins' in the *M. pneumoniae* model (*Wodke et al., 2013*))—reflecting the minimization of the JCVI-syn3A genome. This higher degree of essentiality is also reflected in the differences in individual reactions presented in *Supplementary file 6* (see also Section 'Materials and methods' for details on the model comparison).

Excluding exchange, macromolecular and biomass reactions, a core of 126 reactions is shared between the models, including glycolysis, the pentose phosphate pathway, reactions from nucleotide, cofactor and lipid salvage pathways, and tRNA charging. However, *M. pneumoniae* has 116 reactions not present in JCVI-syn3A, which mainly includes uptake and utilization of additional sugar sources, further nucleotide conversions, more extensive cofactor salvage reactions, and additional lipid-related reactions. Some of these reactions were present in JCVI-syn1.0 but were removed during minimization of the genome to JCVI-syn3A. Furthermore, some of the differences are technical in nature, for example the choice to model amino acid uptake as ABC import reactions in *M. pneumoniae*, or the decision to include amino acid secretion reactions there. Interestingly, in spite of the much smaller genome of JCVI-syn3A, its reconstruction still contains 120 reactions not present in the *M. pneumoniae* model. While a number of these arise from a more detailed description of various transport processes (nucleosides, peptides and ions), we note the presence of some functionalities not present or known in *M. pneumoniae*. These include the production of a polysaccharide capsule (in addition to the monogalactosyl-lipid), some alternative sugar sources specific to JCVI-syn3A and also specific nucleotide conversion and breakdown reactions, perhaps most notably the presence of the essential damage preemption enzyme dUTPase (*dut/*0447, see *Figure 8*).

## Conclusion

We have presented a comprehensive metabolic reconstruction and FBA model of the minimal cell JCVI-syn3A, informed by the extensive experimental information available for the natural precursor, *M. mycoides capri*, in vivo transposon mutagenesis and proteomics data. The metabolic model is

**Table 8.** List of suggested experiments, with rationale behind each suggestion.

| Experiment | Rationale |
|---|---|
| *Nutrient utilization* | |
| Detect lipoylpeptide uptake | Lipoate is cofactor for PDH, whose functionality is unclear after E1 subunit deletion. |
| Detect nucleotide uptake | Activity reported for *M. mycoides capri* without gene assignment; alternative routes present in JCVI-syn3A, but activity not ruled out. |
| Detect nucleobase uptake | Activity reported for *M. mycoides capri*, and uracil uptake essential in model. |
| Demonstrate growth on thiamine diphosphate | Structural data and deletion of thiamine diphosphokinase suggest thiamine diphosphate (ThDP) uptake. Inability to grow on ThDP would imply unidentified kinase activity. |
| Demonstrate growth on pyridoxal phosphate | With no pyridoxal kinase identified, growth on pyridoxal phosphate (P5P) assumed; inability to grow on P5P would imply unidentified kinase activity. |
| Demonstrate growth on acetylmannosamine | Reconstruction suggests operon 0493 through 0495 to be *nagA*/*nanE*/*nagC*; growth on acetylmannosamine would support assignment and imply uptake capability. |
| Demonstrate growth on mannose or glucosamine | Literature suggests mannose and glucosamine import through PtsG; and downstream enzymes are present. |
| *Metabolite production/secretion* | |
| Detect production of acetate | Acetate pathway has been partially removed, but several of the remaining enzymes remain essential in transposon mutagenesis experiments. |
| Investigate production of lipogalactan capsule | Genetic evidence suggests capsule is still being produced. |
| Detect secretion of deoxyuridine or dUMP | Deoxyuridine/dUMP is currently dead-end; secretion would be one possible solution. |
| *Enzymatic activity* | |
| Detect pyruvate oxidation | Conversion of pyruvate to acetyl-CoA by truncated PDH complex has been assumed for the time being but is not supported by experiment. |
| Detect oxidation of acetaldehyde | Conversion of acetaldehyde to acetyl-CoA would provide alternative explanation for presence of truncated PDH complex in JCVI-syn3A in spite of deletion of first subunit. |
| Detect NOX activity | NADH oxidase (NOX) has been deleted but activity would be necessary for PDH activity against both pyruvate and acetaldehyde. |
| Detect transaldolase activity | Activity is essential in model and has been detected in *M. mycoides capri*; no known gene in any mycoplasma though. |
| Detect sedoheptulose-1,7-bisphosphate phosphatase activity | Reaction would provide bypass to transaldolase reaction. |
| Detect phosphatidate phosphatase activity | Gene present in *M. pneumoniae* and reaction is missing link to diacylglycerol, but no gene identified in JCVI-syn3A. |
| Assess phosphatase activity against dCTP, dCDP, GMP, dAMP, dGMP, dUMP, dTMP; pyrophosphatase activity against CTP, dCTP | Activities observed in *M. mycoides capri* but no gene identified in JCVI-syn3A. |
| Detect deoxyuridine phosphorylase activity | Gene has been removed in JCVI-syn3A (MMSYN1_0734), but deoxyuridine/dUMP is currently dead-end, raising the question whether function is carried out by some other gene. |
| Detect thymidylate synthase activity | Extremely low activity detected in *M. mycoides mycoides* SC, but no gene identified. Reaction would be alternative solution to deoxyuridine/dUMP dead-end. |
| *Specific gene function* | |
| Determine substrates for deoxynucleoside kinase *dak2*/0382 | Presence of *dak2*/0382 in addition to the characterized *dak1*/0330 suggests different substrate profile. |
| Verify CTPS activity against dUMP | CTPS (*pyrG*/0129) converting dUMP to dCMP seems most plausible solution to deoxyuridine/dUMP dead-end. |
| Check PGPP activity against phosphatidate | Activity observed for PgpB in *E. coli*; activity for PGPP (*pgpA*/0214) would provide missing link to diacylglycerol in apparent absence of phosphatidate phosphatase gene. |
| Check UMPK activity against CMP and dCMP | Substrate profile for UMP kinase similar to eukaryotic enzyme could explain quasi-essentiality of CMP kinase. |
| *Change of expression levels* | |

*Table 8 continued on next page*

*Table 8 continued*

| Experiment | Rationale |
|---|---|
| Knock out deoxynucleoside kinases and over express RNDR | RNDR and deoxynucleoside kinases provide redundant routes to deoxydinucleotides, suggesting one pathway might be sufficient if expression increased. |
| Reduce expression of RNDR and knock out putative dUTPase simultaneously | RNDR is currently only known source of dUTP. If RNDR knockdown would make the putative dUTPase gene *dut/*0447 nonessential as well, this would corroborate the putative assignment. |

near complete with regard to accounting for all biomass components, with known metabolic functions not included mainly pertaining to damage repair/pre-emption and RNA modification. The high quality of the model is exemplified by the strong support for the network, with 98% of enzymatic reactions in the model justified through gene assignments and/or experimental evidence; and by its good agreement with experimental transposon mutagenesis data showing 92% of the genes included in the model to be essential or quasi-essential. The essential metabolism of this minimal cell consists of only a few subsystems that are only minimally connected with each other. The subsystems for lipids, amino acids, nucleotides and cofactors contain only salvage pathways. An energy analysis shows how this reliance on salvage pathways enables the cell to only spend 9% of its produced ATP on precursor transport and processing while maintaining a doubling time of 2 hr. The experimental transposon mutagenesis data probe individual gene essentialities, which together with the metabolic model point to a few possible remaining redundancies. Comparison with *M. mycoides capri* further suggests that folate metabolism only became quasi-essential by removal of other genes, underlining how different routes of genome minimization could yield different minimal genomes. Model and accompanying experimental data thus not only reveal properties of the minimal metabolic network, but also yield an extensive list of suggested experiments to test the resulting hypotheses. The model, together with the accompanying transposon mutagenesis and proteomics data, provides an excellent foundation for further studies of the minimal cell.

# Materials and methods

## Key resources table

| Reagent type (species) or resource | Designation | Source or reference | Identifiers | Additional information |
|---|---|---|---|---|
| Strain, strain background | JCVI-syn3A | JCVI; this article | GenBank accession number: CP016816.2 | [1] |
| Strain, strain background | JCVI-syn1.0 | doi: 10.1126/science.1190719 | GenBank accession number: CP002027.1 | [1] |
| Genetic reagent | terTufPuro transposome | doi: 10.1126/science.aad6253 | | Constructed by the JCVI |
| Genetic reagent | Yeast tRNA | Life Technologies, Carlsbad, CA, USA | 15-401-029 | |
| Genetic reagent | EZ-Tn5-Transposase | Lucigen, Madison, WI, USA | TNP92110 | |
| Sequence-based reagent | Custom forward primer | Integrated DNA technologies, San Diego, CA, USA | | [2] |
| Commercial assay or kit | Nextera XT DNA library preparation kit | Illumina, San Diego, CA, USA | FC-131-1024 | |
| Chemical compound, drug | Quant-iT PicoGreen | Molecular Probes, Eugene, OR, USA | P7589 | |
| Chemical compound, drug | Puromycin | Molecular Probes, Eugene, OR, USA | A1113802 | |
| Software, algorithm | COBRApy | doi: 10.1186/1752-0509-7-74 | | |
| Software, algorithm | CLC Genomics Workbench | QIAGEN Bioinformatics, Redwood City, CA, USA | | |

*Continued on next page*

*Continued*

| Reagent type (species) or resource | Designation | Source or reference | Identifiers | Additional information |
|---|---|---|---|---|
| Software, algorithm | Proteome Discoverer 2.1.0.81 | Thermo Fisher Scientific | | |

[1] Bacterial strains JCVI-syn3A and JCVI-syn1.0 will be made available to qualified researchers by the JCVI and Synthetic Genomics, Inc under a material transfer agreement. Note that United States scientists must obtain a United States Veterinary Permit for Importation and Transportation of Controlled Materials and Organisms and Vectors from the U.S. Department of Agriculture Animal and Plant Health Inspection Service. The organisms require Biosafety Level 2 containment.

[2] Used for marker-specific sequencing with PCR; sequence under 'Tn5 mutagenesis–Experimental method'.

## Model construction

A genome-scale FBA model requires the reconstruction of the network of metabolic reactions, the assembly of the cellular biomass composition and necessary reaction constraints (e.g. substrate uptake and ATP consumption). The biomass composition of JCVI-syn3A was assembled based on experimental information available for *Mycoplasma mycoides capri* (in a few instances using information from other organisms). The reconstruction of the metabolic network began with the curated annotation published for JCVI-syn3.0 (*Hutchison et al., 2016a*) (which also contained annotations for all genes removed from JCVI-syn1.0). As done in other models (*Suthers et al., 2009*), an existing curated model was used as a reference to construct a first draft reconstruction. Initially, an FBA model for *M. pneumoniae* (*Wodke et al., 2013*) was used, keeping all reactions whose enzymes had an equivalent in JCVI-syn3A. Information from MetaCyc (*Caspi et al., 2008*), KEGG (*Kanehisa et al., 2015*), and an extensive evaluation of primary literature was then used to add reactions for the remaining metabolism-related genes in JCVI-syn3A, as well as reactions without a gene but supported by experimental evidence (including the assembled biomass composition). Experimental evidence was also used to exclude certain candidate reactions. Finally, a few reactions were added as gap-fills to complete the respective pathways. Metabolite and reaction IDs were matched to BiGG IDs (*Schellenberger et al., 2010*; *King et al., 2016*) when possible, otherwise IDs akin to BiGG IDs were assigned. Additionally, KEGG compound IDs were assigned to metabolites using the KEGG API; and InChI keys were assigned using the API for the Chemical Translation Service (*Wohlgemuth et al., 2010*).

Flux constraints for certain reactions were based on in vivo measurements, other models or physicochemical parameters. Reaction reversibilities were based on information from MetaCyc (*Caspi et al., 2008*) and eQuilibrator (*Noor et al., 2013*), inferred by analogy (e.g., fatty acid kinase was set as reversible like acetate kinase) or determined from biochemical context (e.g., $H^+$ diffusive influx is set to irreversible, in accordance with in vivo flow direction).

## Flux-balance analysis

Model assembly and flux-balance analysis (*Orth et al., 2010*) were carried out in COBRApy (*Ebrahim et al., 2013*), a Python module for constraint-based modeling. In flux-balance analysis, a system of $n$ reaction equations featuring in total $m$ reactants is represented as a stoichiometric matrix $S$ of dimensions $m \times n$, where the element $S_{ij}$ denotes the stoichiometric coefficient of reactant $i$ in reaction $j$ (negative for reactants, positive for products). A given set of fluxes through each reaction in the system is represented as a flux vector $\vec{v}$ of length $n$. Any steady-state flux vector then belongs to the solution space of the equation $S \cdot \vec{v} = \vec{0}$. This solution space is further constrained by any other constraints imposed on individual fluxes of the form $V_{\min,j} < v_j < V_{\max,j}$. A default upper bound $V_{\max}$ of 1000 mmol gDW$^{-1}$ h$^{-1}$ was used for all reactions and default lower bounds of $-1000$ mmol gDW$^{-1}$ h$^{-1}$ and 0 mmol gDW$^{-1}$ h$^{-1}$ were used for reversible and irreversible reactions, respectively. Specific constraints were chosen to account for uptake, secretion and ATP consumption restrictions. An optimal flux vector or set of flux vectors within the constrained solution space is then found by maximizing a particular objective function by means of linear programming. We picked biomass production as our objective function, so that the optimal flux vector describes the optimal growth under the chosen constraints.

As the solution to the flux optimization may not be unique, parsimonious FBA (pFBA) (*Lewis et al., 2010*) is employed to obtain a unique solution. In pFBA, the optimal growth rate

obtained by using the original objective function (biomass production in our case) is subsequently set as a constraint and a new objective function is defined with a coefficient of $-1$ for all reactions not part of the original objective function. Optimizing the flux vector under this objective function then yields the solution with the smallest sum of individual fluxes, corresponding to minimal enzyme usage in a biological context. Reversible reactions are split into two irreversible reactions for this purpose so as to avoid negative fluxes being maximized rather than minimized.

## Calculation of energy usage by subsystem

In order to analyze the energy consumption in the metabolic model for JCVI-syn3A, the consumption of ATP equivalents per subsystem was calculated. The term 'ATP equivalent' is used to account for the fact that phosphorylation of all dinucleotides in JCVI-syn3A is assumed to be carried out by the glycolytic enzymes phosphoglycerate kinase and pyruvate kinase, so that the phosphate donors are 1,3-diphosphoglycerate (1,3-DPG) and phosphoenolpyruvate (PEP) instead of ATP (whose role in dinucleotide phosphorylation is effectively bypassed). For all model reactions not involving 1,3-DPG or PEP, the production or consumption of ATP equivalents was calculated from the number of phosphate bonds formed or broken in each reaction producing or consuming ATP multiplied by the flux through that reaction in the FBA solution. Interconversion of ATP and ADP produces/consumes one phosphate bond. Hydrolysis of ATP to AMP (e.g. in tRNA charging) was counted as consuming two phosphate bonds, since the free pyrophosphate can only be hydrolyzed further to two individual phosphates. Consumption of the ATP moiety as a whole (e.g. in NAD+ synthesis) was also counted as consuming two phosphate bonds, accounting for the phosphorylation steps from AMP to ATP; the energy spent in AMP is already accounted for in other reactions (nucleoside uptake and PRPP synthesis for adenine phosphoribosylation). The flux through adenylate kinase (ADK1) phosphorylating AMP to ADP is already accounted for by counting ATP–>AMP hydrolysis as two phosphate bonds; it is thus ignored to avoid double-counting. To properly account for 1,3-DPG and PEP as phosphate donors for trinucleotide production, the fluxes for the ATP-producing PGK and PYK reactions were set equal to the sum of all PGK or PYK fluxes, respectively, in order to obtain the total number of ATP equivalents produced. In turn, the PGK and PYK model reactions phosphorylating dinucleotides other than ADP were counted as consuming one ATP equivalent each. Accuracy and correct accounting of the calculated ATP equivalent creation and consumption fluxes were verified by confirming that all individual fluxes thus calculated added up to zero.

To obtain the total ATP equivalent consumption percentage per category in *Table 3*, the consumption fluxes for all reactions in a given category were added up and normalized by the total ATP equivalent consumption flux in the model. (Central metabolism as the only source of ATP is not included in *Table 3*.) In doing so, an own category 'NGAM$_{Turnover}$' was introduced to include all energy expenses attributable to protein and RNA turnover. This includes the ATP spent on protein degradation itself, as well as the fractions of protein synthesis, RNA synthesis, tRNA charging and phosphorylation of mononucleotides to trinucleotides that produce protein and RNA for turnover only (as determined from the protein and RNA degradation reaction constraints). The remainder of the protein and RNA synthesis fluxes then produces protein and RNA to be consumed in the biomass equation; hence, the associated energy consumption is part of the quantifiable fraction of the growth-associated maintenance (GAM) cost, and is hence included in 'GAM$_{Macromolecules}$' in *Table 3* (DNA synthesis being the other cost included). Similarly, 'GAM$_{tRNA\ charging}$' is the fraction of energy expense in tRNA charging attributable to growth-associated protein synthesis. The consumption of the ATP moiety in RNA and biomass production was included in nucleotide metabolism, in order to stay consistent with the definition of the GAM to only include the ATP *hydrolyzed* for growth (including macromolecular synthesis), but not the consumption of the ATP moiety as a precursor (see also *Figure 4*). Accordingly, the ATP hydrolyzed in RNA synthesis was included under 'GAM$_{Macromolecules}$'. Finally, PRPPS (PRPP synthase) is part of central metabolism in the model but as a reaction is independent from energy production in glycolysis. It was hence assigned its own subsystem ('Pentose phosphate pathway') for the purposes of energy usage breakdown.

## Model comparison

Reactions between the models for JCVI-syn3A and for *M. pneumoniae* (iJW145) (*Wodke et al., 2013*) were compared programmatically by associating with each reaction in either model a set of

involved metabolites, excluding water, $P_i$, and $H^+$. By comparing only the involved metabolites, differences in stoichiometry, reversibility, and mass balance between the two models are not considered. To develop a common language of metabolites between the two models, a mapping from the chemical name in the model SBML file to a KEGG compound identifier (C number) was constructed. The KEGG API was used to search for a C number based on the substrate description in the SBML files. When a C number was not found for a particular substrate, the mapping was created by hand. The name to compound map was verified manually by comparing the name given in the model to the name given in the KEGG database to that C number.

A reaction in the JCVI-syn3A model was determined to be equivalent to a reaction in the iJW145 model if the metabolite sets associated with each reaction were equal. The resulting grouping of reactions into common or model-specific reactions was then manually curated to distinguish reactions where different directionalities/reversibilities between models arose from different roles of these reactions in the model (irreversible amino acid influx in JCVI-syn3A vs. irreversible amino acid efflux in iJW145, which has a separate set of ATP-driven amino acid uptake reactions).

## Growth curve measurements

Growth and rate measurements of minimized synthetic cells have been described in detail elsewhere (*Hutchison et al., 2016a*). Briefly, cells were grown in SP4 medium to mid-late log phase in static cultures, then diluted in fresh pre-warmed (37 °C) medium. Subsequent samples obtained over time were centrifuged to remove medium, cells were lysed with dilute detergent, and released dsDNA was measured using the fluorescent stain Quant-iT PicoGreen (Molecular Probes, Eugene, OR). Fluorescence was measured in a 96-well format using a FlexStation 3 fluorimeter (Molecular Devices, San Jose, CA). The net relative fluorescence units (RFU) of samples (after subtracting RFU from a medium control lacking cells), were plotted as $\ln(RFU)$ vs. time from which the doubling times, $\tau_d$ were calculated from the slopes of exponential regression curves ($k$) as

$$\tau_d = \frac{\ln 2}{k}.$$

Rates were measured from log–linear portions of the growth curve. To avoid minor variables such as batch differences among medium preparations and temperature fluctuations, strains with different genomes were compared under identical conditions and within a single experiment. The accuracy and reproducibility of the measurements (reflected in the observed $R^2$ values, see *Figure 14*) allowed the use of single samples, as also observed previously (*Hutchison et al., 2016a*).

## Tn5 mutagenesis

### Experimental methods

We used the procedure described in *Hutchison et al. (2016a)* with minor modifications. A single experiment was performed, which however yielded ~92,000 transformed colonies (see below), and hence ~92,000 insertion events across a genome with 493 genes. This was deemed to yield sufficient statistics. Preparation of terTufPuro transposomes was as described. JCVI-syn3A cells were grown in SP4 media until reaching pH 6.3–6.8. For one transformation reaction, we used 8.8 ml of culture. The cells were centrifuged for 15 min at 4700 rpm at 10 °C in a 50-ml tube. The pellet was resuspended in 3 ml of S/T buffer (Tris 10 mM, sucrose 0.5 M, pH 6.5). The resuspended cells were centrifuged for 15 min at 4700 rpm at 10 °C and the supernatant was removed. The pellet was resuspended in 250 $\mu$l of 0.1 M $CaCl_2$ and incubated for 30 min on ice. Transposomes (2 $\mu$l) and yeast tRNA (10 µg) (Life Technologies, Carlsbad, CA, USA) were mixed gently with the cells. Two ml of 70% poly(ethylene glycol) (PEG) 6000 (Sigma) dissolved in S/T buffer was added. We allowed a maximum of 2 min in contact with PEG at room temperature. The components were mixed well during the 2 min of incubation. S/T buffer (20 ml) was added immediately after 2 min and mixed well. The tube was centrifuged at 8 °C for 15 min at 10,000× g. The supernatant was discarded and thoroughly drained from the tube by inversion onto a Kimwipe. The cells were resuspended well in 1 ml of warm SP4 media and incubated for 3 hr at 37 °C and then plated on SP4 agar with 2 µg/ml of puromycin (Sigma). The plates were incubated for 3–4 days at 37 °C.

An estimated 92,000 colonies were harvested from the plates in 20 ml of SP4 media (passage zero, $P_0$) and a 45 µl sample was added to 45 ml SP4 media containing puromycin 2 µg/ml and

grown for 24 hr (passage one, $P_1$). A 45-μl sample of $P_1$ culture was added to 45 ml of fresh SP4 media ($P_2$) and grown for 48 hr. Two more passages ($P_3$ and $P_4$) were done.

DNA preparations from each passage were done as described in *Hutchison et al. (2016a)*, and DNA preparations were additionally purified by gel electrophoresis. A Nextera XT DNA library preparation kit was used for paired-end library construction (Illumina, San Diego, CA) by the manufacturer's protocol with the following change. A forward primer 5'-AATGATACGGCGACCACCGAGA TCTACACTCTTTCCCTACACGACGCTCTTCCGATCTNNNNNNGCCAACGACTACGCACTAG designed by us and the reverse primer from the kit were used for the PCR amplification to achieve marker-specific sequencing (*Wright et al., 2017*). To locate points of Tn5 insertion in the JCVI-syn3A genome, sequence reads were searched for the Tn5 19 bp terminus followed by an exact 30 bp match to genome sequence. The Tn5 to genome junction point identified the insertion location. 'Duplicate insertions' (i.e. insertions found in sequences repeated in the genome) were ignored in all further analyses since they could not be unequivocally assigned to a single gene. Tn5 insertions were displayed using CLC Genomics Workbench (QIAGEN Bioinformatics, Redwood City, CA).

## Classification of genes

To place genes into 'essential', 'quasi-essential', and 'non-essential' classifications, a simple statistical model of transposon insertion was used. It assumes that the positions of insertions within a gene are unbiased (Poisson distributed) and that the number of insertions in a particular gene is described by one of two possible distributions. The two distributions separate the genes into groups with few and many insertions. The probability to observe $n_i$ insertions for gene $i$, which has a sequence length of $\ell_i$ is

$$P(n_i|\ell_i) = p_{\mathrm{lo}} e^{k_{\mathrm{lo}}\ell_i} \frac{(k_{\mathrm{lo}}\ell_i)^{n_i}}{n_i!} + p_{\mathrm{hi}} e^{k_{\mathrm{hi}}\ell_i} \frac{(k_{\mathrm{hi}}\ell_i)^{n_i}}{n_i!}, \tag{1}$$

where $k_{\mathrm{lo}}$ and $k_{\mathrm{hi}}$ are the transposon insertion rates for the few and many insertion distributions, and $p_{\mathrm{lo}}$ and $p_{\mathrm{hi}}$ are the probabilities that the insertions follow the few or many insertion distribution, respectively, such that $p_{\mathrm{lo}} + p_{\mathrm{hi}} = 1$. The model is fit to the experimental data using expectation–maximization (*Titterington, 2011*).

The probability that a particular observation of $n_i$ transposon insertions for a gene of length $\ell_i$ follows the fewer insertion distribution for passage $j$, is

$$P_{\mathrm{lo}}(n_{ij}|\ell_i,j) = p_{\mathrm{lo},j} e^{k_{\mathrm{lo},j}\ell_i} \frac{(k_{\mathrm{lo},j}\ell_i)^{n_{ij}}}{n_{ij}!}. \tag{2}$$

Comparing passage 1 and passage 4, the classification probabilities are

$$P(\mathrm{Essential}|n_{i1},n_{i4},\ell_i) = P_{\mathrm{lo}}(n_{i1}|\ell_i,1) \cdot P_{\mathrm{lo}}(n_{i4}|\ell_i,4) \tag{3a}$$

$$P(\mathrm{Quasi\text{-}essential}|n_{i1},n_{i4},\ell_i) = [1-P_{\mathrm{lo}}(n_{i1}|\ell_i,1)] \cdot P_{\mathrm{lo}}(n_{i4}|\ell_i,4) \tag{3b}$$

$$P(\mathrm{Non\text{-}essential}|n_{i1},n_{i4},\ell_i) = [1-P_{\mathrm{lo}}(n_{i1}|\ell_i,1)] \cdot [1-P_{\mathrm{lo}}(n_{i4}|\ell_i,4)]. \tag{3c}$$

A gene is assigned to a category if the classification probability is greater than 0.5. Genes where the classification probabilities are all less than 0.5 (labeled 'unclassifiable' in *Figure 2* and *Figure 2—figure supplements 2,3*, or where $P_{\mathrm{lo}}(n_{i1}|\ell_i,1) \cdot [1-P_{\mathrm{lo}}(n_{i4}|\ell_i,4)] > 0.5$ are not classifiable by this method and were manually assigned an essentiality class.

To differentiate weakly quasi-essential genes from non-essential genes, the genes identified as non-essential were further classified using $k$-means clustering (provided by the SciPy (*Jones et al., 2001*) library) of the ratio of transposon insertion counts in $P_4$ to $P_1$.

## Mass spectrometry based proteomics

### Cell preparation

With the objective of studying protein expression changes along the growth curve (unrelated to the current study), mass spectrometry with tandem mass tag labeling was carried out on JCVI-syn3A

samples from different time points along a growth curve. In the current study, we use the data from the first time point (logarithmic phase). JCVI-syn3A cells used for proteomic analysis were grown as described previously (*Hutchison et al., 2016a*) and in 'Growth curve measurements' above, using SP4 medium that contained heat inactivated horse serum (Invitrogen™) in lieu of FBS. Static cultures were sampled at different times to determine the culture stage (measured as described in 'Growth curve measurements'). Six centrifuge bottles each containing approximately 130 mL of a logarithmic phase culture were centrifuged (10,000× g, 15 min, 20 °C) and the cell pellets were drained and resuspended in a small volume of medium. The suspensions were pooled, redistributed in 1 mL volumes and again centrifuged (16,000× g, 5 min, 20 °C). The resulting pellets were drained and used immediately for lysis. In total, three pellets were obtained from independently grown cultures as biological replicates. Samples of two further time points (early plateau and plateau phase) were prepared in an analogous fashion, with three biological replicates each.

## Cell lysis and protein digestion

Cells were lysed in a buffer comprised of 3% SDS, 75 mM NaCl, 1 mM NaF, 1 mM $\beta$-glycerophosphate, 1 mM sodium orthovanadate, 10 mM sodium pyrophosphate, 1 mM PMSF and 1X Roche Complete mini EDTA-free protease inhibitors in 50 mM HEPES, pH 8.5 (*Villén and Gygi, 2008*). Lysates were passed through a 21-gauge needle 20 times and sonicated for 5 min to ensure full lysis. Debris was pelleted by centrifugation at 14,000 rpm for 5 min, with resultant supernatants used for downstream processing. Briefly, proteins were reduced with DTT (*Haas et al., 2006*) and precipitated with methanol-chloroform (*Wessel and Flügge, 1984*) before re-suspension in 1 M urea in 50 mM HEPES, pH 8.5 for digestion. Digestion was performed in a two-step process, 1) with LysC overnight at room temperature, 2) with trypsin for 6 hr at 37 °C. Digestion was quenched with TFA, and peptides desalted with C18 solid-phase extraction (*Tolonen and Haas, 2014*). Dried peptides were re-suspended in 50% acetonitrile/5% formic acid and quantified via BCA assay.

## Tandem mass tag labeling and fractionation

Lyophilized peptides were re-suspended in 30% dry acetonitrile in 200 mM HEPES, pH 8.5 and 8 µL of the appropriate tandem mass tag (TMT) reagent was added to each sample, incubated for 1 hr at room temperature, and quenched with 5% hydroxylamine. Labeled samples were then acidified with 1% trifluoroacetic acid. Differentially labeled samples were pooled into multiplex experiments and then desalted via solid-phase extraction and lyophilized. Samples were fractionated by basic pH reverse-phase liquid chromatography, using a 4.6 mm × 250 mm C18 column on an Ultimate 3000 HPLC (Thermo Fisher Scientific, Waltham, MA, USA). In total, 96 fractions were collected and combined in a concatenated manner (*Wang et al., 2011*), lyophilized and re-suspended in 5% formic acid/5% acetonitrile for identification and quantification by LC-MS2/MS3.

## LC-MS2/MS3 analysis

All LC-MS2/MS3 experiments were performed on an Orbitrap Fusion mass spectrometer with an in-line Easy-nLC 1000 with chilled autosampler (Thermo Fisher Scientific). Peptides were separated on columns that were packed with C4 resin (5 µm, 100 Å), followed by C18 resin (3 µm, 200 Å) and then to a final length of 30 cm with C18 (1.8 µm, 12 Å). Peptides were eluted with a linear gradient from 11% to 30% acetonitrile in 0.125% formic acid over 165 min at a flow rate of 300 nL/min and heating the column to 60 °C. Electrospray ionization was achieved by applying 2000 V through a stainless-steel T-junction. Mass spectrometer settings were as previously described (*Lapek et al., 2017a*).

## Data processing and analysis

Data were processed using the ProteomeDiscoverer 2.1.0.81 software package (Thermo Fisher Scientific). The built-in version of SequestHT (*Eng et al., 1994*) was utilized to assign identities to MS2 spectra searching against the JCVI-syn3A database downloaded from NCBI. The database was appended to include a decoy database comprised of all protein sequences in reversed order for downstream false discovery estimation (*Peng et al., 2003*; *Elias et al., 2005*; *Elias and Gygi, 2007*). Search parameters included a 50 ppm MS1 mass tolerance (*Huttlin et al., 2010*), 0.6 Da fragment ion tolerance, fully-enzymatic trypsin with a maximum of two missed cleavages per peptide, static modifications of 10-plex TMT tags on lysines and peptide n-termini and carbamidomethylation of

cysteines. Variable modifications included oxidation of methionines and phosphorylation of serine, threonine and tyrosine residues. Data were filtered to a peptide and protein false discovery rate of less than 1% using the target-decoy search strategy (*Elias and Gygi, 2007*). Peptides matching to multiple proteins were assigned to the protein containing the largest number of matched redundant peptides following the law of parsimony (*Huttlin et al., 2010*). TMT reporter ion intensities were extracted from MS3 spectra for quantitative analysis. Spectra used for quantitation had to meet the requirements of greater than 10 average signal-to-noise per label and isolation interference of less than 25% (*Ting et al., 2011*). Data were normalized as previously described (*Lapek et al., 2017b*; *Lapek et al., 2017a*). In order to convert relative protein abundances obtained from the mass spectrometry data to absolute cellular abundances (i.e. number of molecules per cell for each protein species), the average protein length and amino acid composition, and hence average molecular weight, were calculated from the relative abundances and known protein sequences. The molecular weight of the average JCVI-syn3A protein was then used to estimate the total number of all proteins in JCVI-syn3A based on the protein dry mass fraction and cellular dry weight (see Section 'Biomass composition and reaction'). This estimated total number of proteins was used to scale relative abundances of proteins in the proteome to absolute abundances in the average cell of JCVI-syn3A. The mass spectrometry data has been deposited on MassIVE with accession number 000081687 (ftp:// massive.ucsd.edu/MSV000081687). The ProteomeXchange accession number is PXD008159. (*Lapek and Gonzalez, 2017*).

## Omics scale visualization

Voronoi treemaps (*Figures 1*, *3* and *17a*) were constructed following (*Liebermeister et al., 2014*). Briefly, the genetic loci were associated with a KEGG orthology (KO) identifier (*Kanehisa et al., 2015*). Mappings between KO identifiers and locus tags were acquired from KEGG Genomes for *M. pneumoniae* (T00006) and *E. coli* (T00944). A mapping between genes and KO identifiers for JCVI-syn3A was derived from the locus tag/KO map for *M. mycoides capri* LC str. 95010 (T01478) by matching *M. mycoides capri* genes to JCVI-syn3A genes using a reciprocal best hit BLASTp search (*Altschul et al., 1990*). Since the KO identifier, in general, can associate multiple functionality to a single ortholog, it was necessary to choose a single function for each ortholog. Initially, the KO/function assignment was taken from *Liebermeister et al. (2014)*. Mycoplasma specific genes were then added to this hierarchy manually. Genes for which no ortholog could be assigned, but were well annotated in the genome were also added to the hierarchy manually. Voronoi treemaps were constructed by first using the freely available software described by *Nocaj and Brandes (2012)* to generate the vertices of the polygons comprising the Voronoi tessellation, then rendering the resulting treemap using Cairo (*Packard et al., 2018*).

## Acknowledgements

We thank James Daubenspeck and Kevin Dybvig for information on the minimal cell capsule; Emile van Schaftingen for helpful suggestions on possible reactions in central metabolism; and David Bianchi for help with *Figure 15—figure supplement 1*. MB gratefully acknowledges an NSF Postdoctoral Fellowship through the Center for the Physics of Living Cells (CPLC) at the University of Illinois at Urbana-Champaign (grant NSF PHY 1430124). TE and ZLS were supported through grant NSF PHY 1430124. MB, TE and ZLS were partially supported through grants NSF MCB-1244570 and NSF MCB 1818344. PL was supported by the DOE/BER (ORNL 4000134575) as part of the Adaptive Biosystems Imaging Focus at ORNL (Oak Ridge National Laboratory). JL was supported through grant NIH K12 GM06852. VCL and AH were supported through grant NSF MCB-1611711. DG is supported by the Ray Thomas Edwards Foundation and the University of California Office of the President. CM, KW, CH, HOS and JG were supported by the J Craig Venter Institute. The bacterium JCVI-syn3A was created using funding from the J Craig Venter Institute, Synthetic Genomics, Inc, and the Defense Advanced Research Projects Agency's Living Foundries program (contract HR0011-12-C-0063).

# Additional information

## Competing interests

Clyde A Hutchison: is a consultant for Synthetic Genomics, Inc. (SGI), and holds SGI stock and/or stock options. Hamilton O Smith: is on the Board of Directors and cochief scientific officer of Synthetic Genomics, Inc. (SGI) and holds SGI stock and/or stock options. The other authors declare that no competing interests exist.

## Funding

| Funder | Grant reference number | Author |
| --- | --- | --- |
| National Science Foundation | PHY 1430124 Postdoctoral Fellowship | Marian Breuer |
| National Science Foundation | PHY 1430124 | Marian Breuer Emmy E Earnest Zaida Luthey-Schulten |
| National Science Foundation | MCB-1244570 | Marian Breuer Emmy E Earnest Zaida Luthey-Schulten |
| National Science Foundation | MCB 1818344 | Marian Breuer Emmy E Earnest Zaida Luthey-Schulten |
| J Craig Venter Institute | | Chuck Merryman Kim S Wise Clyde A Hutchison Hamilton O Smith John I Glass |
| National Institutes of Health | K12 GM06852 | John D Lapek |
| University of California | Office of the President | David J Gonzalez |
| Ray Thomas Edwards Foundation | | David J Gonzalez |
| National Science Foundation | MCB-1611711 | Valérie de Crécy-Lagard Andrew D Hanson |
| U.S. Department of Energy | ORNL 4000134575 | Piyush Labhsetwar |

The funders had no role in study design, data collection and interpretation, or the decision to submit the work for publication.

## Author contributions

Marian Breuer, Data curation, Formal analysis, Investigation, Visualization, Methodology, Writing—original draft, Writing—review and editing; Emmy E Earnest, Data curation, Software, Formal analysis, Investigation, Visualization, Methodology, Writing—original draft, Writing—review and editing; Chuck Merryman, Clyde A Hutchison, Conceptualization, Data curation, Software, Formal analysis, Validation, Writing—review and editing; Kim S Wise, Conceptualization, Resources, Validation, Investigation, Writing—review and editing; Lijie Sun, Validation, Investigation; Michaela R Lynott, Data curation, Software, Validation; Hamilton O Smith, Conceptualization, Formal analysis, Funding acquisition, Writing—review and editing; John D Lapek, Investigation, Writing—original draft; David J Gonzalez, Conceptualization, Resources, Supervision, Funding acquisition; Valérie de Crécy-Lagard, Investigation, Writing—review and editing; Drago Haas, Investigation; Andrew D Hanson, Funding acquisition, Investigation, Writing—review and editing; Piyush Labhsetwar, Data curation, Investigation, Visualization, Writing—original draft; John I Glass, Conceptualization, Resources, Supervision, Funding acquisition, Project administration, Writing—review and editing; Zaida Luthey-Schulten, Conceptualization, Resources, Supervision, Funding acquisition, Writing—original draft, Project administration, Writing—review and editing

**Author ORCIDs**
Marian Breuer (iD) https://orcid.org/0000-0002-0529-4031
Emmy E Earnest (iD) https://orcid.org/0000-0002-1630-0791
Zaida Luthey-Schulten (iD) http://orcid.org/0000-0001-9749-8367

**Decision letter and Author response**
Decision letter https://doi.org/10.7554/eLife.36842.sa1
Author response https://doi.org/10.7554/eLife.36842.sa2

## Additional files

### Supplementary files

• Supplementary file 1. Gene classification hierarchy, model reaction breakdown and protein gene breakdowns for *M. pneumoniae* and *E. coli*.

• Supplementary file 2. Transposon insertion nucleotide positions.

• Supplementary file 3. Transposon insertion counts and assignment of gene essentiality from both transposon mutagenesis and FBA.

• Supplementary file 4. Reactions and metabolites included in the metabolic reconstruction.

• Supplementary file 5. Data from proteomics experiments.

• Supplementary file 6. Comparison of the JCVI-syn3A metabolic reconstruction to that of *M. pneumoniae* published by *Wodke et al. (2013)*.

• Supplementary file 7. Flux constraints derived from proteomics and turnover numbers and comparison to FBA fluxes.

• Supplementary file 8. Known metabolic reactions removed during genome minimization from JCVI-syn1.0 to JCVI-syn3A.

• Supplementary file 9. FBA model in sbml format.

• Supplementary file 10. FBA model in json format.

• Supplementary file 11. ESCHER network map in json format.

• Transparent reporting form

### Data availability

Proteomics: data were uploaded to MassIVE (massive.ucsd.edu) with dataset identifier MSV000081687 and ProteomeXchange with dataset identifier PXD008159. All other new data are included in the manuscript and supporting files.

The following dataset was generated:

| Author(s) | Year | Dataset title | Dataset URL | Database and Identifier |
|---|---|---|---|---|
| Lapek JD Jr, Gonzalez DJ | 2018 | Data from Essential Metabolism for a Minimal Cell | http://proteomecentral. proteomexchange.org/ cgi/GetDataset?ID= PXD008159 | ProteomeXchange, PXD008159 |

The following previously published datasets were used:

| Author(s) | Year | Dataset title | Dataset URL | Database and Identifier |
|---|---|---|---|---|
| John I Glass | 2017 | Synthetic bacterium JCVI-Syn3.0 strain 6d, complete genome | https://www.ncbi.nlm. nih.gov/nuccore/ CP016816.2 | NCBI Nucleotide, CP0 16816.2 |
| Jeong H, Barbe V, Vallenet D, Choi S-H, Lee CH, Lee S-W, Vacherie B, Yoon SH, Yu D-S, | 2017 | Escherichia coli B str. REL606, complete genome | https://www.ncbi.nlm. nih.gov/nuccore/NC_ 012967.1 | NCBI Nucleotide, NC_012967.1 |

| | | | | |
|---|---|---|---|---|
| Cattolico L, Hur C-G, Park H-S, Segurens B, Blot M, Schneider D, Studier FW, Oh TK, Lenski RE, Daegelen P, Kim JF | | | | |
| Hutchison CA III, Chuang R-Y, Noskov VN, Assad-Garcia N, Deerinck TJ, Ellisman MH, Gill J, Kannan K, Karas BJ | 2016 | Synthetic bacterium JCVI-Syn3.0, complete genome | https://www.ncbi.nlm.nih.gov/nuccore/CP014940.1 | NCBI Nucleotide, CP014940.1 |
| Gibson DG, Glass JI, Lartigue C, Noskov VN, Chuang RY, Algire MA, Benders GA, Montague MG, Ma L, Moodie MM, Merryman C, Vashee S, Krishnakumar R, Assad-Garcia N, Andrews-Pfannkoch C, Denisova EA, Young L, Qi ZQ, Segall-Shapiro TH, Calvey CH, Parmar PP, Hutchison CA III, Smith HO, Venter JC | 2010 | Synthetic Mycoplasma mycoides JCVI-syn1.0 clone sMmYCp235-1, complete sequence | https://www.ncbi.nlm.nih.gov/nuccore/CP002027.1 | NCBI Nucleotide, CP002027.1 |
| Himmelreich R, Hilbert H, Plagens H, Pirkl E, Li BC, Herrmann R, Dandekar T, Huynen M, Regula JT, Ueberle B, Zimmermann CU, Andrade MA, Doerks T, Sanchez-Pulido L, Snel B, Suyama M, Yuan YP, Bork P | 2014 | Mycoplasma pneumoniae M129, complete genome, | https://www.ncbi.nlm.nih.gov/nuccore/U00089.2 | NCBI Nucleotide, U00089.2 |

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

# Appendix 1

## Further model details and justification

In the following, we discuss specific aspects of the derivation of the biomass composition and of the reconstruction of the metabolic network that were not covered in the main text.

### Biomass composition

*Appendix 1—table 1* lists the mass fractions for all components included in the JCVI-syn3A biomass composition (*Figure 4*). Mass fractions for the different biomass components are obtained from various sources; since the mass fractions thus determined initially add up to ~106%, we finally rescale all mass fractions to a total of 100%. (In doing so, we keep the DNA fraction fixed to ensure a total DNA mass corresponding to one chromosome). These rescaled numbers are the ones shown in *Appendix 1—table 1*.

**Appendix 1—table 1.** Reconstructed biomass composition of JCVI-syn3A, listing the fraction of each component as percent of cellular dry mass.

| Category | Component | Fraction [%] |
| --- | --- | --- |
| Macromolecules | Protein | 54.727 |
| Total: 76.521 % | RNA | 16.274 |
| | DNA | 5.5 |
| | Acyl carrier protein | 0.018 |
| | dUTPase | 0.003 |
| Lipids & capsule | Lipogalactan capsule | 6.368 |
| Total: 17.563 % | Phosphatidylglycerol | 2.944 |
| | Cardiolipin | 2.944 |
| | Cholesterol | 1.534 |
| | Diacylglycerol | 1.366 |
| | Gal-DAG | 1.31 |
| | Triacylglycerol | 0.549 |
| | Fatty acid | 0.549 |
| Small molecules & ions | Potassium | 3.285 |
| Total: 5.916 % | Phosphate | 0.375 |
| | Chloride | 0.198 |
| | ATP | 0.167 |
| | L-lysine | 0.133 |
| | Sodium | 0.131 |
| | Spermine | 0.128 |
| | L-isoleucine | 0.115 |
| | GTP | 0.115 |
| | L-leucine | 0.112 |
| | UTP | 0.106 |
| | L-glutamate | 0.085 |
| | L-valine | 0.083 |
| | L-asparagine | 0.083 |
| Small molecules | L-alanine | 0.077 |
| & ions (cont'd) | L-aspartate | 0.072 |
| | L-threonine | 0.071 |

*Continued on next page*

| | |
|---|---|
| L-serine | 0.071 |
| Glycine | 0.068 |
| CTP | 0.053 |
| L-phenylalanine | 0.049 |
| L-glutamine | 0.048 |
| L-arginine | 0.043 |
| L-tyrosine | 0.041 |
| L-proline | 0.035 |
| L-methionine | 0.026 |
| Magnesium | 0.019 |
| L-histidine | 0.019 |
| Calcium | 0.019 |
| FAD | 0.016 |
| 5,10-meTHF(Glu)$_3$ | 0.015 |
| CoA | 0.012 |
| Thiamin diphosphate | 0.009 |
| NADP+ | 0.008 |
| L-tryptophan | 0.008 |
| L-cysteine | 0.008 |
| Pyridoxal phosphate | 0.005 |
| dTTP | 0.003 |
| dATP | 0.003 |
| dCTP | 0.002 |
| dGTP | 0.001 |

## Applicability of macromolecular composition

The macromolecular composition is based on *M. mycoides capri* (*Razin et al., 1963*) whose genome is approximately twice as large as the one of JCVI-syn3A. However, the protein dry mass fraction in *M. mycoides capri* appears to be a good initial approximation for JCVI-syn3A. A protein dry mass fraction of ~40–60% is generally observed in bacteria, for example in different mycoplasmas (*Razin et al., 1963*) or *E. coli* (*Bremer and Dennis, 2008*). In particular, we note that *Acholeplasma laidlawii* PG8, in spite of having a 1.5-times larger genome (*Lazarev et al., 2011*) than *M. mycoides capri* but a comparable cell size (*Folmsbee et al., 2010*), shows a similar protein dry mass fraction (55%) to *M. mycoides capri* (58%) (*Razin et al., 1963*).

While no such argument can be made per se for the conservation of the RNA content, we note that the assumed RNA dry mass fraction agrees reasonably well with ribosomal protein abundance from our proteomics data: The average copy number across all ribosomal proteins in JCVI-syn3A comes out at 340 copies. If this number was interpreted as an estimate of the total number of ribosomes, it would come reasonably close to the upper limit of ~670 ribosomes per average cell if all RNA was present as ribosomal RNA (see Section 'Macromolecules'). The presence of the same number of rRNA operons (2) in both JCVI-syn1.0 and JCVI-syn3A is also consistent with the assumption of comparable rRNA contents in the two organisms. Thus, the RNA content from *M. mycoides capri* should provide a reasonable approximation for the RNA content in JCVI-syn3A.

The only required adaptation was a slight increase in the DNA mass fraction from the 5% reported for *M. mycoides capri* (*Razin et al., 1963*) to 5.5%, since this corresponds to exactly one chromosome of a 543,379 bp genome in a 400 nm spherical cell with a dry weight of ~10.2 fg. The dry weight is obtained assuming a density of 1.1 g/ml, which has been found in different bacterial cells (*Bratbak and Dundas, 1984*); and around 4.8 $\mu$l water/mg cellular protein as measured in *M. mycoides capri* serovar capri PG3 (*Leblanc and Le Grimellec, 1979*), corresponding to a cellular water content of 72% in JCVI-syn3A. Bacterial cells in exponential growth phase can on average

contain more than one chromosome (*Cooper and Helmstetter, 1968*); until such data becomes available for JCVI-syn3A though, we stick to the assumption of one chromosome per cell, since this stays close to the DNA dry mass fraction of 5% reported for *M. mycoides capri*. (E.g., 1.5 chromosomes in an average cell would already imply a dry mass fraction of 8.25%.) As an aside, the above means that the reported DNA fraction for *M. mycoides capri* (*Razin et al., 1963*) is lower than expected for a whole *M. mycoides capri* chromosome. A similar discrepancy has been observed in the reconstruction of the *M. genitalium* biomass (*Karr et al., 2012*).

## Details on lipid composition

The overall lipid composition of *M. mycoides capri* serovar capri PG3 has been studied previously (*Archer, 1975*) and found to comprise phospholipids, glycolipids, cholesterol, free fatty acids and (mono-,di-,tri-)glycerides. Mono- and diglycerides could not be distinguished and are included as diglycerides in the model. In addition, it has been shown that the only phospholipids produced by *M. mycoides capri* LC Y are phosphatidylglycerol and cardiolipin; no phosphatidylcholine or -ethanolamine are synthesized, although lecithin can be incorporated as a whole if present in the media (*Plackett, 1967a*). The phospholipid fraction is assumed to be equal parts phosphatidylglycerol and cardiolipin. The same study also identified the glycolipid as monogalactosyl-diacylglycerol (Gal-DAG) (*Plackett, 1967a*). JCVI-syn3A contains the pathway from glucose-6-phosphate to UDP-galactofuranose, and the glycosyltransferases *cps/*0114 and 0697 are both 20% identical to the experimentally confirmed glucosyl-/galactosyltransferases MG517 from *M. genitalium* (*Andrés et al., 2011*) and MPN483 from *M. pneumoniae* (*Klement et al., 2007*). We therefore assume that the glycolipid in JCVI-syn3A is a Gal-DAG as well.

For the fatty acids, palmitic acid (C16:0) and oleic acid (C18:1 cis-9) are considered to be the two most important representatives. While it is known that the abundance of different fatty acids in mycoplasma cell membranes can be affected by the media composition (*Archer, 1975*; *Razin, 1969*; *McElhaney and Tourtellotte, 1969*), we note that these two fatty acids are the only ones in the minimal media for *M. mycoides capri* LC Y (*Rodwell, 1969*) and *M. pneumoniae* (*Yus et al., 2009*) and also yielded among the highest growth rates and cell yields in a screen of fatty acid combinations (*Rodwell, 1967*). We thus define a metabolite species 'fatty acid' with an average molecular weight between palmitate and oleate, which is used in all lipid species.

## Genetic evidence for capsule production in JCVI-syn3A

While mycoplasmas lack a cell wall (*Razin et al., 1998*), several *Mycoplasma* spp. do produce capsular polysaccharides (CPS) or secrete polysaccharides into the medium (exopolysaccharides, EPS) (*Bertin et al., 2015*). In particular, *M. mycoides capri* LC GM12, the strain from which JCVI-syn3A is derived, has been shown to produce a galactan (specifically, poly-$\beta-1\rightarrow6$-galactofuranose) (*Schieck et al., 2016*); and other *M. mycoides capri* LC strains have been demonstrated to produce a galactan CPS but secrete negligible amounts of EPS (*Bertin et al., 2015*). This galactan has been suggested to play a role in membrane integrity in *M. mycoides capri* LC GM12, and deletion of the *glf* gene decreases the growth rate, possibly due to increased energy expenses to maintain cell homeostasis (*Schieck et al., 2016*). While it is not yet experimentally known whether the minimal cell still produces this galactan polysaccharide, genetic features suggest it does. A galactan CPS is therefore included in the JCVI-syn3A biomass composition, with a dry weight fraction of 6.77% before rescaling (see next subsection).

The assumption of capsule production in JCVI-syn3A rests on two glycosyltransferases, whose homologs in another mycoplasmas are candidates for capsule synthesis and attachment. Specifically, the two putative glycosyltransferases *epsG/*0113 and *cps/*0114 have homologs in *M. mycoides mycoides* PG1$_T$, EpsG/MSC_0108 (84% sequence identity to *epsG/*0113) and cps/MSC_0109 (92% sequence identity to *cps/*0114). Based on in silico and preliminary experimental studies, a tentative mechanism has been suggested for galactan capsule synthesis in *M. mycoides mycoides* PG1$^T$(*Bertin et al., 2013*). Based on its predicted high structural similarity to both the cellulose synthase BcsA of *Rhodobacter sphaeroides* (*Morgan et al., 2013*) as well as the galactosyltransferase GlfT of *Mycobacterium tuberculosis* (*Wheatley et al., 2012*), the glycosyltransferase EpsG of *M. mycoides mycoides* has been hypothesized to cytoplasmically polymerize UDP-galactofuranose and

export it to the cell exterior (*Bertin et al., 2013*), a mechanism generally assumed for bacteria with a single membrane and no periplasmic space (*Daubenspeck et al., 2009*). Polysaccharide synthesis independent from the mono-Gal-DAG would also be consistent with experimental observations indicating that the latter does not serve as precursor for the galactan polymer in *M. mycoides mycoides* strain V5 (*Plackett, 1967b*). cps would then be a candidate to attach the galactan chain to the cell membrane, a hypothesis supported by its differential expression in capsulated vs. non-capsulated colony variants of *M. mycoides mycoides* (*Bertin et al., 2013*). As the sequences of *epsG/*0113 and *cps/*0114 of JCVI-syn3A are very similar to their homologs, and in particular the conserved motifs DXD and QXXRW (common for processive enzymes (*Bertin et al., 2013*; *Saxena et al., 1995*)) are also present in 0113, we assume the same tentative mechanism to apply in JCVI-syn3A as well. The likely lipid acceptor for the galactan chain would then be DAG which is the substrate for the glycosyltransferases MG517 in *M. genitalium* (*Andrés et al., 2011*) and MPN483 in *M. pneumoniae* (*Klement et al., 2007*) (and which we already assume to be the substrate for mono-Gal-DAG production by *cps/*0114 and 0697 as well).

## Composition and mass fraction of capsule

*M. mycoides mycoides* was found to contain ~10% dry weight of galactose in form of galactan (the only other carbohydrate species being ribose from nucleic acids) (*Buttery and Plackett, 1960*), suggesting this polysaccharide to completely account for the 8.1% carbohydrate measured in *M. mycoides mycoides* cell residues defatted prior to measurement (*Razin et al., 1963*) (monogalactosyl-lipids removed during defatting possibly contributing to the remaining difference). We therefore assume that the 6.5% carbohydrate in defatted *M. mycoides capri* (*Razin et al., 1963*) consists mainly of galactan.

The intact lipopolysaccharide moiety studied in *Buttery and Plackett (1960)* contained around 4% lipid. If we assume the latter to be palmityl oleyl glycerol (594.95 g/mol), then the average galactan chain length must be 88 galactosyl residues ($C_6H_{10}O_5$: 162.006 g/mol). We note that with ~0.5 nm per monomer, this yields a polysaccharide chain of ca. 40–50 nm, which seems consistent with the capsular thickness visible in TEM imaging of *M. pulmonis* (*Daubenspeck et al., 2009*). We therefore include $Gal_{88}$-DAG as the lipogalactan moiety in our biomass (before rescaling) with a dry weight fraction of 6.77% (6.5% carbohydrate +0.27% lipid).

As an aside on capsule composition, we note that the monosaccharide rhamnose was unexpectedly not detected using GC/MS (gas chromatography/mass spectrometry) in the minimal cell (personal communication with James Daubenspeck and Kevin Dybvig). It is present in wild-type *M. mycoides capri*. Mycoplasmas can convert oligomeric but not monomeric glucose to rhamnose (*Jordan et al., 2013*). The enzyme that catalyzes rhamnose synthesis has not been identified. Rhamnose is thought to link proteins to phospholipids as a mechanism of trafficking proteins to the membrane. Proteins that are cytoplasmic when not associated with rhamnose, such as the glycolytic enzyme enolase, moonlight on the cell surface when modified by the addition of rhamnose and phospholipid (*Daubenspeck et al., 2016*). Because mutants lacking rhamnose have not been described in global transposon mutagenesis studies of mycoplasmas, it was thought that this system was essential for mycoplasmas and possibly other bacteria.

## Details on small molecule pool composition

In addition to macromolecules, lipids and capsule, pools of free amino acids, nucleotides and deoxynucleotides are also included in the biomass, as well as cofactors and ions expected to be needed in JCVI-syn3A. A minimal medium for JCVI-syn3A has yet to be obtained, so the minimal media reported for *M. mycoides capri* LC Y (*Rodwell, 1969*) and *M. pneumoniae* (*Yus et al., 2009*) are used as a guideline for required ions and cofactors: Any compound present in a minimal medium is required by the cell, and the compound or its downstream product(s) need to be included in the biomass composition. From the two media mentioned, all inorganic ions are included in the JCVI-syn3A biomass composition except for sulfate. There is no known biological need for sulfate in JCVI-syn3A, since sulfur needed for certain tRNA modifications can be derived from cysteine via cysteine desulfurase (*iscS/*0441). Also, there is no obvious transporter candidate: A putative sulfate import system has been identified in *Mycoplasma hyopneumoniae* (*Ferrarini et al., 2016*) (MHP168_157/158) but

their homologs have been deleted in JCVI-syn3A (MMSYN1_0192 and MMSYN1_0193). The only other anion import system, the phosphate import system Pst (*pstS*/0425 through *pstB*/0427) is known to be highly selective for phosphate over sulfate (*Luecke and Quiocho, 1990*; *Elias et al., 2012*) and hence is no plausible candidate for sulfate uptake. Finally, we note that while the minimal media for *M. mycoides capri* LC Y (*Rodwell, 1969*) includes $MgSO_4$, a study on this organism's inorganic requirements (*Rodwell, 1967*) only reported a need for $Mg^{2+}$.

Transition metal ions (possibly present as trace contaminants in the media) are also not included in the biomass. While the need for divalent cations other than $Mg^{2+}$ and $Ca^{2+}$ is not clear, it was observed that addition of 1 μM of $Mn^{2+}$, $Zn^{2+}$, $Co^{2+}$ or $Fe^{2+}$ actually inhibited growth of *M. mycoides capri* LC Y (*Rodwell, 1967*).

From the nine vitamins needed by *M. pneumoniae*, we exclude choline based on experimental evidence that *M. mycoides capri* does not synthesize its own phosphatidylcholine (*Plackett, 1967a*). For the remaining vitamins, there is an apparent need in JCVI-syn3A as cofactors or coenzymes (or in the case of spermine, to stabilize nucleic acids) and they (or their final forms) are thus included in the biomass. For lipoate, an uptake reaction is included in the model (see below), but lipoate is not actually included in the biomass in any form, as the holo-protein, lipoyl-PdhC (*pdhC*/0227), itself turns out to be nonessential in our model. We note that (*Rodwell, 1969*) reports pyridoxal or folate derivatives are not required by *M. mycoides capri* LC Y, which is consistent with reports of *M. mycoides capri* LC Y being able to omit methionyl-tRNA formylation in the absence of folate derivatives without impact on growth (*Neale et al., 1981*). However, the transposon mutagenesis data for JCVI-syn3A gives folate-related enzymes as quasi-essential, and the pyridoxal phosphate-dependent IscS (*iscS*/0441) as essential. Both pyridoxal phosphate and tetrahydrofolate are therefore included in the biomass.

Intracellular concentrations/mass fractions for vitamins and ions are taken from the iJO1366 *E. coli* model (*Orth et al., 2011*), except for potassium and sodium (which have been determined in *M. mycoides capri* PG3 (*Leblanc and Le Grimellec, 1979*)), chloride (which has been determined in *Mycoplasma gallisepticum* (*Linker and Wilson, 1985*)), and phosphate (which is taken from the *M. pneumoniae* model (*Wodke et al., 2013*)). Mass fractions for free nucleotides and deoxynucleotides have been determined in *M. mycoides capri* LC Y (*Mitchell and Finch, 1979*; *Neale et al., 1983a*), and relative mole fractions of free amino acids are approximated to match the average protein composition of JCVI-syn3A. The total amino acid mass fraction is assumed to resemble that of *M. pneumoniae* (*Wodke et al., 2013*).

## Metabolic reconstruction
### Possible oxidation of acetaldehyde

Oxidation of acetaldehyde to acetyl-CoA could provide a rationale for the only partial deletion of pyruvate dehydrogenase (PDH) in JCVI-syn3A, as it would not require a decarboxylation step in the absence of the PDH_E1 subunit (MMSYN1_0225 and MMSYN1_0226). (*Cocks et al., 1985*) ruled out alcohol dehydrogenase or acetaldehyde dehydrogenase activity in *M. mycoides capri* LC Y, consistent with the absence of a corresponding gene in JCVI-syn3A; however, the assay mixture compositions did not mention coenzyme A, so that acetaldehyde oxidation to acetyl-CoA instead of acetate might not have been detectable. Therefore, the model tentatively includes an acetaldehyde oxidation reaction catalyzed by PdhC (*pdhC*/0227). Furthermore, assuming the membrane permeability of acetaldehyde to be comparable to that of acetamide, which is five times higher than for glycerol (*Raven, 1984*), suggests that acetaldehyde could also passively leave the cell. Thus a direct secretion reaction is included for acetaldehyde.

### Presence and absence of specific nucleoside kinase and phosphorylase reactions

The substrate profile of the two nucleoside kinases *dak1*/0330 and *dak2*/0382 is inferred from in vitro (*Welin et al., 2007*) and in vivo (*Mitchell et al., 1978*; *Mitchell and Finch, 1979*) studies. Both proteins show significant sequence identity to the *M. mycoides mycoides* SC kinase MSC_0388. *dak1*/0330 (previously annotated as deoxyguanosine kinase) is identical to MSC_0388 up to a single C-terminal residue, and the putative deoxynucleoside kinase 0382 (*Hutchison et al., 2016a*) shares

37% identity with MSC_0388. MSC_0388 was shown to act on deoxyadenosine, deoxyguanosine and deoxycytidine, with weaker activity against adenosine and guanosine and very weak activity against cytidine (*Welin et al., 2007*). Together with the thymidine kinase *tdk/*0140, this should cover the phosphorylation of all deoxyribonucleosides in JCVI-syn3A, leaving the question of the role of the putative third deoxynucleoside kinase *dak2/*0382. The sequence of this protein shows some changes in the active site residues compared to the crystal structure for MSC_0388 (*Welin et al., 2007*); however, Tyr43 close to the ribosyl-C2 in the *M. mycoides mycoides* SC crystal structure (possibly responsible for the preference for deoxyribonucleosides by blocking the space of a C2-hydroxyl group) is preserved in 0382 (Tyr45), suggesting the same preference for deoxyribonucleosides. Lacking further information, a similar substrate profile for this third kinase is assumed as for the deoxyadenosine kinase, and deoxyadenosine/-guanosine/-cytidine phosphorylation is considered to be carried out by either enzyme. One possible explanation for the presence of both kinases could be complementary activities in vivo: Activity of MSC_0388 for deoxyguanosine/-cytidine was found to be strongly inhibited by the presence of deoxyadenosine (*Wang et al., 2001*), so that in in vivo, not all reactions might be carried out by both kinases. The fact that almost no activity was found against cytidine suggests that the cytidine kinase activity observed in cell extracts of *M. mycoides capri* LC Y (*Mitchell and Finch, 1979*) arises from uridine kinase instead (uridine kinase activity was partially inhibited by cytidine (*Mitchell and Finch, 1979*)), which has been deleted in JCVI-syn3A (MMSYN1_0491). Weak activity against adenosine and guanosine has been observed (*Welin et al., 2007*), but this activity was not found in cell-free extracts (*Mitchell et al., 1978*) and the presence of the corresponding phosphoribosyltransferases in the minimized genome suggests that it could not provide AMP and GMP in sufficient quantities. Thus adenosine (or guanosine) kinase reactions are not included, and consequently there is no direct conversion of adenosine to AMP or guanosine to GMP. Instead, it is assumed that AMP, GMP and UMP are formed from their respective bases by the corresponding phosphoribosyltransferases alone (*hptA/*0216, *apt/*0413, and *upp/*0798).

While purine nucleoside phosphorylase activity is present in JCVI-syn3A (see main text), it appears as if no such activity exists for pyrimidines in JCVI-syn3A. Cytidine phosphorylase activity has been ruled out in *M. mycoides capri* LC Y cell extracts (*Mitchell and Finch, 1979*). Phosphorolysis of uridine (*Mitchell and Finch, 1977*; *Mitchell and Finch, 1979*; *Cocks et al., 1985*), deoxyuridine and thymidine (*Neale et al., 1983a*) has been observed; furthermore, the latter two activities could be attributed to the same enzyme (*Neale et al., 1983a*). This is in agreement with the experimentally observed activity of MHR_0565 (*Vande Voorde et al., 2012*), the homolog in *Mycoplasma hyorhinis* HUB-1 of the putative pyrimidine nucleoside phosphorylase MMSYN1_0734 (45% sequence identity): MHR_0565 was found to phosphorolyse thymidine, deoxyuridine, and uridine, but not cytidine or deoxycytidine. As MMSYN1_0734 has been deleted from JCVI-syn3A, we assume that JCVI-syn3A no longer exhibits pyrimidine nucleoside phosphorolysis activity. We note that the *M. mycoides capri* thymine uptake requirement can also be met by thymidine in the growth media (*Neale et al., 1983a*), which means that there is no need for a thymidine phosphorylase activity by some unidentified paralog of MMSYN1_0734.

## Uptake forms of thiamine and lipoate

Deletion of thiamine diphosphokinase (MMSYN1_0261) suggests that thiamine diphosphate (ThDP) must be taken up directly. The substrate-binding protein 0708 shows similarity to both the ThDP-binding Cypl from *Mycoplasma hyorhinis* (*Sippel et al., 2009*) (24% identity) and the thiamine-binding MG289 from *M. genitalium* (*Sippel et al., 2011*) (23%). However, a sequence alignment reveals that the diphosphate-stabilizing interactions in Cypl are largely missing in 0708, as is the case in MG289. (*Sippel et al., 2011*) note that the mere absence of these interactions would not yet exclude ThDP binding, an idea supported by a structural alignment of ThDP-bound Cypl and MG289 suggesting that ThDP could bind to the MG289 binding pocket as well (see *Appendix 1—figure 1*). In light of this, the conservative assumption is made that thiamine is directly taken up as ThDP.

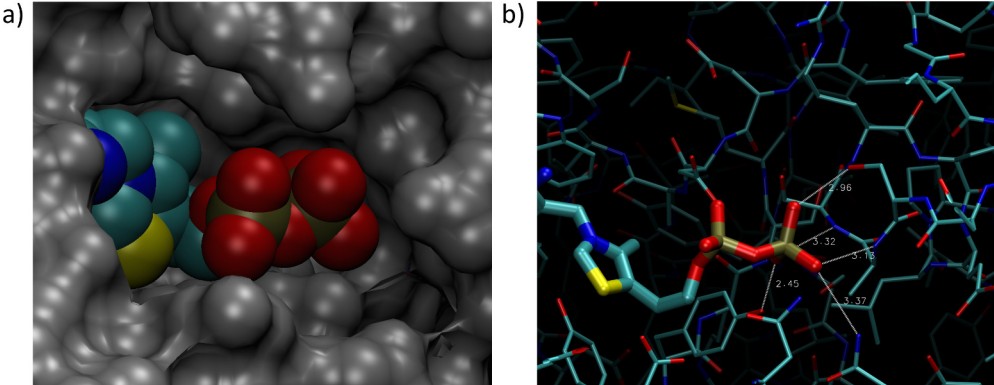

**Appendix 1—figure 1.** Thiamin diphosphate (ThDP) from the *Mycoplasma hyorhinis* Cypl crystal structure (pdb: 3EKI) overlaid onto the crystal structure of MG289 (pdb: 3MYU). (Structures aligned using STAMP (*Russell and Barton, 1992*) in VMD (*Humphrey et al., 1996*; *Eargle et al., 2006*).) a): Space-filling view, with MG289 in gray and ThDP in color. The pyrophosphate tail of ThDP from the Cypl structure would have an appropriate cavity in MG289 as well. b): Visualization of hydrogen bonds for the same alignment. All possible hydrogen bonds are shown between potential donor and acceptor heavy atoms within 3.5 Å or less of each other. Even in the absence of the residues involved in pyrophosphate binding in Cypl (*Sippel et al., 2009*), the alignment suggests other side group and backbone interactions could still allow for pyrophosphate binding.

Lipoate is a required cofactor for PdhC (*pdhC*/0227), but the deletion of two putative lipoyl transferases (MMSYN1_0224 and MMSYN1_0464) suggests an alternative lipoylation mechanism for PdhC. Such an alternative mechanism would be transamidation using a lysine- or peptide-bound lipoate, for which putative peptidases with a covalent mechanism would be candidates. Among the remaining genes in JCVI-syn3A, 0401 has been annotated by RAST as homologous to the sublancin 168 lantibiotic transporter (*Paik et al., 1998*), which exports bactericidal peptides and simultaneously cleaves off a leader peptide containing a double-glycine motif in the process at its peptidase C39 domain. However, no candidate for such a double-glycine motif peptide has been identified in JCVI-syn3A; this is consistent with the occurence of peptidase C39 domain proteins in other mycoplasmas without apparent candidate double-glycine motif peptides (*Dirix et al., 2004*). This suggests another function for 0401; it could potentially import lipoyllysine or a lipoylpeptide and catalyze the transamidation of the lipoyl moiety onto the lipoyl-binding domain of PdhC. To account for the required lipoylation of PdhC in the absence of lipoyl transferases, this mechanism of uptake and transamidation of lipoyllysine is tentatively included in the model for JCVI-syn3A.

## Feasibility of permeative glycerol uptake

In the absence of a dedicated glycerol importer, passive permeative glycerol uptake is assumed. Using a permeation coefficient of 50 nm/s for glycerol through a phospholipid bilayer (*Raven, 1984*) and assuming an external glycerol concentration of 6 mg/l in SP4 medium (*Karr et al., 2012*), an upper limit on glycerol uptake 0.193 mmol gDW$^{-1}$ h$^{-1}$ is obtained for a 0.4 micron diameter spherical cell, assuming a density of 1.1 g/ml (*Bratbak and Dundas, 1984*) and a cellular water content of 72% (*Leblanc and Le Grimellec, 1979*), as in the biomass calculations. We note that the optimal FBA solution of our model demands 0.064 mmol gDW$^{-1}$ h$^{-1}$ glycerol uptake, so that this passive uptake is not growth-limiting, and the ability of the model cell to grow is not dependent on the exact permeation coefficient in vivo, which might not be the same as for the pure phospholipid bilayer.

## ATP costs of protein turnover

The AAA+ protease Lon (*lon*/0394) is assumed to be the main protease for turnover in JCVI-syn3A. Lon from *E. coli* has been found to decompose proteins of different length and composition into oligopeptides of 10–20 residues, consuming an amount of ATP per hydrolyzed peptide bond in the process that depends on pH and ADP concentration but not on the substrate (*Menon et al., 1987*).

Lon from *Salmonella typhimurium* reportedly has a similar ATP/peptide bond stoichiometry to *E. coli* Lon (*Menon et al., 1987*), suggesting that the stoichiometry for *E. coli* can be used to approximate the stoichiometry in other species as well. At pH 7.5 (approximate cytosolic pH of *M. mycoides capri* in neutral medium (*Benyoucef et al., 1981a*)) and 0.5 mM of the inhibitor ADP (comparable to the concentration in *M. mycoides capri* LC Y (*Mitchell and Finch, 1979*)), Lon consumes 9 ATP per peptide bond (*Menon et al., 1987*). Assuming breakdown to oligopeptides of ~15 residues yields an ATP expense of 225 ATP per protein of 385 residues in the model. The resulting oligopeptides would then be further broken down to individual amino acids by the other peptidases without expense of energy.

## Derivation of Na$^+$ and K$^+$ active transport reactions

Potassium and sodium transport in *M. mycoides capri* is known to involve several functionalities. Specifically, from studies on *M. mycoides capri* PG3, three functionalities have been proposed (*Benyoucef et al., 1982a*; *Benyoucef et al., 1982b*):

- A K$^+$ uniport functionality that can only import K$^+$ until its chemical potential is in equilibrium.
- A K$^+$/Na$^+$ antiport functionality that consumes ATP and is able to concentrate K$^+$ inside the cell (beyond equilibrium).
- An Na$^+$/H$^+$ antiport functionality that extrudes Na$^+$, powered by the proton gradient established by the proton-extruding ATPase.

The two K$^+$ transport functionalities compare well to the properties of the KtrAB K$^+$ import system (*natA*/0685 and *trkA*/0686) as characterized in *Vibrio alginolyticus* (*Tholema et al., 2005*). KtrB alone was found to slowly import K$^+$ (in an Na$^+$-independent manner), and also import Na$^+$. The complete KtrAB system, in contrast, was found to import K$^+$ two orders of magnitude faster and in an Na$^+$-dependent manner, but no longer imported Na$^+$ (extrusion not studied). These findings would be consistent with the KtrAB system providing both the passive K$^+$ uniport functionality as well as the active K$^+$/Na$^+$ antiport functionality. We thus include an ATP-consuming K$^+$/Na$^+$ antiport reaction catalyzed by KtrAB (*natA*/0685 and *trkA*/0686) in the model (but not the uniport reaction, since it does not participate in yielding the concentrated K$^+$ level in the cytoplasm). We also include an Na$^+$/H$^+$ antiport reaction (without any gene assignment for the time being).

## Sensitivity analysis

As the constraint parameters used in our model stem from other mycoplasmas, the sensitivity of the model to variations in the parameters is of particular interest. *Appendix 1—figure 2* shows the doubling time obtained for varying different constraints around the respective value chosen in the model (indicated by a blue filled circle in each plot). In addition, we calculated elasticities of the doubling time with respect to each constraint per the formula:

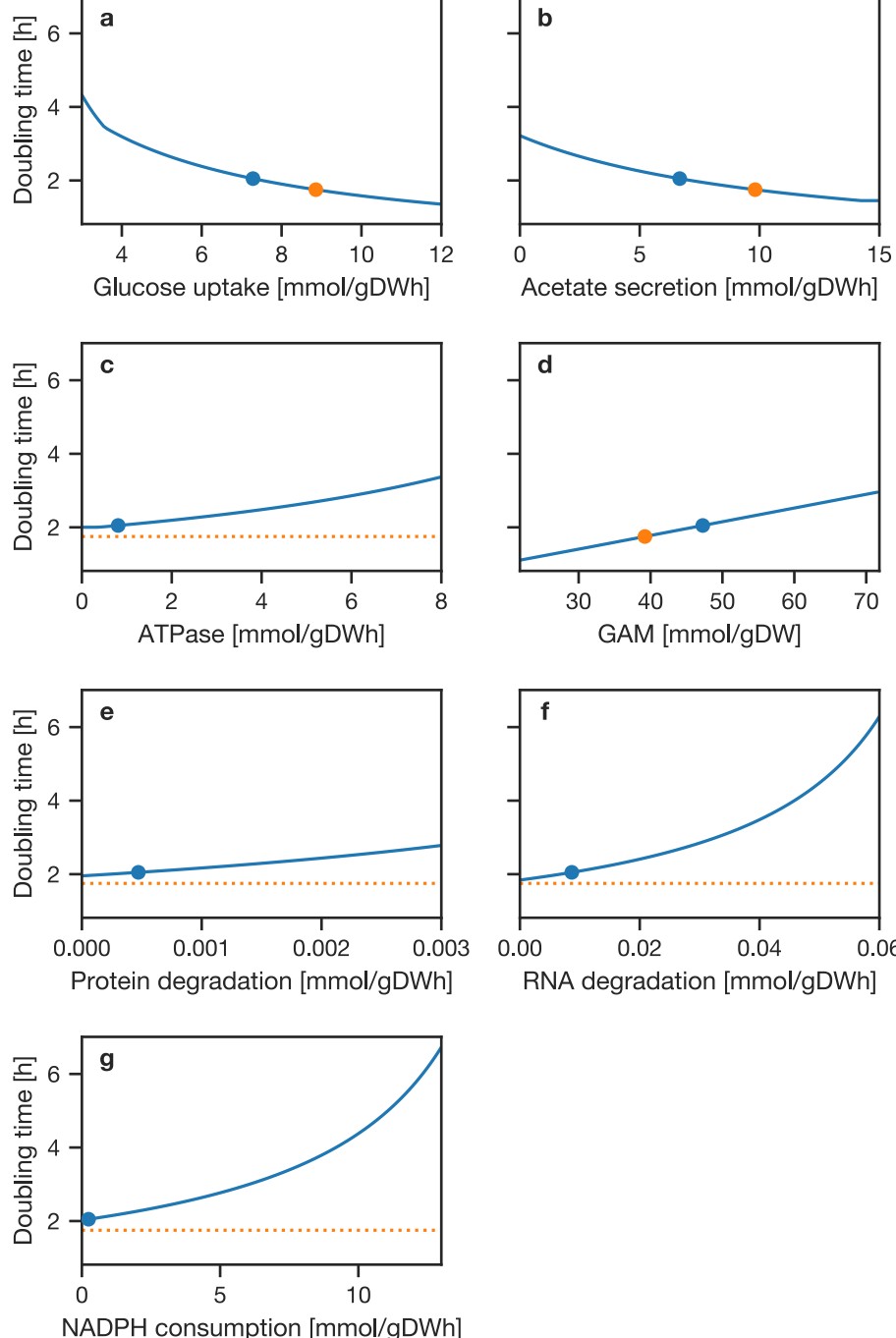

**Appendix 1—figure 2.** Sensitivity analysis of model doubling time with respect to model constraints. In each panel, the stated parameter was varied over the indicated range and the model doubling time calculated while keeping all other constraints constant. (**A:**) Maximal glucose uptake. (**B:**) Maximal acetate secretion. (**C:**) ATPase ATP cost. D: GAM ATP cost. (**E:**) Protein degradation rate. (**F:**) RNA degradation rate. (**G:**) Imposed NADPH consumption. The blue circle marks the value used in the FBA model and resulting doubling time; the orange circle indicates the parameter that would yield the experimental doubling time. If there is no value of the parameter which would yield the experimental doubling time, a horizontal line is plotted.

$$E_c t_{\mathrm{d}}\big|_{c=c_0} = \frac{c_0}{t_{\mathrm{d}}\big|_{c=c_0}} \frac{\partial t_{\mathrm{d}}}{\partial c}\bigg|_{c=c_0} \qquad (4)$$

for a given constraint $c$ at its reference value $c_0$ (i.e. the value used in the model) while keeping all other constraints fixed at their reference values. These elasticities describe the ratio in relative changes of growth rate and constraint for varying the given constraint around its reference value. (I. e. , a (locally constant) elasticity of 0.1 would imply a growth rate change by 1% for a change in the constraint by 10%.)

Low elasticities ($\leq 0.1$) are obtained for the ATPase efflux (0.03), protein degradation (0.04) and RNA degradation (0.1), that is the three parameters that together determine the model NGAM– demonstrating the model's relative insensitivity to variations in these parameters. A moderate elasticity of 0.37 is observed for acetate secretion, while higher elasticities of 0.80 and 0.86 are found for glucose uptake and the total GAM, respectively (ignoring signs). While this elasticity suggests a considerable sensitivity of model growth rate on the estimated GAM, we note that nearly half of the GAM (21.54 mmol gDW$^{-1}$) can be accounted for by macromolecular synthesis cost, specifically protein synthesis cost (~21.2 mmol gDW$^{-1}$ in the model). The protein synthesis cost depends on the ATP cost per amino acid in the proteome (~4 ATP/amino acid) which is deemed conserved, and on the protein fraction in the cellular dry mass. The protein dry mass fraction is based on that of the natural *M. mycoides capri*, which is assumed to provide a reasonable first approximation for JCVI-syn3A (see also Further model details and justification in Appendix 1). The quantifiable fraction of the GAM thus carries considerably less uncertainty than the non-quantifiable fraction.

It is thus instructive to consider the two contributions to the GAM in our model separately, by decomposing the total GAM ($G$) into its quantifiable ($G_q$) and nonquantifiable ($G_{nq}$) parts, $G = G_q + G_{nq}$; and evaluating the elasticity with respect to $G_q$ and $G_{nq}$ separately. We have:

$$\frac{\partial t_d}{\partial G_q} = \frac{\partial t_d}{\partial G} \times \frac{\partial G}{\partial G_q} = \frac{\partial t_d}{\partial G} \tag{5}$$

which analogously holds for $G_{nq}$. Furthermore, since $G_0 = G_{q,0} + G_{nq,0}$:

$$t_d\big|_{G=G_0} = t_d\big|_{\substack{G_q = G_{q,0} \\ G_{nq} = G_{nq,0}}} \tag{6}$$

where as in Equation 11, the subscript 0 denotes the value adopted in the model for each quantity. Summing the elasticities with respect to $G_q$ and $G_{nq}$ then yields:

$$E_{G_q} t_d\big|_{\substack{G_q = G_{q,0} \\ G_{nq} = G_{nq,0}}} + E_{G_{nq}} t_d\big|_{\substack{G_q = G_{q,0} \\ G_{nq} = G_{nq,0}}}$$

$$= \frac{G_{q,0}}{t_d\big|_{\substack{G_q = G_{q,0} \\ G_{nq} = G_{nq,0}}}} \frac{\partial t_d}{\partial G_q}\bigg|_{\substack{G_q = G_{q,0} \\ G_{nq} = G_{nq,0}}} + \frac{G_{nq,0}}{t_d\big|_{\substack{G_q = G_{q,0} \\ G_{nq} = G_{nq,0}}}} \frac{\partial t_d}{\partial G_{nq}}\bigg|_{\substack{G_q = G_{q,0} \\ G_{nq} = G_{nq,0}}}$$

$$= \frac{G_{q,0}}{t_d\big|_{\substack{G_q = G_{q,0} \\ G_{nq} = G_{nq,0}}}} \frac{\partial t_d}{\partial G}\bigg|_{\substack{G_q = G_{q,0} \\ G_{nq} = G_{nq,0}}} + \frac{G_{nq,0}}{t_d\big|_{\substack{G_q = G_{q,0} \\ G_{nq} = G_{nq,0}}}} \frac{\partial t_d}{\partial G}\bigg|_{\substack{G_q = G_{q,0} \\ G_{nq} = G_{nq,0}}}$$

$$= \frac{G_{q,0} + G_{nq,0}}{t_d\big|_{\substack{G_q = G_{q,0} \\ G_{nq} = G_{nq,0}}}} \frac{\partial t_d}{\partial G}\bigg|_{\substack{G_q = G_{q,0} \\ G_{nq} = G_{nq,0}}}$$

$$= \frac{G_0}{t_d\big|_{G=G_0}} \frac{\partial t_d}{\partial G}\bigg|_{G=G_0}$$

$$= E_G t_d\big|_{G=G_0}$$

That is the elasticity with regard to the total GAM is equal to the sum of elasticies with regard to its quantifiable and non-quantifiable fraction. This yields elasticities of the model doubling time with

respect to $G_q$ and $G_{nq}$ of 0.40 and 0.46, respectively. The former, then, describes the elasticity with respect to a parameter where comparatively less uncertainty is in fact expected; from the overall high elasticity of 0.86 with respect to the GAM, only 0.46 fall to a quantity of significant uncertainty.

Regarding the high elasticity of 0.8 with regard to glucose uptake rate, we note that since the carbon source uptake directly determines the overall ATP production in the model, a significant sensitivity of model growth rate with regard to that uptake rate is in fact to be expected. For example, we observe a growth rate elasticity of 1.01 with respect to glucose uptake rate (10 mmol gDW$^{-1}$ h$^{-1}$) for the *E. coli* model iJO1366 (*Orth et al., 2011*). For our model, where glucose uptake rate and other parameters are adopted from other organisms and models, this means that the growth rate predicted should hence be more considered a qualitative prediction, and provisional until corresponding measurements become available for JCVI-syn3A.

Finally, we note that the only consumer of reduction equivalents in the model is currently the ribo-dinucleotide reductase system (RNDR). There are however likely other significant demands for reduction equivalents that currently can not be quantified, like repair of oxidative damage of proteins (cysteine oxidation to disulfides). To probe the possible impact of this demand, we introduced an artificial NADPH oxidation reaction for testing purposes and calculated the doubling time as a function of imposed NADPH consumption (panel G), as NADPH production through GAPDP diverts flux from the ATP-producing GAPD/PGK branch. As the NADPH consumption through this artificial oxidation reaction is absent/zero in our model outside of this test, the associated elasticity at this point is necessarily also zero and thus not informative. Similarly, the elasticity has very small positive values for small nonzero values of NADPH consumption (e.g. 0.0005 for an imposed NADPH consumption of 0.01 mmol gDW$^{-1}$ h$^{-1}$). It is thus of interest to consider the elasticity at higher NADPH consumption values. At 1.0 mmol gDW$^{-1}$ h$^{-1}$, the observed elasticity is still low with 0.06. The in silico growth rate at this value is also still 2.14 hr, demonstrating that while we cannot accurately capture NADPH demand in our model, the model can sustain a certain level of NADPH demand without significant impact on growth rate.

## Tolerable protein dilution as possible cause of time-delayed gene disruption lethality

As discussed in Section 'Transposon mutagenesis experiments probe in vivo gene essentiality' in the main text, a gene can in principle appear quasi-essential in the transposon mutagenesis analysis if its disruption is lethal in principle (rather than just causing a growth defect) but will take time to take effect. One reason for such time-delayed lethality of a gene knockout could be an initial gene product abundance that is high enough to sustain cellular demands over several generations (during which the protein concentration is diluted by half at each cell division). Specific candidates for such a scenario would be genes identified as quasi-essential where not only the FBA model, but also biological context would otherwise strongly suggest the gene to be essential. As mentioned in Interpretation of individual gene essentialities in the main text, this is the case for the genes *nrdE/*0771 and *nrdF/*0773 (subunits of RNDR) and 0214 (PGPP).

The capacity of the cell to tolerate disruption of a gene and maintain the required metabolic fluxes with an ever-decreasing protein abundance (twofold dilution at cell division, no new synthesis of functional protein due to gene disruption) could be estimated by the ratio of the protein abundance based flux constraint $V_{\max}$ to the reaction flux required by the model. The estimated $V_{\max}$ for PGPP is ~2.4 mmol gDW$^{-1}$ h$^{-1}$, assuming an initial cellular abundance of ~170 copies. (I.e., assuming an abundance equal to the average copy number across the proteome; PgpA was not quantified in the proteomics experiment.) The PGPP flux required by the optimal FBA solution is 0.028 mmol gDW$^{-1}$ h$^{-1}$, that is ~100 times lower. Assuming twofold dilution of the initial PgpA copy number at each cell division would then allow the cell to divide 6–7 times before the flux limit would not suffice anymore to sustain the flux required for optimal growth. that is for these first 6–7 generations, the cell would not experience any loss of fitness yet. A caveat lies in the turnover number $k_{cat}$ found for PGPP, which stems from *E. coli* and a slightly different substrate (phosphatidate instead of phosphatidylglycerophosphate).

A similar argument could be made for RNDR. The total flux through all four RNDR reactions required by the model is 0.005 mmol gDW$^{-1}$ h$^{-1}$. Assuming a similar $k_{cat}$ for all four dinucleotide substrates would yield a total flux limit $V_{\max}$ of 0.378 mmol gDW$^{-1}$ h$^{-1}$, ~70 times higher than the

required flux; assuming simple dilution, this would then allow for six cell divisions before the flux limit does not support the optimal flux anymore, similar to the situation for *pgpA*. It should be noted however that RNDR is essential in the model not as a source of deoxynucleotides (which can be obtained from deoxynucleosides as well), but rather as the only consumer of NADPH (produced by *folD/*0684/MTHFD). Accordingly, the in silico flux through the RNDR reactions equals the NADPH production via MTHFD, and amounts to 12% of the total dA/G/CDP production flux of 0.038 mmol gDW$^{-1}$ h$^{-1}$, with the bulk of the flux in the model instead carried by the deoxynucleoside/-mononucleotide kinases. (dTDP cannot be made through RNDR and thus is not relevant for the RNDR flux limit). In vivo, other NADPH sinks should remove the need to use RNDR for NADP regeneration, so that the in vivo flux should only be determined by the deoxynucleotide synthesis partioning between RNDR and the deoxynucleoside kinases for deoxyadenosine, -guanosine and -cytidine. The deoxynucleoside kinases 0330 and 0382 are running not too far from their $V_{\max}$ limit in the model: The reaction DADNK (accounting for 70% of the flux through *dak1/*0330 and *dak2/*0382) has a $V_{\max}$/FBA flux ratio of 2.4; as elaborated in the main text, running enzymes close to $V_{\max}$ creates unstable conditions in case of changing substrate concentrations. RNDR might then be required in vivo to take flux load off the deoxynucleoside kinases, which would explain why it was still essential even if not needed as an NADPH sink. At the same time, since the deoxynucleoside kinases are capable of carrying most of the flux, the in vivo RNDR flux can be expected to not significantly exceed the one observed in the model. Thus, the scenario of flux capacity buffering against enzyme dilution as a cause for the apparent quasi-essentiality of the RNDR genes *nrdE* and *nrdF* seems conceivable even if more definitive statements cannot be made.

## Interpretation of YgfA essentiality amidst a quasi-essential folate metabolism

5-formyl-THF cyclo-ligase (*ygfA/*0443) is the only essential folate-related enzyme whereas all other genes in folate cycle and uptake are merely quasi-essential in vivo. This seems plausible at first given that 5-formyl-THF is a known inhibitor of folate-related enzymes (*Stover and Schirch, 1992*), rendering prevention of its buildup by YgfA an important metabolic damage repair function. However, for JCVI-syn3A a paradox arises: The potential inhibition targets for 5-formyl-THF, namely the other folate enzymes, are themselves only quasi-essential—raising the question why preventing buildup of 5-formyl-THF should then be essential. Unlike a knockout of any other individual folate-related enzyme, 5-formyl-THF buildup as a result of YgfA knockout would inhibit several folate-related enzymes at once. However, if this were the reason for the essentiality of YgfA, the uptake protein FolT should be essential too. In addition, an individual knockout of GlyA (GMHT, *glyA/*0799) or FolD (MTHFD/MTHFC, *folD/*0684) would also in effect disrupt the complete folate cycle. This suggests that 5-formyl-THF buildup might have detrimental effects outside its known range of interactions with folate-related enzymes—or that there are essential folate-related genes in JCVI-syn3A yet to be identified. It should in this context be noted however that *ygfA* is classified as essential in the transposon data with a probability of only 0.58, compared to a probability of being quasi-essential of 0.42—rendering the classification much less certain than for nearly all other genes classified as essential. As an aside, the proposed folate uptake gene *folT/*0822 is the only *ecfS* gene that is quasi-essential rather than fully essential—lending further support to its assignment as *folT*.

## Proteomics derived constraints

Genome-scale metabolic reconstructions can be used to predict flux distributions through the metabolic network. Metabolic reconstructions represent the network topology of all metabolic reactions that can occur in a given organism. The network topology imposes a linear dependence between fluxes through the various reactions based on the stoichiometry and connectivity of the metabolites involved, assuming steady state conditions. Typically the number of reactions (variables) is higher than the number of metabolites (and hence mass balance equations) which results in an under-determined system of linear equations for reaction fluxes. Additional constraints based on the environment (growth medium), experimentally determined housekeeping requirements (NGAM *etc.*) and thermodynamics (reversibilities) further reduce the possible flux values through individual reactions (see Section 'Metabolic reconstruction' in the main text). This solution space of feasible fluxes can be

further reduced based on proteomics constraints (*Labhsetwar et al., 2013*; *Labhsetwar et al., 2017*). Proteomics constraints are applied using the Gene-Protein-Reaction (GPR) rules which list all genes whose products catalyze a particular reaction. The copy number of proteins available in the cell combined with their catalytic capacity gives an upper limit on the flux possible through a particular reaction by

$$V_{\mathrm{max}} = \frac{1000\, N_{\mathrm{prot}}\, k_{\mathrm{cat}}}{N_{\mathrm{A}}\, m_{\mathrm{cell}}}. \tag{7}$$

Here, $V_{\mathrm{max}}$ is the upper bound on the flux through a particular reaction with units of mmol gDW$^{-1}$ h$^{-1}$, $k_{\mathrm{cat}}$ is the turnover number of a given enzyme (in 1/hr), $N_{\mathrm{A}}$ is Avogadro's constant, $m_{\mathrm{cell}}$ is the dry weight of a single JCVI-syn3A cell, and $N_{\mathrm{prot}}$ is the copy number of a given protein per cell. For reactions catalyzed by a protein complex, the lowest copy number among the measured subunits is used for $N_{\mathrm{prot}}$, whereas in the case of reactions which can be catalyzed by multiple enzymes independently, the sum of the copy numbers of all isozymes is used for $N_{\mathrm{prot}}$. In such cases it is required that all isozymes are measured to estimate a $V_{\mathrm{max}}$.

Protein copy numbers are obtained from quantitative proteomics (see Section 'Mass Spectrometry Based Proteomics' in Materials and methods). The median of relative abundances obtained from the three replicates of time point 1 (exponential growth phase) is used. Relative abundances are converted to absolute abundances (cellular copy numbers) per the total protein biomass fraction (see Section 'Biomass composition and reaction' in main text) and protein average molecular weight as reconstructed from the proteomics data. This yields an estimated total number of ~77,000 protein molecules per cell. $k_{\mathrm{cat}}$ values are obtained from BRENDA (*Scheer et al., 2010*) with careful consideration for reaction substrate, physiological conditions of measurement and source species for the enzyme of interest, as presented in the following section.

For each EC number, the BRENDA database was queried for all available turnover data across all organisms. Since kinetic data is available for many different substrates, the results must be filtered to only include the natural substrates. The chemical names in the BRENDA database are not regular and can be specified under many synonyms. Thus, the Kyoto Encyclopedia of Genes and Genomes (KEGG) database (*Kanehisa et al., 2015*) was used to specify the preferred names of the substrates and products. To derive a mapping between all possible synonyms and the preferred chemical name, the set of all natural product and substrate names from BRENDA were compiled for each EC number. A Python script was used to efficiently construct the synonym to preferred name mapping by displaying the KEGG chemical names prefixed with a number and allowing the user to indicate which of the compounds listed in the BRENDA entries are equivalent. After regularizing the chemical names using the synonym map, and filtering out entries which do not correspond to the natural substrates/products, a unique entry must be chosen for each EC number.

To choose the most appropriate database entry, we first define an ordering function

$$\omega(E) = [e, d, m, t, p, c], \tag{8}$$

over the database entries $E$, where the quantities in the list are the selection criteria in decreasing order of importance. First, $e$ is one if the entry's commentary indicates that the temperature of the measurement was performed above 50 ℃, allowing for extremophiles to be filtered out. Second, the phylogenetic distance, $d$, between *Mycoplasma mycoides* and the entry organism is considered. The distance utilizes the dataset available at the Interactive Tree of Life (iTOL) (*Letunic and Bork, 2016*) which was first described by *Ciccarelli et al. (2006)*. For organisms missing from the (*Ciccarelli et al., 2006*) data set, if there is a species available with the same genus, that distance is used. Otherwise, the species is classified into Gram-positive or gram-negative bacterium, eukaryote, or archaeon and assigned the largest distance to a member of those groups. Third, the commentary field is searched for the strings 'wild-type' and 'mutant'. Entries containing 'wild-type' but not 'mutant' ($m = 0$) are preferred over entries with neither string present ($m = 1$). Then entries containing 'mutant' are considered, with entries containing both 'wild-type' and 'mutant' ($m = 2$) preferred over those with 'mutant' alone ($m = 3$). Fourth, the commentary field is searched for temperature and the entry closest to 30 ℃ preferred ($t = |T - 30℃|$). If no temperature is listed, it is treated as a 5 ℃ difference. Fifth, the commentary field is searched for pH ($p = |\mathrm{pH} - 7.0|$), preferring entries closest to pH 7.0. If no pH is listed, the entry is treated as having a difference of 0.1 units from pH 7.0. Finally, the

entry with the shortest commentary string is preferred ($c = length(commentary)$). Since longer comments are more likely to be for measurements taken at non-standard conditions, shorter comments are more likely to be appropriate for our purposes. The group of entries with the minimum value of the ordering function, $\omega(E)$, is then selected (lexicographically) for further processing.

At this point, if there is no unique best candidate in the selected group, the median turnover number among the remaining candidates is taken as $k_{cat}$. When there are an even number of values, the geometric mean is taken of the two middle values since it is possible that the two values could span multiple orders of magnitude. This prevents the larger value from dominating the median. The $V_{max}$ values used as well as the $k_{cat}$ entries used to compute them are available in *Supplementary file 7*, along with the reason that particular value was chosen. The column $V_{min}$ in *Supplementary file 7* refers to $V_{max}$ for the reverse reaction assuming the forward direction as shown in the reaction equation.

There were instances where substrates identified using KEGG did not match the substrate in the model reaction. In these cases, $k_{cat}$ values were assigned manually following the above selection criteria. In addition, $k_{cat}$ measurements found during the literature survey for the metabolic reconstruction which were not available in BRENDA were used over the automated selections if they were better matches with respect to *Equation 8*. These manual changes are also listed in *Supplementary file 7*.

## Comparison of JCVI-syn1.0 and JCVI-syn3A

It is instructive to consider the differences between JCVI-syn1.0 and JCVI-syn3A. As outlined above, in regards to the biomass composition, the overall protein dry mass fraction is not expected to differ much, and the RNA content of *M. mycoides capri* is expected to provide a reasonable approximation for JCVI-syn3A as well. An important change in the biomass composition is of course the smaller size of the JCVI-syn3A genome (by a factor of two); the different genome is also expected to affect the proteome composition, which in the JCVI-syn3A reconstruction is given directly by the JCVI-syn3A proteomics data.

When considering the functional content of the genome in JCVI-syn3A versus the one in JCVI-syn1.0, it is informative to consider their respective origins. While the genes encoded by the near minimal bacterial cell JCVI-syn3A are a subset of the gene content of both the naturally occurring bacterium *M. mycoides capri* and JCVI-syn1.0 (the wild type organism encodes a few genes deleted from JCVI-syn1.0 to reduce possible pathogenicity and to facilitate genome synthesis), the minimized and full-size genomes were evolved for life under very different conditions. The naturally occurring bacterium evolved for life in both the upper and lower respiratory systems of goats. Initial infection would take place in the upper respiratory system, which has different epithelial cells and a lower temperature than would be encountered in the lower respiratory system, that is the lungs, which are about 37°C. As an example of those evolutionary forces, DNA methylation analysis of the closely related goat pathogen *Mycoplasma capricolum* showed 315 GATGA sites with adenine methylated at the first A in cells grown at 30°C and a different 15 sites when the cells were grown at 37°C (data not shown). We assume the methylation machinery that causes this is an adaptation to enable the bacterium to grow in the two different goat milieus. Wild type *M. mycoides capri* likely has similar methylation machinery; however when we designed the JCVI-syn3A genome, the only environment we wanted the organism to grow in was SP4 media. Those DNA methylases are no longer present. Because of their high mutation rate and apparent evolutionary pressure to minimize their gene content, mycoplasmas tend to lose genes that are not essential for life in their preferred host (*Maniloff, 1992*; *Sirand-Pugnet et al., 2007*). The entire gene content of wild type *M. mycoides capri* is optimal for life in a goat. Removal of any gene would probably result in that mutant failing to survive long in an infected animal or herd of animals. In JCVI-syn3A, the evolutionary pressure, which of course is channeled through the choices it JCVI designers made in choosing its gene content, is quite different. It was not based on ideal growth, but rather on coming up with a genome that had as few genes as possible, but could still divide in two hours or less in SP4 growth media. Gene content is based on an artificial criterion of minimization rather than evolutionary advantage.

To elucidate the impact of genome reduction on metabolic capabilities, *Supplementary file 8* lists known reactions expected to be present in JCVI-syn1.0 but absent in JCVI-syn3A after genome

minimization. The format is the same as in *Supplementary file 4*. The list contains 53 reactions connected to 48 removed genes of metabolic function: 20 genes (17 reactions) pertain to uptake of alternative sugars and conversion to glycolytic intermediates, demonstrating the wider scope of carbon sources JCVI-syn1.0 can use to generate energy. The removal of five genes in nucleotide metabolism further tightened the nucleotide precursor requirements in JCVI-syn3A, which cannot utilize uridine and thymine anymore and also requires adenine (in free base or nucleoside form) since guanine cannot serve as precursor for all purines anymore. Seven genes in JCVI-syn1.0 enable the degradation of certain amino acids as further ATP sources.

Depending on the information available, some of the listed reactions are described in a generic fashion. This list is not meant to be exhaustive in the sense of providing a complete reconstruction for JCVI-syn1.0, but contains reactions encountered during the reconstruction process for JCVI-syn3A and/or that could easily be inferred from the JCVI-syn1.0 genome annotation.

