## [Decision Letter]

Thank you for submitting your article "Essential Metabolism for a Minimal Cell" for consideration by *eLife*. Your article has been reviewed by Naama Barkai as the Senior Editor, a Reviewing Editor, and two reviewers. The following individual involved in review of your submission has agreed to reveal his identity: Ron Milo (Reviewer #1).

The reviewers have discussed the reviews with one another and the Reviewing Editor has drafted this decision to help you prepare a revised submission.

Summary:

The manuscript by Breuer et al., addresses the important question of a minimal cell and provides a valuable resource in the field of synthetic biology. The extensive manual curation, the reconstruction of the metabolic network, and its characterization in silico and with usage of proteomics data provide a valuable foundation for studying the features of a minimal cell. The experiments and hypotheses suggested in the discussion are particularly appreciated. While the presented work provides a knowledge base resource to the community, the predictions formulated by the metabolic model should be described in a more nuanced way.

Essential revisions:

One of the main concerns deals with the lack of organism-specific incidental to the model predictions, namely: biomass composition, substrate uptake rates and secretion rates. While it is challenging to obtain such level of detail for a minimal organism, and the employed data pertain to the parental strain, it is important to explain how the model represents Syn3.0A and not Syn1.0. Although a comparison of the proposed and existing models has been attempted, the authors should provide a direct comparison of the metabolic capabilities between the original Syn1.0 strain and the reduced Syn3.0A to better delineate the latter from the former. The authors should clearly describe the approach (automated (how) or manual) used in comparing JCVI-syn3.0A model and iJW145.

A related concern is the approach used to infer internal composition from the growth media, since the media composition is not representative of the cytosol concentration. The authors should clarify the logic and approach used; if already published, the followed procedure should be appropriately referenced.

A third concern is the support of the claim that the model correctly predicts gene essentiality and could help further reduce the model namely through double knock-out simulations. Since Table S6 describes model accuracy, it should be moved to the main text. Given the availability of the data on categorization of the genes in 3 categories (essential, non-essential and quasi-essential), the authors should calculate the Matthews Correlation Coefficient: (1) assuming the quasi-essential genes to be essential and (2) assuming quasi-essential genes to be non-essential. The obtained metric should be compared to other Mollicute metabolic models.

The authors should verify the way in which the sensitivity analysis is conducted. A reasonable way to do the sensitivity analysis is to investigate what percentage of growth rate change is found for a given percentage of change in a parameter. Such an analysis may imply very strong variation in contrast to what is stated by the authors. In particular, the effect of the non-growth associated maintenance (NGAM) should be particularly re-investigated since the assumption that the NGAM would be strictly limited to the ATP efflux seems likely to be an under-estimation as many cellular processes may not be described.

The authors should use correct terminology when describing predictions from flux balance analysis throughout the manuscript, as it predicts yield; the conversion to growth rate is dictated by what the authors assume for uptake rate which if not measured but is a gross proxy, as indicated above.

[Editors' note: further revisions were requested prior to acceptance, as described below.]

Thank you for resubmitting your work entitled "Essential Metabolism for a Minimal Cell" for further consideration at *eLife*. Your revised article has been favorably evaluated by Naama Barkai (Senior Editor), a Reviewing Editor, and two reviewers.

The manuscript has been improved but there are some remaining issues that need to be addressed before acceptance, as outlined below:

*Reviewer #2:*

We wish to thank the authors for taking time to address our previous comments. The addition of the whole metabolic map for JCVI Syn3.0 is integrative and should help readers understand the metabolism of the minimal cell. We appreciate that the model predictions were nuanced according to the available data (i.e. steady-state fluxes, in silico gene knockout mapping, etc.). While these comments were properly addressed, a second view of the (lengthy) paper yielded new comments:

Abstract:

The Abstract is overly concise for a paper that spans 76 pages including references. In general, the conclusions exposed in the abstract should be extended and "98% of enzymatic reactions supported by annotation." This seems like a wrong and inflated number that does not reflect the uncertainty in the network. Excluding exchange reactions, biomass, growth-associated maintenance and gas diffusion (O2 and CO2) 36 reactions remain orphan, including many transports. This correction brings the percentage down to 85.6% when applying a denominator of 250 (again excluding the mentioned reactions above).

"The model agrees well with genome-scale in vivo transposon mutagenesis experiments" Perhaps provide a quantitative statement rather than a qualitative one.

"The genes in the reconstruction have a high in vivo essentiality or quasi-essentiality of 92% , compared to 79% in silico essentiality." While the results presented in supplementary table 2 are consistent with these two numbers, we find 76% in silico essentiality when testing the model provided for single gene deletions. Also, the in vivo essentiality of 92% is obtained when assuming quasi-essential genes as essential and should be specified or the number with only essential genes (68%) should be included or at least mentioned.

"The reconstruction is the starting point for studying the evolution of metabolic subsystems and analyzing the effects of introducing alternative pathways." If claiming this in the abstract perhaps include subsystems to the reactions in the model.

"Finally, the identification of 30 essential genes with unknown function will motivate the search for new biological mechanisms beyond metabolism." After reading through the entire paper multiple times I still cannot see where this number is fetched from, so I cannot imagine how a reader could get to it.

Rather than pointing to a very accurate model of Syn3.0, the abstract should demonstrate that despite being the smallest organism known to date, Syn3.0 has many unknown elements. While the amount of information provided in the paper is highly valuable for the community, presenting the reconstruction as a way to clearly identify the grey areas of the genome is key for the development and substantial efforts have been made in that sense through 1- a detailed manual curation process, 2- proteomic validation of enzyme expression, 3- in vivo gene essentiality. This sentence of the abstract should be of prime importance: "This comparison together with proteomics data yields new hypotheses on gene functions as well as suggestions for several further gene removals".

---

## [Author Response]

Summary:The manuscript by Breuer et al., addresses the important question of a minimal cell and provides a valuable resource in the field of synthetic biology. The extensive manual curation, the reconstruction of the metabolic network, and its characterization in silico and with usage of proteomics data provide a valuable foundation for studying the features of a minimal cell. The experiments and hypotheses suggested in the discussion are particularly appreciated. While the presented work provides a knowledge base resource to the community, the predictions formulated by the metabolic model should be described in a more nuanced way.

We appreciate that the comprehensive nature of the work on a minimal bacterial cell (metabolic model, manual curation, genome-scale gene essentiality and proteomics studies) was acknowledged and have endeavored to formulate the predictions from the metabolic model in a more nuanced way.

Essential revisions:1) One of the main concerns deals with the lack of organism-specific incidental to the model predictions, namely: biomass composition, substrate uptake rates and secretion rates. While it is challenging to obtain such level of detail for a minimal organism, and the employed data pertain to the parental strain, it is important to explain how the model represents Syn3.0A and not Syn1.0. Although a comparison of the proposed and existing models has been attempted, the authors should provide a direct comparison of the metabolic capabilities between the original Syn1.0 strain and the reduced Syn3.0A to better delineate the latter from the former. The authors should clearly describe the approach (automated (how) or manual) used in comparing JCVI-syn3.0A model and iJW145.

We should have made the point more clearly that in preparing the metabolic model for JCVI-syn3A the authors did use as much of the existing information on the new organism as possible at the time, especially with regard to its DNA genome size, proteomics data for amino acid composition in the biomass equation and its reduced metabolic capabilities. Unfortunately, this information was distributed throughout the manuscript and was not concentrated in one section. We have tried to make the point more clearly with a paragraph summarizing the differences between JCVI-syn1.0 and JCVI-syn3A in the Supplementary Text. The details are provided below. We do intend to continue to characterize more completely the biomass composition and determine uptake/secretion rates in the future. We thank the reviewers for acknowledging that this can be challenging for a minimal organism. We were planning to do a careful comparison once the defined media have been determined for both strains. At the moment we only have a semi-defined medium for JCVI-syn1.0.

We would like to stress that experiments to accurately determine cellular composition and dry weight are far from trivial (Beck et al., Processes 2018; doi:10.3390/pr6050038). We thus capitalize on the extensive experimental information available for *M. mycoides capri* in the current model and justify the adoption or modifications of this information at the appropriate places in the manuscript. We will eventually measure the cellular composition of JCVI-syn3A experimentally once a defined media for JCVI-syn3A is developed.

Metabolic capabilities:

To our knowledge, there is no metabolic reconstruction for JCVI-syn1.0; the model comparison in Section 'Comparison to *M. pneumoniae*' is between JCVI-syn3A and *Mycoplasma pneumoniae*, another important model mycoplasma. To quantify the metabolic differences between JCVI-syn1.0 and JCVI-syn3A, we added a Supplementary Spreadsheet with metabolic reactions that were removed in the minimization from JCVI-syn1.0 to JCVI-syn3A. Many of the genes removed during the minimization process have unknown function, but through the work presented in this manuscript we have been able to assign essentiality scores to the 91 remaining genes of unknown function in JCVI-syn3A. (See Figure 3 and Table 1 in manuscript.)

The spreadsheet contains 53 reactions connected to 48 removed genes of metabolic function: 20 genes (17 reactions) pertain to uptake of alternative sugars and conversion to glycolytic intermediates, demonstrating the wider scope of carbon sources JCVI-syn1.0 can use to generate energy. The removal of five genes in nucleotide metabolism further tightened the nucleotide precursor requirements in JCVI-syn3A, which cannot utilize uridine and thymine anymore and also requires adenine (in free base or nucleoside form) since guanine cannot serve as precursor for all purines anymore. Seven genes in JCVI-syn1.0 enable the degradation of certain amino acids as further ATP sources. This paragraph about comparison of metabolic capabilities has been added to the Supplementary Text of the manuscript.

Depending on the information available, some of the reactions in the sheet are described in a generic fashion. (The sheet lists reactions we came across during the reconstruction process for JCVI-syn3A and/or could easily infer from the JCVI-syn1.0 genome annotation. It is not meant to be exhaustive in the sense of providing a complete reconstruction for JCVI-syn1.0, but rather to provide the requested overview of identifiable differences in metabolic capabilities.) The sheet is in the same format as Supplementary file 3 (listing all model reactions).

We would like to stress that there may be genes of metabolic function but unknown substrates not only among the 91 unknown genes retained in JCVI-syn3A, but also among the genes removed from JCVI-syn1.0. As such, there could be additional metabolic functions among the deleted genes that currently cannot be precisely described yet. As outlined in Hutchison et al., 2016, the majority of the ~400 genes removed from JCVI-syn1.0 to JCVI-syn3.0 (just marginally smaller than JCVI-syn3A) were lipoproteins of unclear function, mobile elements/DNA restriction genes, or completely unassigned altogether. The deleted non-metabolic genes include amongst others e.g. a large number of genes used by *M. mycoides capri* to evade the immune system of its host. *M. mycoides capri* is a pathogen of goats.

Biomass composition:

In spite of the differences in metabolic capabilities between JCVI-syn1.0 and JCVI-syn3A, there are reasons to believe (as outlined below) that the overall biomass composition should be similar between the two organisms. The published experimental data and biomass composition for *M. mycoides capri* (the species to which JCVI-syn1.0 belongs) and several other mycoplasmas provided the starting point to estimate the biomass equation and coefficients for JCVI-syn3A.

Regarding the applicability of the *M. mycoides capri* composition, while the main text describes the biomass reconstruction in a summary fashion, the Supplementary Text already featured several sections on the topic of how certain features of the macromolecular composition of *M. mycoides capri* and other mycoplasmas were adapted to JCVI-syn3A (see subsections 'Applicability of macromolecular composition', 'Details on lipid composition', 'Genetic evidence for capsule production in JCVI-syn3A'). We have re-worded some of the text to make the approximations and justifications more clear. The most important aspects are the expected conservation of the protein dry mass fraction, which is the largest dry mass fraction among the macromolecules, the retention of specific genes deemed relevant for certain biomass components (in particular the polysaccharide capsule), accounting for the “ions and small molecules” dry mass fraction, and re-scaling of the DNA mass fraction to account for the shorter genome in JCVI-syn3A. Finally, we would like to emphasize that the average protein composition (average length of 386 aa and corresponding amino acid composition) in the model is directly based on JCVI-syn3A proteomics data.

A protein dry mass fraction of 40-60% is generally observed in bacteria, e.g. in different mycoplasmas (Razin, 1963) or *E. coli* (Bremer and Dennis, 2008). In particular, we note that *Acholeplasma laidlawii* PG8, in spite of having a 1.5-times larger genome than *M. mycoides capri* but a comparable cell size, shows a similar protein dry mass fraction (55% ) to *M. mycoides capri* (58% ). This suggests that the protein content in *M. mycoides capri* should provide a very good initial approximation for JCVI-syn3A as well.

The JCVI-syn3A proteomics data also gave an abundance estimate of the ribosomal proteins (average 340) which seemed reasonably consistent with the expected upper limit on ribosome number (670) coming from the RNA dry mass fraction of ~16% assuming all the RNA to be ribosomal. The presence of the same number of rRNA operons (2) in both JCVI-syn1.0 and JCVI-syn3A is also consistent with the assumption of comparable rRNA contents in the two organisms.

We note that the DNA content adopted from the natural *M. mycoides capri* comes out at approximately one chromosome in JCVI-syn3A (final number slightly adjusted from 5.0% to 5.5% to match exactly one chromosome).

The largest contributions to the small molecule/ion mass fraction come from potassium, phosphate, chloride, ATP and sodium, some of which have been measured directly in strains of *M. mycoides capri*. We used the direct measurement of potassium and sodium ions per cellular protein by Leblanc et al., (1979), the estimates of the pools of ATP and other nucleotides from Mitchell and Finch, 1979, and the estimate of the total amino acid pool (assuming each component to be proportional to the JCVI-syn3A proteomics data) from the *M. pneumoniae* study by Wodke et al., 2013. We feel that the final values are within the expected range and acceptable until the biomass composition can be measured directly for JCVI-syn1.0 and JCVI-syn3A.

Uptake/secretion rates:

A range of glucose uptake rates in *E. coli* and other bacteria has been measured by Fuhrer et al., 2005. For *E. coli* with a two hour doubling time, they reported a value of 7.8 mmol gDW^-1^h^-1^. We decided to use the value 7.4 mmol gDW^-1^h^**-1**^for the glucose uptake in JCVI-syn3A as it was taken from the *Mycoplasma pneumoniae* study by Wodke et al., 2013 which measured the uptake in rich medium; and *M. pneumoniae* had a comparable number of PtsG proteins based on the JCVI-syn3A proteomics data. Again, once the defined media have been determined for JCVI-syn1.0 and JCVI-syn3A, the uptake rates will be determined experimentally. A cautionary note has been added to the manuscript about the use of inherited uptake rates even when the high affinity glucose transporters are similar.

iJW145 model comparison:

We added a subsection 'Model comparison'. In brief, a script was used for a first round of automatic comparison, followed by manual curation to catch common or distinct reactions the script could not correctly classify.

2) A related concern is the approach used to infer internal composition from the growth media, since the media composition is not representative of the cytosol concentration. The authors should clarify the logic and approach used; if already published, the followed procedure should be appropriately referenced.

We have revised subsection 'Small molecules and Ions' (and the corresponding subsection in the Supplementary Text, "Details on small molecule pool composition") to clarify the approach taken. The media composition is not taken to be representative of cytosolic concentrations. Rather, the cited minimal media are used to infer what compounds are required by the cell in the first place: Any compound present in a minimal medium is (per definition of a minimal medium) required by the cell, and thus this compound (or its downstream product(s)) need to be included in the biomass. As had already been stated at the end of that subsection, the cellular dry mass fractions (and thus cytosolic concentrations) were taken from separate measurements or other models. The revised subsection should convey the procedure more clearly.

3) A third concern is the support of the claim that the model correctly predicts gene essentiality and could help further reduce the model namely through double knock-out simulations. Since Table S6 describes model accuracy, it should be moved to the main text. Given the availability of the data on categorization of the genes in 3 categories (essential, non-essential and quasi-essential), the authors should calculate the Matthews Correlation Coefficient: (1) assuming the quasi-essential genes to be essential and (2) assuming quasi-essential genes to be non-essential. The obtained metric should be compared to other Mollicute metabolic models.

We moved Supplementary file 6 to the main text (together with Figure 16—figure supplement 1) and also introduced another table with performance metrics (accuracy, sensitivity, specificity, Matthews Correlation Coefficient). We also merged the subsection 'Mapping in silico to in vivo essential genes' in the Supplementary Text into an expanded subsection, 'In silico gene knockouts and mapping to in vivo essentiality' in the main text.

As suggested by the reviewers, we calculated the Matthews Correlation Coefficient (MCC) assuming the in vivo quasi-essential genes (1) to be essential and (2) to be non-essential. As the identification of quasi-essential genes was crucial for the successful genome minimization in JCVI-syn3.0 (Hutchison et al., 2016), the former assumption might be biologically more relevant. For our model, the MCC comes out at ~0.59 in either case.

Table 1: Naming convention for calculation of Matthews Correlation Coefficient.

Exp. ¯ / ModelEssentialNon-essentialEssentialTrue positive (TP)False negative (FN)Quasi-essential“Weak false positive”“Weak false negative”Non-essentialFalse positive (FP)True negative (TN)

Table 2: Confusion matrix for JCVI-syn3A model (including quasi-essentials).

Exp. ¯ / ModelEssentialNon-essentialEssential1014Quasi-essential2214Non-essential012

While this does not amount to perfect agreement, we note that the quasi-essentials giving “weak false [positives/negatives]” in Table 2 above actually encompass the vast majority of false model predictions. Thus, in addition to the two limiting cases presented above, it is also instructive to consider the MCC for only those genes that can be classified as essential or non-essential in vivo, i.e. those genes that can be compared to the model classification without further assumptions. In that case, we get an MCC of 0.85, in line with only 4 genes being incorrectly predicted by the model (false negatives). This demonstrates that the lower MCC otherwise obtained really arises from the large number of quasi-essential genes included in the model, that are inherently difficult to describe in an FBA model. (Consider e.g. nucleic acid stabilization by polyamines: This is known to be an essential process in biology, yet the corresponding uptake genes are only quasi-essential in JCVI-syn3A.)

Similarly, it is of interest to consider one set of genes whose functionality is difficult to capture precisely based on the currently available information, namely the genes pertaining to uptake and utilization of amino acids (in free or peptide form). As already discussed in the manuscript, the in vivo essentiality of these genes is likely affected by their exact substrate profiles and maximal uptake rates. In line with this, 10 out of 14 weak false negatives in the model are accounted for by genes from amino acid utilization (uptake and peptidases). If we hence exclude all 12 related genes including these 10 weak disagreements, the MCC for case (1) increases to 0.72. (In case (2) the MCC decreases to 0.46, as the removed quasi-essentials would have been classified as true negatives here – so the agreement worsens by excluding "true negatives" that in the actual data are weak false negatives. The MCC increase in case (1) thus seems more informative.)

Table 3: Confusion matrix for the M. genitalium metabolic submodel in Karr et al., 2012, assuming any in silico knockout with nonzero growth rate to be nonessential. Treating knockouts yielding < 0.01*WT growth rate as essential leads to reassignment of two false negatives as true positives.

Exp. ¯ / ModelEssentialNon-essentialEssential7735Non-essential822

Among other Mollicutes with published metabolic reconstructions, *Mycoplasma genitalium* is the only one for which comprehensive experimental essentiality data has been published to our knowledge (Glass et al., 2006). Comparing in silico gene essentiality of the *M. genitalium* metabolic submodel used in the whole-cell model of Karr et al., 2012) to the Glass et al., 2006 data set yields an MCC of 0.35 (0.37 if we soften the essentiality criterion to any in silico knockout yielding < 0.01*WT growth rate). Our MCCs calculated above thus compare very favorably to the *M. genitalium* model.

We note that to date, no single published *M. pneumoniae* transposon mutagenesis data set is comprehensive enough to be used as a whole organism reference for gene essentiality. The numbers reported for essentiality prediction accuracy by the *M. pneumoniae* metabolic model (Wodke et al., 2013; Table III columns ii and iii) yield MCCs of 0.84 and 0.91 but are based on ortholog-based comparison to the *M. genitalium* data cited above; with an *M. pneumoniae* transposon library only screened in case the *M. genitalium* data was in disagreement with the model.

Finally, we would like to stress that the suggestions for gene removal experiments in subsection 'Targeted gene removal experiments' only involve the subset of true negative predictions, i.e. genes found to be non-essential in both model and experiment. These suggestions are therefore not affected by any mispredictions. The role of the model in this context is not to predict possible removals independently from the experimental data, but rather to suggest gene knockouts that might be possible simultaneously: While the transposon mutagenesis study could only probe individual gene essentialities, all of the true negative predictions can be knocked out simultaneously in silico with limited impact on growth rate. (The in-silico doubling time reported in the original submission of 11 h when simultaneously knocking out MMSYN1_0330, MMSYN1_0382 and MMSYN1_0227 arose from an NADH redox balance issue that we now corrected by adding the experimentally known capability of pyruvate secretion to the model.)

We would like to clarify that the in silico double knockouts reported at the end of subsection 'In silico gene knockouts and mapping to in vivo essentiality' were not meant to yield suggestions for gene removal experiments, but were meant to probe model behavior in general; they were performed for all in silico non-essential genes, not just for the true negative predictions discussed in subsection 'Targeted gene removal experiments' The addition of a pyruvate secretion reaction to the model mentioned above changed the double knockout results with respect to the original submission: Double knockouts of lactate dehydrogenase (MMSYN1_0475) and any gene along the acetate fermentation branch are not lethal anymore. This leaves the previously reported double knockout of both amino acid permeases as the only synthetic lethality in the model.

4) The authors should verify the way in which the sensitivity analysis is conducted. A reasonable way to do the sensitivity analysis is to investigate what percentage of growth rate change is found for a given percentage of change in a parameter. Such an analysis may imply very strong variation in contrast to what is stated by the authors. In particular, the effect of the non-growth associated maintenance (NGAM) should be particularly re-investigated since the assumption that the NGAM would be strictly limited to the ATP efflux seems likely to be an under-estimation as many cellular processes may not be described.

We have done the sensitivity analysis in a more rigorous fashion by calculating elasticities, as suggested by Ron Milo, in subsection 'Macromolecules and amino acids', which is equivalent to the ratio of percentage changes in model growth rate and changed parameter. This does yield a high elasticity of 0.8 for the glucose uptake rate, i.e. a change in glucose uptake rate by 10% would yield a growth rate change of 8%. (The elasticity with respect to the GAM is similarly high with 0.86, but of this total elasticity, 0.4 falls to the rather well-known macromolecular synthesis cost; only 0.46 fall to the more uncertain non-quantifiable fraction.)

In light of this, we have revised the beginning of subsection 'Steady state fluxes' (results for steady state fluxes) to emphasize that while the chosen constraints happen to reproduce the experimental growth rate, this agreement is sensitive to the model parameters (in particular glucose uptake rate) and hence, the growth rate prediction should be seen as provisional. As also noted in that revised paragraph, our subsequent gene essentiality analysis is not affected by the uncertainty in the predicted growth rate.

Rather than qualifying every reference in the manuscript to the model reproducing the experimental growth rate, we removed most of these references as this is not a main aspect of the work anyway.

Regarding the NGAM, we would like to note that the growth rate-independent energy expenses in the model not only encompass the proton efflux through ATPase but also turnover of protein and RNA. The lower bounds on protein and RNA degradation imply that a constant part of the protein/RNA synthesis flux is routed through protein/RNA degradation, and only the surplus beyond this constant flux actually contributes to model growth. We had opted to only refer to the single contribution of the ATPase as "NGAM" in our model for simplicity, as these other constant expenses are spread across various reactions. However, we realized that the same holds for the quantifiable part of the GAM; thus, we changed our language in the manuscript to refer to both ATPase- and turnover-related ATP expenses as components of the overall NGAM. We have calculated the growth rate elasticities with respect to each of these three constraints and they all come out rather low: 0.1 (RNA degradation), 0.04 (protein degradation) and 0.03 (ATPase).

5) The authors should use correct terminology when describing predictions from flux balance analysis throughout the manuscript, as it predicts yield; the conversion to growth rate is dictated by what the authors assume for uptake rate which if not measured but is a gross proxy, as indicated above.

While uptake rates might not be available in many cases as also noted by Ron Milo in his comments, our glucose uptake rate is actually taken from a measurement in another mycoplasma (cited in subsection 'Central metabolism') and is in the range measured for *E. coli* with a two hour doubling time (Fuhrer et al., 2005; Labhsetwar et al., 2013) which also has the high affinity glucose transporter PtsG as in JCVI-syn3A.

As to the conversion from yield to growth rate, we revised the corresponding paragraph in the Introduction to emphasize or clarify:

- that the conversion from yield (growth rate per uptake rate) to growth rate requires an uptake rate;

- that while no such values are available for JCVI-syn3A itself, measurements are available for *Mycoplasma pneumoniae* and *E. coli* for a similar high affinity glucose transporter;

- that adopting related uptake rates and not experimentally determined ones leads to a provisional growth rate prediction;

- that when model growth rates are stated henceforth in the manuscript, it comes with the understanding that they are provisional and have a degree of uncertainty due to the sensitivity with regard to the assumed uptake rate (see also response to previous Essential Revision point).

[Editors' note: further revisions were requested prior to acceptance, as described below.]

Reviewer #2:We wish to thank the authors for taking time to address our previous comments. The addition of the whole metabolic map for JCVI Syn3.0 is integrative and should help readers understand the metabolism of the minimal cell. We appreciate that the model predictions were nuanced according to the available data (i.e. steady-state fluxes, in silico gene knockout mapping, etc.). While these comments were properly addressed, a second view of the (lengthy) paper yielded new comments:Abstract:1) The Abstract is overly concise for a paper that spans 76 pages including references. In general, the conclusions exposed in the abstract should be extended and "98% of enzymatic reactions supported by annotation." This seems like a wrong and inflated number that does not reflect the uncertainty in the network. Excluding exchange reactions, biomass, growth-associated maintenance and gas diffusion (O2 and CO2) 36 reactions remain orphan, including many transports. This correction brings the percentage down to 85.6% when applying a denominator of 250 (again excluding the mentioned reactions above).

The full claim in the abstract reads: "98% of enzymatic reactions supported by annotation or experiment." In summary, most of our gap fills are supported by experimental observations; and our number of gap fills itself is actually smaller than stated by the reviewer, as many transport reactions are assumed to be passive, thus not requiring a gene product in the first place.

Table 6 in the main text shows that there are indeed 21 gap fills in the model (please see below for how we arrive at that number); however, 17 of these are supported by experimental evidence, and only four are gap fills based on the network only. (We must have counted one supported gap fill as unsupported before, as the table was showing 16 and five reactions; this has been fixed.) As biochemical evidence provides strong support for inclusion of a reaction (with the caveat that the evidence is based on the parent species), this means that 240 out of our 244 "non-pseudo reactions" (see below for definition), or 98% , are well supported.

As our accounting yields different numbers than those reported by the reviewer, we would like to lay out how we count reactions. As mentioned in the explanation to Table 6, we only consider "non-pseudo reactions", i.e. only reactions that correspond to individual chemical or transport reactions. This excludes exchange, biomass and growth-associated maintenance reactions (in agreement with the reviewer); however, we also exclude the lumped reactions pertaining to macromolecule metabolism. E.g. DNA polymerization has no metabolic genes assigned and would therefore constitute a "gap fill", which would be misleading. We do not exclude any transport reactions, and therefore also include gas transport. By this account, there are 244 non-pseudo reactions: 339 model reactions – 85 exchange reactions – 2 biomass/GAM reactions – 8 macromolecule reactions.

From these 244 non-pseudo reactions, there are 35 reactions without an associated gene; however, as outlined in the metabolic reconstruction results, several transport reactions are assumed to be passive, and therefore assumed to not require a gene product in the first place (see e.g. discussion on passive glycerol uptake). Subtracting these 14 passive transport reactions yields the 21 gap fills listed in Table 6. (We had previously missed to include transport of formate, O_2_, CO_2_, NH_3_ and H_2_O among the passive reactions in Table 6; instead, they were accidentally included in "Annotation-supported", thus not changing the number of gap fills however. They have been included in the passive reactions now.)

It appears the reviewer might have included macromolecule reactions in his count, three of which have no gene assigned. This would yield his stated 250 reactions if O_2_ and CO_2_ transport are excluded from the beginning (244 reactions in our count + 8 macromolecule reactions – 2 transport reactions). It also would yield his stated 36 gap fill reactions (35 reactions without associated gene in our count + 3 “gap fill” macromolecule reactions – 2 transport reactions for O_2_ and CO_2_). As outlined above, the inclusion of macromolecule and passive transport reactions however leads to an overestimation of the actual number of gap fills.

We hope this explains and justifies how we arrive at a count of 98% of non-pseudo reactions supported by annotation or experiment. We have also slightly expanded the discussion around Table 6 to lay out our counting of reactions more clearly. The abstract refers to enzymatic reactions, which technically are a subset of the set of non-pseudo reactions (obtained by omitting the few passive reactions); but the annotation/experiment coverage comes out at 98% (rounded) in either case so we refrain from using the expression "98% of non-pseudo reactions" in the abstract, as the meaning might not be apparent without the definition provided in the main text.

2) "The model agrees well with genome-scale in vivo transposon mutagenesis experiments" Perhaps provide a quantitative statement rather than a qualitative one.

We have added the MCC as quantitative information.

"The genes in the reconstruction have a high in vivo essentiality or quasi-essentiality of 92% , compared to 79% in silico essentiality." While the results presented in supplementary table 2 are consistent with these two numbers, we find 76% in silico essentiality when testing the model provided for single gene deletions. Also, the in vivo essentiality of 92% is obtained when assuming quasi-essential genes as essential and should be specified or the number with only essential genes (68%) should be included or at least mentioned.

The reported apparent discrepancy of nine genes (123 essential genes reported by us vs. 114 reported by the reviewer further down) arises from the eight genes for ATPase (MMSYN1_0789-0796) and the one gene for Lon (MMSYN1_0394); these genes are connected to model reactions carrying a lower flux bound due to their assumed biological importance (maintaining transmembrane PMF/protein degradation). Manually setting the upper bounds on these reaction fluxes to zero in COBRApy 0.4 renders the FBA problem infeasible as expected (due to inconsistent flux bounds). However, the single gene deletion routine in COBRApy sets both flux bounds (upper and lower) to zero, which therefore removes the pre-imposed lower flux bounds. (COBRApy 0.13 also enforces this behavior when manually attempting to set the upper bound to zero.) As the ATPase and Lon reactions do not produce any required biomass precursors, they therefore show up as “non-essential” in the COBRApy routine. The reported in silico essentiality is therefore based on the manual essentiality check for the mentioned genes. (Note: Technically, the flux bound on ATPase is an upper bound of negative value, as this flux direction corresponds to proton efflux; the manual test for ATPase therefore involves setting the lower bound to zero, which again creates an inconsistency.)

Regarding the in vivo essentiality of model-included genes, the sentence in the abstract actually states: "in vivo essentiality or quasi-essentiality of 92% ", clarifying that the 142 genes in question are not all essential. We now also added the subset of 68% essential genes however.

3) "The reconstruction is the starting point for studying the evolution of metabolic subsystems and analyzing the effects of introducing alternative pathways." If claiming this in the abstract perhaps include subsystems to the reactions in the model.

The revised json model file contains subsystems, KEGG identifiers and InChI keys (SBML does not store the "subsystems" and "notes" attributes).

4) "Finally, the identification of 30 essential genes with unknown function will motivate the search for new biological mechanisms beyond metabolism." After reading through the entire paper multiple times I still cannot see where this number is fetched from, so I cannot imagine how a reader could get to it.

This number is taken from Table 1, Subtotal of "Unclear" genes that are essential; this number and its potential pertinence to new biological mechanisms are also mentioned in subsection 'Transposon mutagenesis experiments probe in vivo gene essentiality' on in vivo gene essentiality. For consistency, we have changed the language in the abstract from “unknown function” to “unclear function”.

5) Rather than pointing to a very accurate model of Syn3.0, the abstract should demonstrate that despite being the smallest organism known to date, Syn3.0 has many unknown elements. While the amount of information provided in the paper is highly valuable for the community, presenting the reconstruction as a way to clearly identify the grey areas of the genome is key for the development and substantial efforts have been made in that sense through 1- a detailed manual curation process, 2- proteomic validation of enzyme expression, 3- in vivo gene essentiality. This sentence of the abstract should be of prime importance: "This comparison together with proteomics data yields new hypotheses on gene functions as well as suggestions for several further gene removals".

We thank the reviewer for this remark. We have slightly expanded the abstract to more clearly emphasize how the presented coherent model of the metabolic network at the same time points toward specific questions, and how the model, its comparison to in vivo essentiality and proteomics data guide future work through the specific hypotheses provided.